# Merging Smarter, Generalizing Better: Enhancing Model Merging on OOD Data

## Abstract

Multi-task learning (MTL) concurrently trains a model on diverse task datasets to exploit common features, thereby improving overall performance across the tasks. Recent studies have dedicated efforts to merging multiple independent model parameters into a unified model for MTL, thus circumventing the need for training data and expanding the scope of applicable scenarios of MTL. However, current approaches to model merging predominantly concentrate on enhancing performance within in-domain (ID) datasets, often overlooking their efficacy on out-of-domain (OOD) datasets. In this work, we propose LwPTV (Layer-wise Pruning Task Vector) by building a salience score, measuring the redundancy of parameters in task vectors. Designed in this way LwPTV can construct mask vector for each task and thus perform layer-wise pruning on the task vectors, only keeping the pre-trained model parameters at the corresponding layer in merged model. Owing to its flexibility, our method can be seamlessly integrated with most of existing model merging methods to improve their performance on OOD tasks. Extensive experiments demonstrate that the application of our method results in substantial enhancements in OOD performance while largely maintaining competitive performance on ID tasks.

## 1 Introduction

Pre-trained models (PTMs) constitute a cornerstone of deep learning, supporting numerous contemporary methodologies by virtue of their capacity to extract generalized features from vast data repositories (Bommasani et al., 2021; Zhuang et al., 2020). Typically, the fine-tuning of pre-trained models with task-specific data is employed to enhance performance (Shnarch et al., 2022; Kenton & Toutanova, 2019). This process yields numerous checkpoints originating from the PTMs (Poth et al., 2021b). Fine-tuning individual models for each task incurs substantial storage and deployment costs (Dettmers et al., 2023). Multi-task learning (MTL) offers an alternative by training a single model on multiple tasks, reducing storage demands. However, MTL introduces significant computational overhead and is constrained by privacy concerns due to the necessity of aggregating diverse datasets (Jin et al., 2023). Recently, the trend in research has shifted to an alternative paradigm: merging various individual fine-tuned models into one single multi-task model, bypassing the requirement for the original training data. This approach, known as model merging, addresses these limitations by merging model parameters rather than engaging in further training (Wortsman et al., 2022a; Matena & Raffel, 2022; Jin et al., 2023).

The most typical technique is Weighted Averaging, which directly averages the weights of multiple fine-tuned models (Wortsman et al., 2022a). However, averaging often leads to a significant degradation in performance. To this end, various techniques, including Fisher Merging (Matena & Raffel, 2022), RegMean(Jin et al., 2023), and Task Arithmetic (Ilharco et al., 2023), have been proposed. Although these techniques improve performance, they primarily focus on improving accuracy within ID datasets - the datasets used to fine-tune task-specific models. However, real-world applications often encounter data distributions that differ from those seen during training. The ability of a model to perform well in out-of-domain (OOD) data directly impacts its reliability, robustness, and deployment in dynamic environments such as healthcare, autonomous systems. Despite its importance, OOD robustness remains an underexplored aspect in model merging. As shown in Fig.1, although having better performance on in-domain (ID) data, merged models with current

Table 1: Summarization of model merging methods: "No Training/ Test" means no need Training/Test set in designing merging coefficient or merging models, "Pruning" indicates discarding parameters in task vectors, "Orthogonal" in merging coefficients indicates that method is orthogonal to merging coefficient designed by others, "Designed for OOD" indicates that the method is specifically designed for merging model on OOD, "Applicable to OOD evaluation" indicates that the merging model is available to OOD data.

| Method | No Training set | No Test | Pruning | Merging coefficients | Applicable to OOD evaluation | Designed for OOD | Storage cost |
|---|---|---|---|---|---|---|---|
| Weight Averaging (Wortsman et al., 2022a) | ✓ | ✓ | ✗ | Uniform | ✓ | ✗ | high |
| Fisher Merging (Matena & Raffel, 2022) | ✗ | ✓ | ✗ | Fisher Matrix | ✓ | ✗ | high |
| RegMean (Jin et al., 2023) | ✗ | ✓ | ✗ | Inner Product Matrix | ✓ | ✗ | high |
| Task Arithmetic (Ilharco et al., 2023) | ✓ | ✗ | ✗ | Uniform | ✓ | ✗ | high |
| Ties-Merging (Yadav et al., 2023) | ✓ | ✓ | Magnitude Metric | Uniform | ✓ | ✗ | low |
| AdaMerging (Yang et al., 2024b) | ✓ | ✗ | ✗ | Optimized | ✓ | ✗ | high |
| Surgery (Yang et al., 2024a) | ✓ | ✗ | ✗ | Orthogonal | ✗ | ✗ | high |
| PCB-MERGING (Du et al., 2024) | ✓ | ✓ | PCB matrix | $(\hat{\beta}_k \odot \lambda_k)/\sum_{k=1}^K \hat{\beta}_i$ | ✓ | ✗ | low |
| FR-Merging (Zheng & Wang, 2025) | ✓ | ✓ | ✗ | $\lambda_k \mathbb{E}(\tau_k)\left(\sum_{k=1}^K \mathbb{E}(\tau_k)\right)^{-1}$ | ✓ | ✗ | high |
| LwPTV (Ours) | ✓ | ✓ | Salience Score | Orthogonal | ✓ | ✓ | low |

merging methods usually exhibit significant performance degradation on OOD tasks when compared with pre-trained models, which have stronger generalization capabilities. The primary reason for the performance drop lies in: fine-tuning updates the parameters of pre-trained models to fit ID datasets, often at the cost of perturbing generalizable features (Kumar et al., 2022; Zhu et al., 2024). Therefore, a better balance is needed between preserving generalization and incorporating task-specific adaptations.

In this work, we investigate the feasibility of pruning parameters of task vectors and only keeping the corresponding parameters in pre-trained models to enhance the generalization performance on OOD data of merged models. The central challenge lies in designing a principled criterion for identifying and removing task vector components that do not contribute meaningfully to ID tasks. Recall that task vectors can be interpreted as perturbations that align pre-trained models to specific tasks (Yadav et al., 2023; Yang et al., 2024b;a; Ilharco et al., 2023; Du et al., 2024). Shown in (Li et al., 2025), these vectors encode both discriminative and redundant information. With in-depth analysis, we posit that task-specific discriminative features tend to be diverse across tasks, while redundant or low-signal features exhibit consistency across task vectors, indicating low task-specific relevance. This motivates the hypothesis that

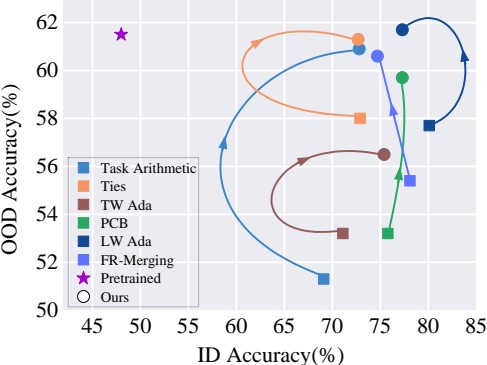

Figure 1: ID and OOD performance of model merging methods on ViT-B/32; see ID and OOD tasks in Experiments. Square and circle markers denote baseline and baseline+ours.

shared patterns in task vectors reflect low-salience, potentially non-discriminative modifications that do not meaningfully contribute to performance, especially in OOD settings. More interestingly, we find that this diversity of task-specific discriminative features is not uniformly distributed across the model, but rather exhibits a layer-wise structure. Based on these insights, we propose an adaptive method, LwPTV (**L**ayer-**w**ise **P**runing **T**ask **V**ector for model merging). We define a layer-wise salience score, quantifying the deviation of a given task vector from the layer-wise mean across tasks. Formally, for each parameter in a layer, this score captures the absolute difference between its task-specific value and the mean across task vectors. A low salience score implies the parameter shift is consistent across tasks and thus likely redundant, motivating its removal in favor of the original pre-trained value. We construct layer-wise masks based on this score and additionally define a shared mask to preserve ID performance by retaining critical task-specific adaptations across all tasks. LwPTV can be used as a plug-and-play approach to enhance OOD generalization of most of model merging methods due to its flexibility. Extensive experiments prove that ours effectively maintains the pre-trained model's generalization ability while leveraging task-specific adaptations, leading to improved OOD robustness while largely maintaining ID performance.

## 2 Related work

**Multi-task Learning.** MTL involves training a model on data from multiple tasks simultaneously, with the aim of leveraging shared representations to enhance performance across all tasks (Vandenhende et al., 2021; Zhang et al., 2023). However, when confronted a new task, MTL requires access to labeled data from

multiple tasks to train from scratch. This leads to high training costs and limited data accessibility due to privacy concerns (Pruksachatkun et al., 2020; Poth et al., 2021a; Weller et al., 2022; Fifty et al., 2021).

**Model Merging.** Model merging (Wortsman et al., 2022a; Matena & Raffel, 2022; Ilharco et al., 2023; Yadav et al., 2023) aims to combine several fine-tuned task-specific models into a single, unified multi-task model, by utilizing the existing task-specific model weights without requiring additional training. Model merging can be classified into two types. The first type involves merging multiple models that were trained on the same task, with the goal of improving the overall generalization of the model (Gupta et al., 2020; Cha et al., 2021; Ainsworth et al., 2023; Singh & Jaggi, 2020). Another type focuses on merging models for different tasks in order to enable MTL (Wortsman et al., 2022a; Ilharco et al., 2023; Yadav et al., 2023; Yang et al., 2024a).

The second type is our focus. A naive method is model averaging technique, which often leads to significant performance degradation (Wortsman et al., 2022a). To address the performance gap between fine-tuned models and the merging model in ID tasks, various methodologies have been proposed. Fisher Merging (Matena & Raffel, 2022) and RegMean (Jin et al., 2023) enhance model merging by utilizing Fisher information matrices and inner-product matrices, respectively, to calculate weighted coefficients for individual models. Task Arithmetic (Ilharco et al., 2023) introduces task vectors, defined as the parameter differences between fine-tuned models and the pre-trained model, which are effective for model merging. As a follow up, some works are further proposed. For example, Ties-Merging (Yadav et al., 2023) aims to tackle the task conflicts among multiple models; AdaMerging (Yang et al., 2024b) introduces the adaptive learning about the merging coefficients; Surgery method (Yang et al., 2024a) addresses representation bias between merged model and task-specific models; DARE (Yu et al., 2024) introduces a preprocessing step called drop and rescale, which reduces interference by randomly eliminating most elements and rescaling the remaining ones in each task vector before merging fine-tuned LLMs; WEMoE (Anke et al., 2024) dynamically combines shared and task-specific knowledge based on the input sample; PCB-MERGING (Du et al., 2024) effectively addresses parameter competition by adjusting parameter coefficients during the merging process; FR-Merging (Zheng & Wang, 2025) leverages the Fourier transform to remove low-frequency interference components from model parameters, effectively mitigating cross-task interference.

The methods discussed above focus on enhancing the ID performance of merging model. While Ties-Merging, AdaMerging, WEMoE, and PCB evaluate their performance on OOD data, their primary goal is still the ID tasks. In contrast, we aim to improve the performance of merged models on OOD tasks while preserving ID task performance. Besides, LwPTV can be achieved only based on task vectors and can be combined with existing task vector-based methods in a flexible way.

**Out-of-Distribution.** A challenge commonly referred to as OOD generalization, continues to pose a substantial challenge in the field of machine learning. Despite the remarkable zero-shot capabilities demonstrated by large pre-trained models, such as CLIP (Li et al., 2022; Radford et al., 2021b), further finetuning on downstream tasks could potentially lead to decreased performance with OOD data (Nguyen et al., 2024; Kumar et al., 2022; Chen et al., 2023; Shuttleworth et al., 2024). Recent studies have proposed methods to mitigate this issue. WiSE-FT (Wortsman et al., 2022b) enhances the OOD performance of fine-tuned models by performing linear interpolation between the fine-tuned model and its corresponding pre-trained model. Model Stock exploits the anchoring effect of pretrained models and the geometric properties of fine-tuning parameters to utilize two fine-tuned models, obtained through optimization with different random seeds for the same task, to approximate the center of the parameter distribution (Jang et al., 2024). LiNeS (Wang et al., 2024a) employs a depth-dependent scaling strategy for parameter updates. These strategies aim to enhance the fine-tuned model's performance on OOD tasks. Different from them, ours aims to enhance the generalization capabilities of merging models, where we design the plug-and-play salience score to prune redundant parameters of task vectors in a layer-wise manner.

## 3  Preliminaries

**Problem setup.** Denote the pre-trained model as $f(\mathbf{x}; \boldsymbol{\theta}_{pre})$, where $\boldsymbol{\theta}_{pre} = \{\boldsymbol{\theta}_{pre}^1, ... \boldsymbol{\theta}_{pre}^l, ... \boldsymbol{\theta}_{pre}^L\}$, $L$ is the number of layers and $\boldsymbol{\theta}_{pre}^l$ is the parameter of the $l$-th layer. We aim to fine-tune the model on $K$

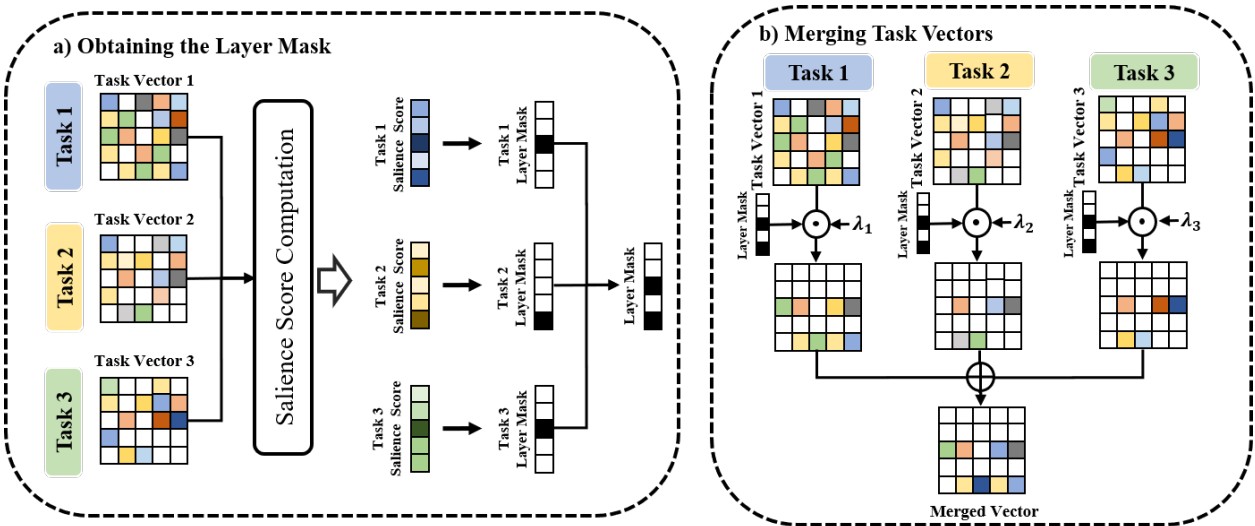

Figure 2: Illustration of our **LwPTV** framework: (a) Obtaining the Layer Mask; (b) Merging Task Vectors. Each row in the task-vector block corresponds to a layer, where white indicates 0 and black indicates 1. We compute layer-wise salience scores, threshold them to obtain masks, prune each task vector accordingly, and finally merge the pruned vectors into the pretrained model.

downstream tasks $\{T_k\}_{k=1}^K$ to get $K$ finetuned models $\{f(\mathbf{x};\boldsymbol{\theta}_K)\}_{k=1}^K$, where each finetuned model has the same size parameters with pretrained model and $\boldsymbol{\theta}_k = \{\boldsymbol{\theta}_k^1, ...\boldsymbol{\theta}_k^l, ...\boldsymbol{\theta}_k^L\}$. Model merging aims to combine the weights $\{\boldsymbol{\theta}_k\}_{k=1}^K$ into a new set of weights $\boldsymbol{\theta}_m$, enabling $f(\mathbf{x};\boldsymbol{\theta}_m)$ to perform $K$ tasks without retraining using task-specific data.

**Task vector-based methods.** A recent study (Ilharco et al., 2023) proposed the idea of "task vectors" for model merging, which has been further explored by subsequent research (Yadav et al., 2023; Yang et al., 2024b). A task vector $\boldsymbol{\tau}_k$ is defined as the difference between the fine-tuned model parameters $\boldsymbol{\theta}_k$ and the initial pre-trained parameters $\boldsymbol{\theta}_{pre}$. Specifically, for the $k$-th task, its task vector is given as:

$$\boldsymbol{\tau}_k = \boldsymbol{\theta}_k - \boldsymbol{\theta}_{pre}, \quad \boldsymbol{\tau}_k = \{\boldsymbol{\tau}_k^1, ...\boldsymbol{\tau}_k^l, ...\boldsymbol{\tau}_k^L\}. \tag{1}$$

Now, based on the task vectors, the merged function can be expressed as $\mathcal{F}(\cdot)$ as follows:

$$\boldsymbol{\theta}_m = \mathcal{F}(\boldsymbol{\theta}_{pre}, \boldsymbol{\tau}_1, \boldsymbol{\tau}_2, \cdots, \boldsymbol{\tau}_K). \tag{2}$$

For example, in terms of Task Arithmetic, the merged model weights can be expressed as $\boldsymbol{\theta}_m = \boldsymbol{\theta}_{pre} + \lambda \sum_{k=1}^K \boldsymbol{\tau}_k$, where $\lambda$ denotes the merging coefficient. As for Task-wise AdaMerging, the merged model weights are computed as $\boldsymbol{\theta}_m = \boldsymbol{\theta}_{pre} + \sum_{k=1}^K \lambda_k \boldsymbol{\tau}_k$, where $\lambda_k$ denotes the merging coefficient corresponding to the task vector $\boldsymbol{\tau}_k$. However, existing methods primarily focus on improving the accuracy of ID data, neglecting the performance on out-of-distribution OOD data.

# 4 Our proposed method

This work introduces a novel plug and play method, denoted as Layer-wise Pruning Task Vector for model merging (LwPTV), to improve the generalization capabilities of merging models, whose overview is shown in Fig.2. We first give the motivation and analysis about the task vector in Sec.4.1 and then introduce the proposed method in Sec.4.2.

## 4.1 Motivation and analysis about task vector

As shown in Fig.1, the merged model exhibits inferior generalization capabilities on OOD tasks but superior performance on ID tasks when compared with the pre-trained model. According to Eq. 2, merged model $\boldsymbol{\theta}_m$ is composed of $\boldsymbol{\theta}_{pre}$ and $\{\boldsymbol{\tau}_k\}_{k=1}^K$. The pre-trained model mainly contains valuable parameters for handling

OOD tasks as indicated by Fig.1, and $\boldsymbol{\tau}_k$ have the parameters beneficial for the $k$-th ID task. Motivated by this observation, our idea to improve OOD generalization for merged models by discarding certain redundant parameter layers within the task vectors and replacing them with the corresponding layers from the pretrained model, making the merged models "closer" to pretrained models where necessary. By doing this, we face two fundamental questions: (1) **Which parameters from task vectors should be pruned?** (2) **Can the pruning maintain the performance of merged model on ID tasks?**

Before answering these questions, we provide a theoretical analysis of the task vector inspired by Li et al. (2025), a state-of-the-art theoretical work on the generalization of task vectors. Li et al. (2025) theoretically analyzes one-layer Transformer models based on discriminative patterns that are unique for different tasks. From Corollary 2 and Lemma 1 of Li et al. (2025), one can see that, some neurons of the task vectors for Transformer models can learn the discriminative patterns, while other neurons cannot. Without loss of generality, our theoretical analysis focuses on Transformer models e.g., ViT (Dosovitskiy et al., 2020) and considers one layer, where the task vector in (2) is simplified to $\boldsymbol{\tau}_k = \boldsymbol{\tau}_k^1$. Let $g(\boldsymbol{\tau}_k, \mathcal{S})$ denote the weights of the neuron at the index set $\mathcal{S} \subset \mathcal{L}$ in the task vector $\boldsymbol{\tau}_k$, where $\mathcal{L}$ is the set of all neuron indices. Define the diversity $(DV)$ of $\boldsymbol{\tau}_k$ at neurons from $\mathcal{S}$, with respect to a collection $\{\boldsymbol{\tau}_j\}_{j=1}^K$, as

$$\mathrm{DV}\left(g(\boldsymbol{\tau}_k, \mathcal{S}); \{\boldsymbol{\tau}_j\}_{j=1}^K\right) = \mathbb{E}\left[\left\|g(\boldsymbol{\tau}_k, \mathcal{S}) - \frac{1}{K}\sum_{j=1}^K g(\boldsymbol{\tau}_j, \mathcal{S})\right\|\right]. \tag{3}$$

Then, we can derive the following proposition, the proof of which can be found in Appendix A.1.

**Proposition 1.** *With a high probability, there exists a set of neurons with indices in $\mathcal{S} \in \mathcal{L}$, $|\mathcal{S}| \geq 1$, where all neurons in $\mathcal{S}$ for $\boldsymbol{\tau}_{k_1}$ learn discriminative features, while all neurons in $\mathcal{S}$ for $\boldsymbol{\tau}_{k_2}$ fail to learn discriminative features, $k_1 \neq k_2 \in [K]$. Then, we have*

$$DV(g(\boldsymbol{\tau}_{k_1}, \mathcal{S}); \{\boldsymbol{\tau}_j\}_{j=1}^K) > \sqrt{K} \cdot \Omega(DV(g(\boldsymbol{\tau}_{k_2}, \mathcal{S}); \{\boldsymbol{\tau}_j\}_{j=1}^K)) \tag{4}$$

This result formalizes the intuition that discriminative neurons (parameters) of task vectors exhibit significantly higher variability across tasks compared to non-discriminative ones. It thus provides theoretical motivation for using diversity as a criterion for selective pruning. In particular, parameters with low diversity can be considered as potentially less task-specific components, which may contain redundant or less informative updates. Consequently, pruning these low-diversity components from $\boldsymbol{\tau}_k$ and replacing them with their counterparts from the pre-trained model $\boldsymbol{\theta}_{pre}$ is unlikely to degrade performance on ID tasks. Instead, it preserves ID performance while recovering the generalization ability of the merged model by reintroducing OOD-relevant information encoded in $\boldsymbol{\theta}_{pre}$.

## 4.2 Layer-wise pruning of task vectors

Based on the aforementioned analysis, we consider a straightforward method to enhance the generalization ability of merging models while preserving its ID performance: Pruning the parameters in task vectors with low diversity. Recalling that in the deep neural network model, the information learned by each layer is usually different, and the different layers in each task vector usually have different contributions for the merging model as stated by Yang et al. (2024b). Motivated by Eq.3, we introduce a layer-wise salience score:

$$s_k^l = \mathbb{E}\left[\left\|\boldsymbol{\tau}_k^l - \frac{1}{K}\sum_{j=1}^K \boldsymbol{\tau}_k^l\right\|\right], s_k^l \in \mathbb{R}_{\geq 0}, \tag{5}$$

where $s_k^l$ represents the salience score of the $l$-th layer of the $\boldsymbol{\tau}_k$. In Eq.5, we first compute the difference between $k$-th task vector and mean of all task vectors in layer $l$, which is a $d_l$-dimensional vector with $d_l$ denoting the dimension of $\boldsymbol{\tau}_k^l$; then we compute the absolute average of the vector along the dimension $d_l$, resulting in a scalar value $s_k^l$. We use $s_k^l$ as an empirical heuristic, motivated by task-vector diversity, to estimate the task-specific relevance of the $l$-th layer in $\tau_k$. A larger deviation from the cross-task mean leads

to a higher $s_k^l$, suggesting that the corresponding layer contains more task-specific information. Conversely, a smaller score indicates that the update is more consistent across tasks and may contain less task-specific or potentially redundant information, making it a candidate for pruning.

To solve the first problem discussed above, we denote the mask vector as a $L$-dimensional vector, i.e., $\boldsymbol{m}_k = \{m_k^l\}_{l=1}^L \in \mathbb{R}^L$ for the $k$-th task and we design it according to salience scores:

$$m_k^l = \begin{cases} 1, & if \ \ s_k^l > \text{sorted}(\mathbf{s}_k)[\lfloor L \cdot \eta \rfloor], \\ 0, & \text{otherwise}, \end{cases} \tag{6}$$

where $\eta$ is the hyperparameter controlling the pruning ratio, and $\text{sorted}(\mathbf{s}_k)[\lfloor L \cdot \eta \rfloor]$ represents the $\lfloor L\eta \rfloor$-th smallest salience score, ensuring that only the top $1 - \eta$ fraction of layers with the highest salience scores are retained. In other words, for each task vector, layers whose salience scores significantly larger are retained, while layers with lower salience scores—indicating redundant parameters—are replaced with the corresponding pre-trained parameters. This adaptive pruning mechanism ensures that the merged model maintains task-specific adaptations while leveraging the generalization capabilities of the pre-trained model, thereby enhancing OOD robustness. Considering the mask defined independently for each task vector, there remains a risk that some task-specific mask vectors may remove parameter layers that are crucial for their respective tasks. To solve the second question mentioned earlier, i.e, maintaining the ability of the merged model on ID tasks as much as possible, we introduce a shared mask vector that consolidates information across all mask vectors:

$$\hat{\mathbf{m}} = \mathbf{m}_1 \vee \mathbf{m}_2 \vee \cdots \vee \mathbf{m}_K, \tag{7}$$

where $\hat{\mathbf{m}}_k \in \mathbb{R}^L$ and $\vee$ is the OR operation. That is to say, a layer $l$ in the merged model will be pruned only if all task-specific mask vectors agree to remove it, i.e., $\hat{m}_k^l = 0$ if and only if $m_{1:K}^l = 0$. This ensures that if at least one task requires a particular layer to be retained, we remain it in the merged model, thereby preserving critical task-specific information.

Our proposed masking strategy is flexible so that it can be integrated with existing task vector-based merging methods. Specifically, it can be applied to the Task Arithmetic, where the merging model $\hat{\boldsymbol{\theta}}$ is formulated as follows:

$$\hat{\boldsymbol{\theta}}_m = \boldsymbol{\theta}_{pre} + \hat{\mathbf{m}} \odot \sum_{k=1}^K \lambda_k \boldsymbol{\tau}_k. \tag{8}$$

In which, the $\{\lambda_k\}_{k=1}^K$ values are identical. Besides, this mask vector can also be applied to other methods. For example, for AdaMerging series methods, which are designed to optimize coefficient $\lambda_k$ or $\lambda_k^l$ with entropy minimization, we can optimize $\lambda_k$ or $\lambda_k^l$ based on equation 8. Considering the learnable $\lambda_k^l$ will be amplified after introducing the mask, we can further use $\hat{\lambda}_k^l = \eta \cdot \lambda_k^l$ to scale the coefficient after optimizing the equation 8. A detailed analysis is in the B.1 of Appendix. We defer the workflow when combining ours with existing methods to Algorithm B.2 of Appendix.

**Remark 1.** *By combining Proposition 1 with the definition of the salience score in Eq. (5), parameters that exhibit greater task-specific variation are expected to receive larger salience scores. The corresponding layer is therefore more likely to be retained, as it may contain useful task-specific information. In contrast, parameters with smaller cross-task variation tend to receive lower salience scores and are treated as potentially less task-specific or redundant updates, making the corresponding layer a candidate for pruning. This analysis provides theoretical motivation for using task-vector diversity to guide pruning.*

## 5 Experiments

**Datasets.** Following prior work (Ilharco et al., 2023; Yadav et al., 2023; Yang et al., 2024b), we conduct model merging on eight image classification datasets: Cars (Krause et al., 2013), DTD (Cimpoi et al., 2014), EuroSAT (Helber et al., 2019), GTSRB (Stallkamp et al., 2011), MNIST (Deng, 2012), RESISC45 (Cheng et al., 2017), SUN397 (Xiao et al., 2016), and SVHN (Netzer et al., 2011). To assess OOD generalization, we evaluate the merged models on 13 OOD benchmarks (Wang et al., 2024b; Zhou et al., 2022): CIFAR10 (C10) (Krizhevsky et al., 2009), CIFAR100 (C100) (Krizhevsky et al., 2009), EMNIST (EMN) (Cohen et al., 2017),

Table 2: Performance comparison for ID and OOD datasets on ViT-B/32 and ViT-L/14.

| | Dataset (→) Method (↓) | In Domain Avg. | Out of Domain C10 | C100 | EMN | FMN | FER | F102 | F101 | KMN | OxP | PCA | RSS | STL | Ima | Avg. | H-score |
|---|---|---|---|---|---|---|---|---|---|---|---|---|---|---|---|---|---|
| | Pretrained | 48.0 | 89.8 | 64.2 | 17.2 | 63.0 | 39.0 | 66.3 | 82.7 | 9.8 | 87.4 | 60.6 | 58.6 | 97.1 | 63.3 | 61.5 | 53.9 |
| | Individual | 90.5 | 65.5 | 37.3 | 18.2 | 54.8 | 32.2 | 47.8 | 58.3 | 8.5 | 76.9 | 54.1 | 55.9 | 88.4 | 48.5 | 49.7 | 64.2 |
| | Weight Averaging (Wortsman et al., 2022a) | 65.8 | 89.5 | 64.2 | 28.8 | 65.1 | 36.4 | 64.5 | 79.7 | 7.4 | 86.7 | 59.4 | 57.7 | 96.4 | 61.6 | 61.3 | 63.5 |
| | Fisher Merging (Matena & Raffel, 2022) | 68.3 | 88.7 | 63.3 | 29.4 | 63.9 | 35.6 | 62.9 | 77.9 | 7.4 | 86.1 | 59.9 | 57.9 | 96.0 | 60.9 | 60.8 | 64.3 |
| | RegMean (Jin et al., 2023) | 71.8 | 84.6 | 56.7 | 25.1 | 64.6 | 36.8 | 64.7 | 78.6 | 8.7 | 85.3 | 60.0 | 56.1 | 95.3 | 60.1 | 59.7 | 65.2 |
| | Task Arithmetic (Ilharco et al., 2023) | 69.1 | 76.3 | 41.9 | 28.5 | 63.9 | 26.3 | 49.4 | 55.8 | **9.0** | 75.4 | 54.0 | 53.4 | 87.8 | 45.0 | 51.3 | 58.9 |
| | **w/ LwPTV** (Ours) | **72.8** | **88.3** | **60.1** | **30.4** | **65.9** | **36.5** | **62.9** | **79.9** | 7.4 | **85.4** | **59.5** | **59.0** | **96.4** | **60.6** | **60.9**$^{+9.6}$ | **66.3**$^{+7.4}$ |
| ViT-B/32 | Ties (Yadav et al., 2023) | **72.9** | 85.3 | 55.2 | 26.5 | 64.1 | 33.6 | 59.0 | 74.0 | **8.5** | 83.0 | 58.4 | 55.6 | 94.4 | 56.3 | 58.0 | 64.6 |
| | **w/ LwPTV** (Ours) | 72.7 | **89.1** | **61.5** | **29.3** | **65.7** | **37.0** | **63.5** | **80.4** | 7.4 | **85.5** | **59.8** | **59.4** | **96.6** | **61.1** | **61.3**$^{+3.3}$ | **66.5**$^{+1.9}$ |
| | TW AdaMerging (Yang et al., 2024b) | 71.1 | 78.6 | 45.9 | 30.6 | **63.5** | 28.7 | 51.1 | 59.9 | **8.2** | 79.0 | 53.4 | 54.0 | 89.1 | 49.5 | 53.2 | 60.9 |
| | **w/ LwPTV** (Ours) | **75.4** | **81.2** | **47.0** | **31.3** | 62.6 | **32.3** | **56.3** | **73.1** | 7.7 | **81.2** | **55.7** | **57.8** | **93.7** | **54.1** | **56.5**$^{+3.3}$ | **64.6**$^{+3.7}$ |
| | TW AdaMerging++ (Yang et al., 2024b) | 73.7 | 79.3 | 45.5 | 28.1 | **63.2** | 30.4 | 51.7 | 64.3 | **8.3** | 78.1 | 54.6 | 54.1 | 90.6 | 50.6 | 53.7 | 62.1 |
| | **w/ LwPTV** (Ours) | **76.1** | **83.2** | **49.1** | **30.7** | 62.8 | **32.5** | **55.4** | **73.8** | 8.0 | **79.9** | **56.3** | **56.7** | **94.2** | **53.5** | **56.6**$^{+2.9}$ | **64.9**$^{+2.8}$ |
| | LW AdaMerging (Yang et al., 2024b) | **80.1** | 82.1 | 50.7 | 28.7 | 63.2 | 37.8 | 58.3 | 73.8 | **8.8** | 82.4 | 57.3 | 58.0 | 94.3 | 54.6 | 57.7 | 67.1 |
| | **w/ LwPTV** (Ours) | 77.3 | **88.8** | **61.3** | **30.6** | **66.1** | **38.4** | **64.3** | **80.6** | 7.6 | **86.3** | **60.1** | **59.4** | **96.6** | **61.2** | **61.7**$^{+4.0}$ | **68.6**$^{+1.5}$ |
| | LW AdaMerging++ (Yang et al., 2024b) | **81.1** | 83.9 | 51.7 | 27.2 | 62.7 | 36.1 | 57.4 | 74.2 | **9.1** | 82.4 | 56.4 | 57.8 | 94.3 | 54.7 | 57.6 | 67.4 |
| | **w/ LwPTV** (Ours) | 77.1 | **89.0** | **61.8** | **29.9** | **65.8** | **37.5** | **63.9** | **80.4** | 7.8 | **86.0** | **59.6** | **58.9** | **96.5** | **61.1** | **61.4**$^{+3.8}$ | **68.4**$^{+1.0}$ |
| | PCB-MERGING (Du et al., 2024) | 75.8 | 77.7 | 43.6 | 26.0 | 61.2 | 29.4 | 52.9 | 65.4 | **8.8** | 79.3 | 53.6 | 52.2 | 90.5 | 50.8 | 53.2 | 62.5 |
| | **w/ LwPTV** (Ours) | **77.3** | **86.5** | **57.1** | **29.8** | **64.3** | **35.9** | **60.7** | **78.2** | 7.6 | **84.6** | **58.4** | **57.7** | **95.6** | **59.5** | **59.7**$^{+6.5}$ | **67.4**$^{+4.9}$ |
| | FR-Merging (Zheng & Wang, 2025) | **78.1** | 76.6 | 45.9 | 23.8 | 61.7 | 38.5 | 54.8 | 69.6 | **8.9** | 82.6 | 55.8 | 55.4 | 90.7 | 55.5 | 55.4 | 64.8 |
| | **w/ LwPTV** (Ours) | 74.7 | **87.1** | **60.3** | **26.7** | **64.5** | **38.7** | **63.8** | **79.1** | 7.2 | **86.4** | **60.4** | **56.5** | **95.6** | **61.5** | **60.6**$^{+5.2}$ | **66.9**$^{+2.1}$ |
| | Pretrained | 64.5 | 95.6 | 75.8 | 15.6 | 66.9 | 38.2 | 79.2 | 93.1 | 10.4 | 93.4 | 51.2 | 68.9 | 99.4 | 75.5 | 66.4 | 65.4 |
| | Individual | 94.2 | 88.6 | 65.2 | 19.3 | 66.6 | 38.5 | 73.7 | 89.4 | 10.2 | 92.4 | 59.4 | 62.4 | 97.8 | 72.5 | 64.3 | 76.4 |
| | Weight Averaging (Wortsman et al., 2022a) | 79.6 | 95.9 | 79.6 | 22.2 | 71.5 | 38.7 | 79.0 | 92.4 | 8.7 | 93.6 | 59.0 | 64.1 | 99.2 | 75.8 | 67.7 | 73.2 |
| | Fisher Merging (Matena & Raffel, 2022) | 82.2 | 96.0 | 80.0 | 22.6 | 71.8 | 38.7 | 78.8 | 92.3 | 8.6 | 93.5 | 60.0 | 63.1 | 99.1 | 75.6 | 67.7 | 74.2 |
| | RegMean (Jin et al., 2023) | 83.7 | 95.5 | 78.7 | 24.2 | 71.9 | 38.1 | 79.0 | 92.3 | 8.5 | 93.4 | 56.7 | 63.9 | 99.1 | 75.8 | 67.5 | 74.7 |
| | Task Arithmetic (Ilharco et al., 2023) | 84.5 | 93.4 | 73.0 | **28.7** | 73.2 | 38.2 | 74.2 | 88.6 | **9.3** | 92.7 | **61.4** | 58.6 | 98.2 | 72.9 | 66.3 | 74.3 |
| ViT-L/14 | **w/ LwPTV** (Ours) | **85.6** | **95.6** | **78.8** | 24.4 | **73.4** | **40.1** | **77.6** | **92.5** | 8.4 | **93.8** | 60.2 | **63.4** | **99.1** | **76.0** | **67.9**$^{+1.6}$ | **75.7**$^{+1.4}$ |
| | Ties (Yadav et al., 2023) | 86.0 | 95.1 | 74.6 | **27.2** | 72.5 | 38.8 | 75.1 | 90.3 | 8.5 | 92.9 | **60.5** | 59.7 | 98.5 | 74.2 | 66.7 | 75.1 |
| | **w/ LwPTV** (Ours) | **86.9** | **95.1** | **76.9** | 26.2 | **73.2** | **39.6** | **76.5** | **92.3** | **8.7** | **93.7** | 59.5 | **63.7** | **99.0** | **75.7** | **67.7**$^{+1.0}$ | **76.1**$^{+1.0}$ |
| | TW AdaMerging (Yang et al., 2024b) | 84.3 | **90.9** | **67.1** | 25.2 | **71.9** | 36.6 | **69.3** | 85.2 | 9.2 | 91.8 | **59.4** | 57.2 | 97.2 | 69.6 | 63.9 | 72.7 |
| | **w/ LwPTV** (Ours) | **89.0** | 90.4 | 65.4 | **25.4** | 70.7 | **39.7** | 68.5 | **87.5** | 9.2 | **92.7** | 58.6 | **63.2** | **97.3** | **70.6** | **64.5**$^{+0.6}$ | **74.8**$^{+2.1}$ |
| | TW AdaMerging++ (Yang et al., 2024b) | 87.5 | **91.3** | 66.7 | 24.5 | **71.0** | 36.9 | **69.1** | 86.7 | 8.2 | 91.8 | 57.5 | 57.6 | 97.5 | 70.2 | 63.7 | 75.1 |
| | **w/ LwPTV** (Ours) | **89.7** | **91.3** | **67.0** | **25.8** | 70.6 | **39.2** | 68.6 | **88.4** | **8.4** | **92.6** | **57.8** | **65.3** | **97.6** | **71.2** | **64.9**$^{+1.2}$ | **75.3**$^{+1.6}$ |
| | LW AdaMerging (Yang et al., 2024b) | **90.8** | 92.2 | 69.5 | **29.9** | 72.1 | 38.5 | 70.8 | 90.1 | 8.6 | 92.6 | **60.1** | 63.3 | 98.0 | 72.5 | 66.0 | 76.4 |
| | **w/ LwPTV** (Ours) | 89.8 | **95.1** | **77.0** | 26.2 | **73.3** | **40.4** | **75.2** | **92.1** | **8.7** | **93.6** | 59.9 | **65.5** | **98.9** | **75.4** | **68.4**$^{+2.4}$ | **77.7**$^{+1.3}$ |
| | LW AdaMerging++ (Yang et al., 2024b) | **91.0** | 93.5 | 71.8 | **29.9** | 72.3 | 38.4 | 71.2 | 90.3 | 8.7 | 92.8 | **60.6** | 60.4 | 98.2 | 72.7 | 66.2 | 76.6 |
| | **w/ LwPTV** (Ours) | 89.7 | **95.1** | **77.3** | 28.0 | **73.1** | **40.1** | **75.0** | **92.1** | **8.7** | **93.6** | 59.3 | **65.7** | **99.0** | **75.3** | **67.9**$^{+1.7}$ | **77.3**$^{+0.7}$ |
| | PCB-MERGING (Du et al., 2024) | 87.6 | 93.0 | 70.9 | **27.6** | 72.3 | 38.8 | 74.0 | 89.6 | **9.1** | 92.7 | **61.1** | 57.8 | 98.3 | 73.5 | 66.0 | 75.3 |
| | **w/ LwPTV** (Ours) | **88.1** | **94.6** | **75.3** | 27.5 | **73.0** | **39.8** | **76.2** | **91.9** | 8.8 | **93.5** | 60.1 | **62.6** | **98.9** | **75.5** | **67.5**$^{+1.5}$ | **76.4**$^{+1.1}$ |
| | FR-Merging (Zheng & Wang, 2025) | **88.3** | 92.3 | 71.3 | 27.3 | 71.6 | 38.9 | 75.7 | 90.9 | **10.1** | 93.6 | **59.9** | 61.7 | 98.5 | 74.5 | 66.6 | 75.9 |
| | **w/ LwPTV** (Ours) | 87.8 | **94.0** | **75.3** | **28.6** | **72.5** | **39.2** | **76.9** | **92.0** | 9.1 | **93.9** | 59.6 | **62.9** | **98.8** | **75.5** | **67.6**$^{+1.0}$ | **76.4**$^{+0.5}$ |

FashionMNIST (FMN)(Xiao et al., 2017), FER2013 (FER) (Goodfellow et al., 2013), Flowers102 (F102) (Nilsback & Zisserman, 2008), Food101 (F101) (Bossard et al., 2014), KMNIST (Clanuwat et al., 2018) (KMN), OxfordIIITPet (OxP) (Parkhi et al., 2012), PCAM (PCA) (S.Veeling et al., 2018), RenderedSST2 (RSS) (Socher et al., 2013; Radford et al., 2019), STL10 (STL) (Coates et al., 2011) and ImageNet-1k (Ima) (Deng et al., 2009). Additional dataset details are provided in Appendix B.3. Since CLIP's instance-level pre-training data are not publicly available, we cannot verify whether these evaluation datasets overlap with the pre-training corpus. Therefore, we consider these benchmarks as relative OOD evaluations with respect to the downstream fine-tuning tasks.

**Baselines & details.** We consider the following baselines, including Weight Averaging (Wortsman et al., 2022a), Fisher Merging (Matena & Raffel, 2022), RegMean (Jin et al., 2023), Task Arithmetic (Ilharco et al., 2023), Ties-Merging (Yadav et al., 2023), AdaMerging (Yang et al., 2024b), Surgery (Yang et al., 2024a), PCB-MERGING (Du et al., 2024) and FR-Merging (Zheng & Wang, 2025). We employ ViT-B/32, ViT-L/14 and ViT-H/14 from CLIP (Radford et al., 2021a) as our pre-trained models, respectively. Motivated by (Zhu et al., 2024), we employ the H-score as evaluation metric, defined as the harmonic mean of the performance on ID and OOD datasets and stated in Appendix B.4. For image classification tasks, $\eta = 0.7$ is used as a strong default. We select this value on ViT-B/32 based on ID performance without using OOD data, and keep it fixed in all remaining image classification experiments. Detailed configurations are provided in Appendix B.25.

## 5.1 Main results

**Comparison with other methods.** We summarize the characteristics of baselines and ours in Tab.1 for a clear comparison. As listed in Tab.2, we evaluate the performance of merging model ViT-B/32 and

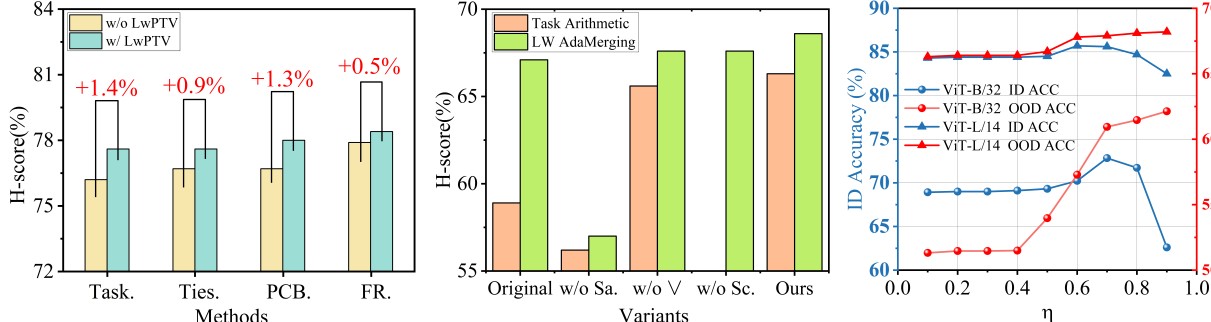

Figure 3: H-score of existing model merging methods with and without ours on ViT-H/14.

Figure 4: Ablation on ViT-B/32; Sa, ∨, and Sc denote Salience score, OR, and scaling.

Figure 5: Performance of Task Arithmetic+LwPTV with varying $\eta$.

ViT-L/14 on ID and OOD datasets respectively. Individual indicates $K$ task-specific fine-tuned model parameters $\{\boldsymbol{\theta}_k\}_{k=1}^K$ without model merging. For ID tasks, each task is only tested by the corresponding individual model; for OOD tasks, each OOD task can be evaluated by all individual models, respectively, followed by the average operation. The Surgery is not available in OOD tasks since it needs to learn a representation surgery module on ID task. The detailed performance of Surgery on the ID dataset are deferred to Appendix B.5 due to limited space.

From Tab.2, we have the following observations: (1) For the non-model merging baseline, the Pretrained model exhibits robust generalization capabilities on OOD datasets, whereas the Individual fine-tuned models demonstrates inferior OOD performance and superior ID performance. The underlying cause is that the Pretrained model acquires generalized features from the large-scale pre-training datasets and the Individual Fine-tuned model undergoes feature distortion specific to the current task. (2) For the model merging baselines, Weight Averaging is equivalent to computing the mean of various task vectors and incorporating these into the pretrained model. This method minimizes alterations to the parameters of the pre-trained model, thereby yielding superior OOD generalization performance. Ties prunes task vectors, yielding good OOD performance but relatively weak ID results. TW/LW AdaMerging and PCB aggressively reweight task vectors, and FR-Merging amplifies task-specific high-frequency signals, which improves ID accuracy but harms OOD robustness due to heavy perturbations to pretrained parameters. (3) LwPTV is a plug-and-play technique that enhances the generalization of existing model merging schemes on OOD datasets while minimizing the degradation on (even improving) the ID performance. For instance, on ViT-B/32, LwPTV yields additional improvements of 7.4%, 1.9%, 1.5%, 4.9%, and 2.1% in H-score values compared to the baseline results of Task Arithmetic, Ties, LW AdaMerging, PCB and FR-Merging, respectively. (4) To investigate whether LwPTV is effective for larger pretrained models, we conducted experiments on ViT-H/14 in Fig.3, proving its effectiveness and robustness on pre-trained model; see more details in Appendix B.6.

**Effect of our components.** We conduct ablation studies based on Task Arithmetic and LW AdaMerging with ViT-B/32. LwPTV includes three components: salience score, OR operation ($\vee$), and coefficient scaling (scale) for LW AdaMerging, which learns the layer-wise merging weights. To systematically investigate the contribution of each individual component, we remove each component and observe the resultant impact. Especially, for w/o salience score, we design the pruning metric based on the mean absolute value of the task vectors across each layer i.e., $\mathbb{E}(|\boldsymbol{\tau}_k^l|)$. The results are illustrated in Fig.4; see details in Tab.10 and Tab.11 of Appendix. We can find that w/o salience score but using $\mathbb{E}(|\boldsymbol{\tau}_k^l|)$ can degrade the H-score a lot, where the performance drop is mainly on ID. Besides, those variants with salience score are better than original merging model methods, indicating the effectiveness of our designed salience score in balancing ID and OOD. In addition, w/o OR operation means that we discard more layer-wise parameters in task vectors depending only on salience score, which also reduces the H-score of merged model. When combining ours with LW AdaMerging, removing scaling strategy is also harmful for the merged model. It proves the effectiveness of our proposed OR operation and scaling strategy.

**Comparison with OOD methods for fine-tuned models.** To explore the advantages of ours, we consider following methods specifically designed for enhancing the OOD performance of fine-tuned models: WiSE-FT (Wortsman et al., 2022b), Model Stock (Jang et al., 2024), and LiNeS (Wang et al., 2024a). Among

them, WiSE-FT and Model Stock are specifically designed for the CLIP architecture, and LiNeS targets the ViT architecture within CLIP. We implement these methods on fine-tuned models and then merge the multiple fine-tuned models with Task Arithmetic. As listed in Table 3, all methods can enhance both ID and OOD performance of the merged model, but ours performs best. As a strong baseline, LiNeS outperforms LwPTV in terms of ID performance, which can be attributed to its amplification of task-specific parameters. However, it is still inferior to LwPTV about the OOD performance and overall H-score. Additionally, for different methods and architectures, LiNeS requires the validation set to determine the optimal scaling coefficient, whereas our pruning ratio of 0.7 is generally applicable. This further demonstrates the advantage of LwPTV in balancing the ID and OOD performance of the merged model. See more details in Appendix B.8.

**Comparison with other pruning methods.** To investigate the superiority of our method over other pruning methods, we employ the following pruning baselines: (1) DARE (Yu et al., 2024), Drops delta parameters with a ratio p And REscales; (2) Magnitude-based Weight Pruning (MWP) (Sanh et al., 2020), which determines the mask based on the absolute value of each parameter within task vector; (3) random

Table 3: Comparison with other OOD methods (above) and pruning methods (below) on ViT-B/32.

| | Performance (%) ($\rightarrow$)
Method ($\downarrow$) | ID. Avg | OOD. Avg | H-score |
|---|---|---|---|---|
| Baseline | Task Arithmetic (Ilharco et al., 2023) | 69.1 | 51.3 | 58.9 |
| OOD | w/ WiSE-FT (Wortsman et al., 2022b) | 70.5 | 55.4 | 62.0 |
| | w/ Model Stock (Jang et al., 2024) | 72.0 | 58.7 | 64.7 |
| | w/ LiNeS (Wang et al., 2024a) | 74.1 | 58.0 | 65.1 |
| Pruning | w/ DARE (Yu et al., 2024) | 49.1 | 61.5 | 54.6 |
| | w/ MWP (Sanh et al., 2020) | 71.7 | 56.4 | 63.1 |
| | w/ random mask | 49.1 | 61.7 | 54.7 |
| | w/ absolute value | 48.9 | 61.5 | 54.5 |
| **Ours** | **w/ LwPTV** | 72.8 | 60.9 | **66.3** $^{(+7.4)}$ |

mask, wherein the mask is stochastically generated via a Bernoulli distribution; (4) the mean absolute value of the task vectors per layer, denoted as absolute value. As depicted in Tab.3, LwPTV's H-score exceeds that of the most favorable Magnitude-based Weight Pruning among the four baselines by 3.2%. Although random mask exhibits superior performance on the OOD dataset, it incurs a significant performance deterioration on the ID dataset. This demonstrates that LwPTV's effectiveness of proposed salience score in pruning parameters. See more details in Appendix B.9.

## 5.2 Additional analysis

**Trade-off between ID and OOD.** To explore the impact of pruning ratio $\eta$ on the trade-off between ID and OOD task performance, we conducted the experiments using Task Arithmetic w/ LwPTV on the ViT-B/32 and ViT-L/14. Notably, $\eta$ serves as a parameter to balance the pretrained model and task vectors of the fine-tuned models. $\eta = 1$ signifies a predominance of the pretrained model, while $\eta = 0$ represents the original merging model. We can see that the OOD performance gradually increases as the pruning ratio increases. Conversely, the ID task performance shows an initial improvement followed by a decline. It is reasonable since a proper pruning ratio can improve both ID and OOD performance by removing redundant parameters. And a too large pruning ratio will prune the useful parameters for ID tasks and hurt its performance, indicating the trade-off between ID and OOD tasks. Besides, a pruning ratio $\eta$ between 0.6 and 0.8 offers a relatively desired trade-off, ensuring improved OOD performance while maintaining desired ID accuracy, where we set $\eta = 0.7$ for all experiments.

**Salience score and mask vectors.** To facilitate an intuitive analysis of salience score, masks associated with each task vector and their interrelationships, we visualize them for ViT-B/32 in Fig.6. We can see that, for different tasks, the salience score vectors usually have low values in certain same layers, indicating their redundancy in these layers. Besides, in each salience score vector, most of the elements have low values, which explains why the model produces better OOD and ID performance even when we throw away most layers (pruning ratio $\eta = 0.7$). Therefore, the mask vectors of different tasks have a lot of overlapping layers, which can be viewed as those parameters unimportant to all tasks. Those non-overlapping mask information might be viewed as the beneficial parameters for the corresponding ID task, which is the reason that we choose to keep them. Therefore, we adopt the OR operation to extract a shared mask vector from all masks, i.e., replacing these overlapping layers with the corresponding layers of pre-trained model. To provide a qualitative visualization, we present t-SNE results of Task Arithmetic and Task Arithmetic+LwPTV on OOD tasks in Fig. 7. The visualization provides an intuitive view of the feature distributions, where LwPTV shows clearer cluster separation. We further quantify the representation drift from the pretrained ViT-B/32

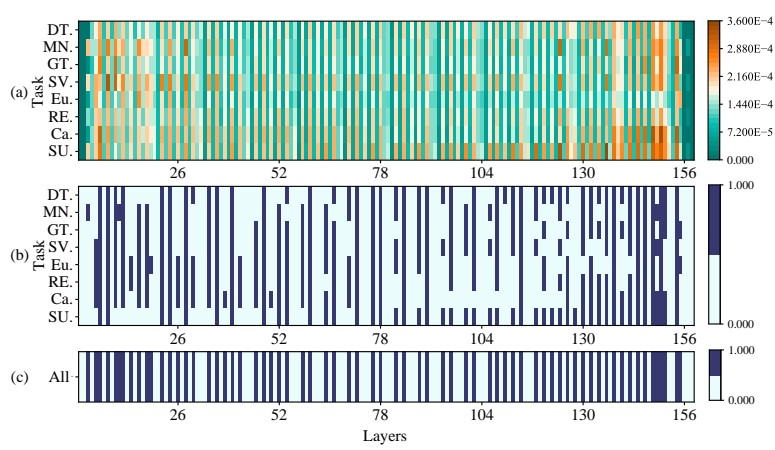

Figure 6: Visualization of (a) salience score matrix, (b) mask vector for each task, and (c) final mask vector, all obtained on ViT-B/32, where x-axis denotes the layer index, y-axis in (a-b) denotes task name.

Figure 7: T-SNE visualizations of (a) Task Arithmetic, and (b) Task Arithmetic w/ LwPTV on CIFAR10.

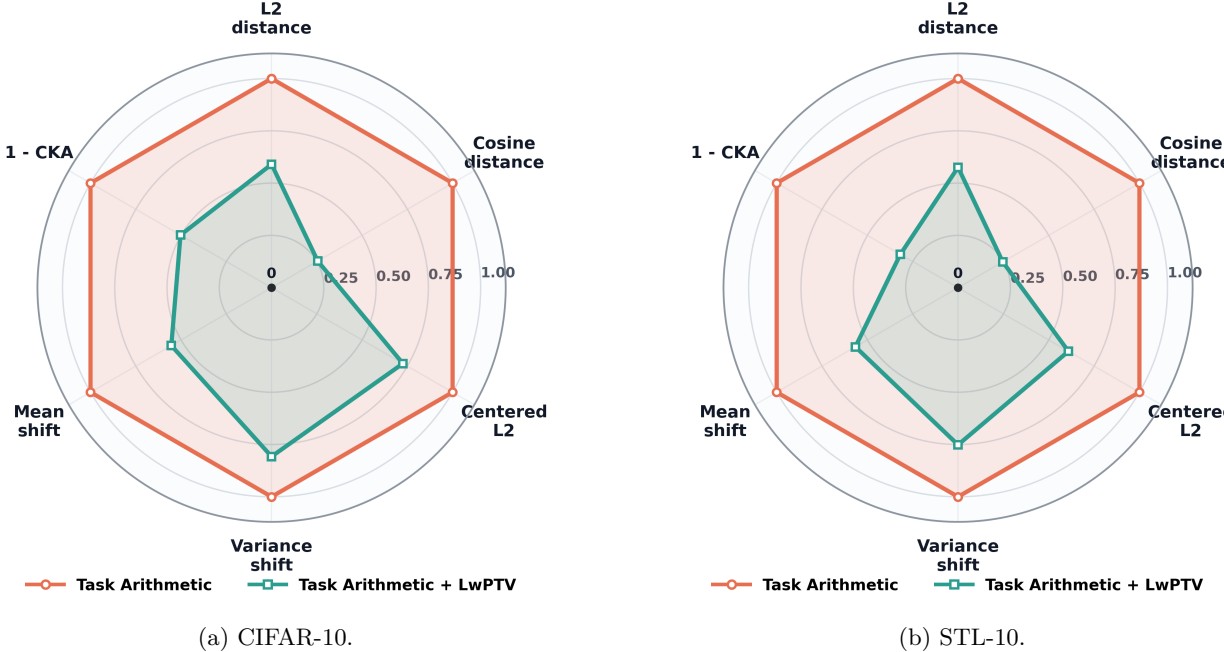

(a) CIFAR-10.

(b) STL-10.

Figure 8: Representation distance from the pretrained ViT-B/32 model on CIFAR-10 and STL-10.

model to measure how well different merging methods preserve pretrained representations. As shown in Fig. 8, Task Arithmetic+LwPTV consistently exhibits smaller distances to the pretrained representation than Task Arithmetic on CIFAR-10 and STL-10 under multiple metrics. This indicates that LwPTV better preserves pretrained representations while reducing feature distortion caused by task-vector updates. Additional results, including NLP experiments, storage overhead analysis, and further ablations, are provided in Appendix B and Appendix C.

## 5.3 Discussion with Multi-domain Adaptation Methods

Recent studies have explored multi-domain adaptation by reducing conflicts during training. For example, Liang et al. (2025) improves multi-domain fine-tuning through evolving interactions among samples, while Liang et al. (2026) modifies the optimization process to encourage beneficial cross-domain gradient and

curvature interactions. Different from these training-time adaptation methods, LwPTV operates after task-specific checkpoints are obtained and performs post-hoc, training-free model merging by pruning low-diversity components in task vectors.

These methods address different issues at different stages of the learning pipeline. Liang et al. (2025) focuses on sample-level interactions during fine-tuning, and Liang et al. (2026) considers gradient- and curvature-level interactions during optimization. In contrast, LwPTV focuses on the merging stage, where low-diversity task-vector components are identified as potentially redundant updates and pruned to improve OOD generalization.

These methods can also be complementary. Training-time multi-domain adaptation methods can potentially reduce interference during checkpoint construction and produce better task-specific models, which may provide a better starting point for model merging. LwPTV can then further improve the merged model by pruning potentially redundant task-vector components and enhancing OOD generalization.

### 5.4 Broader Impact and Limitations

Although LwPTV improves OOD robustness in our experiments, it may introduce an ID performance trade-off in certain merging settings. For example, when combined with LW AdaMerging, pruning and scaling may reduce the contribution of task-specific updates, resulting in lower ID accuracy. Therefore, practitioners should evaluate the ID performance of LwPTV before deployment. The pruning factor $\eta$ should be selected based on ID performance and the requirements of the target application. This consideration is particularly important in safety-critical scenarios, such as autonomous driving and medical imaging, where even small performance degradations can have meaningful consequences.

In addition, the OOD results should be interpreted within the scope of the current evaluation setting. Since CLIP's instance-level pre-training data are not publicly available, we cannot verify whether the evaluated datasets overlap with the pre-training corpus. Therefore, our results should be understood as demonstrating improved robustness under relative OOD evaluation with respect to the downstream fine-tuning tasks, rather than as evidence of absolute distributional generalization to entirely unseen distributions.

## 6 Conclusion

In this work, we first systematically analyze existing model merging techniques and identify their limitations in generalizing to OOD data. To address this issue, we propose a novel plug-and-play framework, termed LwPTV, and introduce a training-free salience score to measure the redundancy of task vectors. By pruning the layer-wise parameters with low salience score, LwPTV enhances the OOD generalization capability of merged models while preserving ID performance. Extensive experimental results demonstrate that our approach significantly improves the OOD generalization performance of existing model merging algorithms.

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

# A  Theoretical analyses

## A.1  Proof of proposition 1

We start by introducing the theoretical setting.

**Theoretical formulation.** Consider a binary classification problem, where the target is to predict $y \in \{+1, -1\}$ given the input $\boldsymbol{X} = (\boldsymbol{x}_1, \boldsymbol{x}_2, \cdots, \boldsymbol{x}_P) \in \mathbb{R}^{d \times P}$ that contains $P$ $d$-dimensional tokens. Following the state-of-the-art theoretical works (Li et al., 2023; Makkuva et al., 2024; Li et al., 2024; Jiang et al., 2024) on the generalization and learning dynamics of neural models, we use a one-layer single-head Transformer as the learner model for theoretical analysis, which can be characterized as $f(\boldsymbol{X}; \boldsymbol{\theta}) = 1/P \sum_{l=1}^{P} \boldsymbol{a}_l^\top \mathrm{Relu}(\boldsymbol{V} \sum_{s=1}^{P} \boldsymbol{x}_s \mathrm{softmax}_l(\boldsymbol{x}_s^\top \boldsymbol{W} \boldsymbol{x}_l))$, where $\boldsymbol{\theta} = \{\{\boldsymbol{a}_l\}_{l=1}^{P}, \boldsymbol{V}, \boldsymbol{W}\}$ with $\boldsymbol{a}_l \in \mathbb{R}^m$, $\boldsymbol{V} \in \mathbb{R}^{m \times d}$, and $\boldsymbol{W} \in \mathbb{R}^{d \times d}$. $\mathrm{softmax}_l(\boldsymbol{x}_s^\top \boldsymbol{W} \boldsymbol{x}_l) = e^{\boldsymbol{x}_s^\top \boldsymbol{W} \boldsymbol{x}_l} / \sum_{j=1}^{P} e^{\boldsymbol{x}_j^\top \boldsymbol{W} \boldsymbol{x}_l}$. $\mathrm{Relu}(\boldsymbol{z}) = \max\{0, \boldsymbol{z}\}$.

Denote $\mathcal{T}_k$, $k \leq K$ as the $i$-th task function that we aim to learn with the training dataset $\{\boldsymbol{X}^n, y^n\}_{n=1}^{N}$. Starting from the initialization $\boldsymbol{\theta}_{pre}$, we train the model using stochastic gradient descent with Hinge loss to obtain the fine-tuned model $\boldsymbol{\theta}_i$. The task vector is then computed as $\boldsymbol{\tau}_k = \boldsymbol{\theta}_k - \boldsymbol{\theta}_{pre}$.

The data formulation follows Definition 2 of Li et al. (2025), where the label of each data for task $\mathcal{T}_k$ is determined by the majority between tokens with two discriminative features $\boldsymbol{\mu}_{\mathcal{T}_k}$ and $-\boldsymbol{\mu}_{\mathcal{T}_k}$. If the number of tokens equal to $\boldsymbol{\mu}_{\mathcal{T}_k}$ (or $-\boldsymbol{\mu}_{\mathcal{T}_k}$) is large than that of $-\boldsymbol{\mu}_{\mathcal{T}_k}$ (or $\boldsymbol{\mu}_{\mathcal{T}_k}$), then $y = +1$ (or $y = -1$). Each data can also contain task-irrelevant tokens from an orthonormal set $\{\boldsymbol{v}_1, \boldsymbol{v}_2, \cdots, \boldsymbol{v}_M\}$ that are orthogonal to discriminative features. Given $\boldsymbol{\tau}_k$, $k \in [K]$, trained with a batch size of $B \geq \Omega(\epsilon^{-2} \log M)$ to achieve an $\epsilon$ generalization error on $\mathcal{T}_k$, denote $\boldsymbol{u}_k^i$ as the $i$-th row of the part of $\boldsymbol{V}$ in $\boldsymbol{\tau}_k$. Then, we can prove Proposition 1 as follows.

*Proof.* By Corollary 2 and Lemma 5 of Li et al. (2025), we know that with a constant probability, which we denote by $p$, $\boldsymbol{u}_k^i$, $k \in [K]$, $i \in [m]$, learns the discriminative feature. Then, with a high probability of $1 - p^K$, there exists $k_1 \neq k_2 \in [K]$, such that the $i$-th neuron of $\boldsymbol{\tau}_{k_1}$ learns discriminative features, while the $i$-th neuron of $\boldsymbol{\tau}_{k_2}$ fails to learn discriminative features.

By their Definition 4 of Li et al. (2025), we have that for any $i\mathcal{S}$,

$$\|\boldsymbol{u}_k^i\| \geq \Omega(m^{-1/2}). \tag{9}$$

By Lemma 5 in Li et al. (2025), we know that neurons of $\boldsymbol{\tau}_{k_1}$ in $\mathcal{S}$ are mainly in the direction of $\boldsymbol{\mu}_{\mathcal{T}_k}$ or $-\boldsymbol{\mu}_{\mathcal{T}_k}$, with the norm of all other directions smaller than a $1/\sqrt{M}$ scaling of that in the direction of the discriminative pattern. Then, the variance of lucky neurons in different task vectors can be computed as

$$\mathrm{DV}(g(\boldsymbol{\tau}_{k_1}, \mathcal{S}); \{\boldsymbol{\tau}_j\}_{j=1}^{K}) = \mathbb{E}\left[\left\|g(\boldsymbol{\tau}_{k_1}, \mathcal{S}) - \frac{1}{K} \sum_{j=1}^{K} g(\boldsymbol{\tau}_j, \mathcal{S})\right\|\right]$$
$$\geq |\mathcal{S}| \cdot \Omega(m^{-1}). \tag{10}$$

Hence, the diversity of neurons that learn discriminative patterns is in the order of $|\mathcal{S}| \cdot \Omega(m^{-1/2})$.

For neurons that learn no discriminative features, $k \in [K]$, $i \in [m]$, we have

$$\mathrm{DV}(g(\boldsymbol{\tau}_{k_2}, \mathcal{S}); \{\boldsymbol{\tau}_j\}_{j=1}^{K}) = \mathbb{E}\left[\left\|g(\boldsymbol{\tau}_{k_2}, \mathcal{S}) - \frac{1}{K} \sum_{j=1}^{K} g(\boldsymbol{\tau}_j, \mathcal{S})\right\|\right]$$
$$\leq |\mathcal{S}| \cdot O(m^{-1}) \cdot \frac{1}{\sqrt{K}}. \tag{11}$$

The last step comes from that (i) $\|g(\boldsymbol{\tau}_{k_2}, \mathcal{S})\| \leq O(m^{-1/2})\sqrt{\log B / B}$. (ii)

$$\left\|\frac{1}{K} \sum_{j=1}^{K} g(\boldsymbol{\tau}_j, \mathcal{S})\right\| \leq O(m^{-1/2}) \cdot \frac{1}{\sqrt{K}}. \tag{12}$$

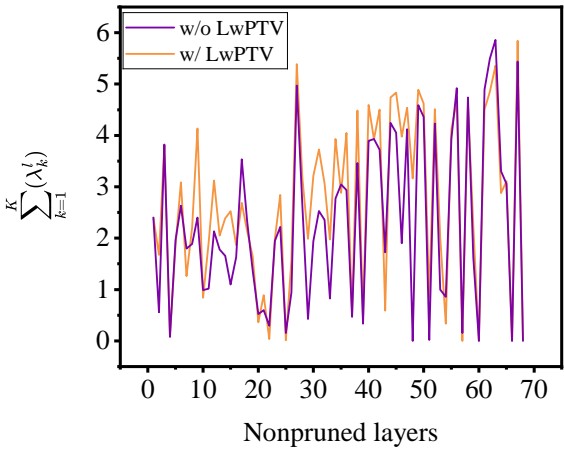

(a) Lw AdaMerging w/ LwPTV *vs.* Lw AdaMerging on ViT-B/32.

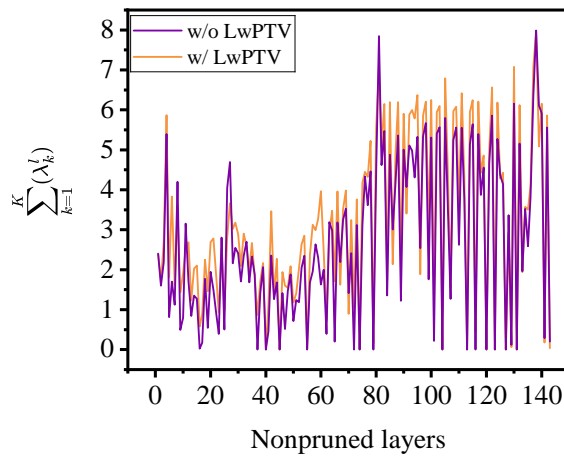

(b) Lw AdaMerging w/ LwPTV *vs.* Lw AdaMerging on ViT-L/14.

Figure 9: The variations in $\lambda_k^l$ of unpruned layers subsequent to the application of the LW AdaMerging and LW AdaMerging w/ LwPTV.

Then, by combining (10) and (12), we have

$$\text{DV}(g(\boldsymbol{\tau}_{k_1}, \mathcal{S}); \{\boldsymbol{\tau}_j\}_{j=1}^K) \geq \sqrt{K} \cdot \Omega(\text{DV}(g(\boldsymbol{\tau}_{k_2}, \mathcal{S}); \{\boldsymbol{\tau}_j\}_{j=1}^K)). \tag{13}$$

$\square$

# B Experimental details

## B.1 The analysis of scaling

To investigate the impact of LwPTV (w/o scale) on the layer-wise coefficients $\lambda_k^l$ of LW AdaMerging, we have plotted a comparative figure of the unpruned layer coefficients under two scenarios: LW AdaMerging and LW AdaMerging w/ LwPTV. The results are illustrated in Fig.9a and Fig.9b. In these figures, $\sum_{k=1}^K(\lambda_k^l)$ represents the sum of the coefficients of $K$ tasks on the $l$-th layer. It can be clearly observed from these two figures that the $\lambda_k^l$ obtained after pruning and optimization are significantly

Table 4: Performance comparison under different values of $\eta$ on ViT-B/32.

| Method | $\eta$ | ID Avg | OOD Avg | H-score |
|---|---|---|---|---|
| LW AdaMerging | – | **80.1** | 57.7 | 67.1 |
| LW AdaMerging w/ LwPTV | 0.5 | 73.9 | **62.2** | 67.6 (+0.5) |
| | 0.6 | 76.3 | 62.0 | 68.4 (+1.3) |
| | 0.7 | 77.3 | 61.6 | **68.5** (+1.4) |
| | 0.8 | 77.8 | 60.9 | 68.3 (+1.2) |
| | 0.9 | 69.8 | 60.7 | 64.9 (-2.2) |

increased. This is because, during the process of optimizing $\lambda_k^l$ by entropy minimization, the adjustable coefficients $\lambda_k^l$ are amplified to enhance the influence of unpruned parameters in order to minimize the loss function as much as possible. This allows the merged model to maintain its ID performance even as the pruning ratio continuously increases, but its OOD performance does not show a significant improvement. In order to improve the OOD performance, we further use $\hat{\lambda}_k^l = \eta \cdot \lambda_k^l$ to scale the coefficient after optimizing the equation 8. We next provide both a gradient-based analysis and empirical evidence to demonstrate the effectiveness and robustness of the proposed scaling strategy.

**(a) Why the mask disrupts AdaMerging's learned coefficients.** For non-adaptive merging baselines such as Task Arithmetic and Ties, adding LwPTV does not introduce a new coefficient-learning stage. We keep the original merging rule unchanged and apply LwPTV as a post-hoc pruning module on the task vectors. This allows us to isolate the effect of task-vector pruning itself.

AdaMerging is different. Its merging coefficients are learned as part of the method by minimizing an entropy objective on unlabeled target-task samples. Therefore, these coefficients depend on the task-vector parameterization used during optimization.

After applying LwPTV, the parameterization changes because some layer-wise task-vector components are masked out. As a result, the coefficient-learning problem is no longer the same as in the original unmasked task-vector space. Therefore, for AdaMerging, we first apply the LwPTV mask and then optimize the coefficients in the masked task-vector space. This is not an extra tuning advantage; it simply ensures that coefficient learning and final inference are based on the same parameterization.

Table 5: Performance comparison with different numbers of ID tasks.

| $K$ | Method | ID | OOD | H-score |
|---|---|---|---|---|
| 4 | LW AdaMerging | **83.4** | 58.3 | 68.6 |
| | LW AdaMerging w/ LwPTV | 80.0 | **61.8** | **69.7** (+1.1) |
| 5 | LW AdaMerging | **83.3** | 58.0 | 68.4 |
| | LW AdaMerging w/ LwPTV | 79.7 | **61.7** | **69.6** (+1.2) |
| 6 | LW AdaMerging | **82.9** | 57.5 | 67.9 |
| | LW AdaMerging w/ LwPTV | 78.9 | **61.3** | **69.0** (+1.1) |
| 7 | LW AdaMerging | **84.1** | 57.5 | 68.3 |
| | LW AdaMerging w/ LwPTV | 80.5 | **61.5** | **69.7** (+1.4) |
| 8 | LW AdaMerging | **80.1** | 57.7 | 67.1 |
| | LW AdaMerging w/ LwPTV | 77.3 | **61.7** | **68.6** (+1.5) |

This behavior is particularly important for LW AdaMerging, where each task and each layer has an independent coefficient. Once some layers are removed, entropy minimization can only adjust the coefficients of the retained layers. The remaining coefficients may then increase to compensate for the missing update directions. This can amplify the retained task-specific updates and partially weaken the OOD-preserving effect of LwPTV, which motivates the scaling fix used for LW AdaMerging.

Formally, for LW AdaMerging with LwPTV, the merged model is

$$\hat{\theta}_{\mathrm{m}} = \theta_{\mathrm{pre}} + \hat{\mathbf{m}} \odot \left\{ \sum_{k=1}^{K} \lambda_k^l \tau_k^l \right\}_{l=1}^{L}. \tag{14}$$

Let the entropy objective be

$$\mathcal{L}_{ent} = \sum_{k=1}^{K} \sum_{x_i \in B_k} H(f_{\hat{\theta}_{\mathrm{m}}}(x_i)), \tag{15}$$

where $B_k$ represents the unlabeled batch corresponding to the $k$-th task. The gradient with respect to a layer-wise coefficient is

$$\frac{\partial \mathcal{L}_{ent}}{\partial \lambda_k^l} = \hat{m}^l \left\langle \nabla_{\theta^l} \mathcal{L}_{ent}, \tau_k^l \right\rangle. \tag{16}$$

When $\hat{m}^l = 0$, this gradient becomes zero, meaning that the masked layer cannot help reduce the entropy objective. The optimization is therefore restricted to the retained layers. Since LW AdaMerging uses layer-wise coefficients, the remaining layers may absorb part of the contribution of the removed layers, leading to coefficient amplification.

This is why we introduce the scaling fix $\hat{\lambda}_k^l = \eta \lambda_k^l$ for LW AdaMerging. This scaling mitigates excessive compensation among the unmasked layers.

**(b) Robustness of the scaling factor.** We added two analyses to examine the robustness of this scaling strategy. First, we sweep $\eta$ on ViT-B/32. As shown in Table 4, LW AdaMerging obtains an H-score of 67.1. After adding LwPTV, $\eta = 0.5, 0.6, 0.7, 0.8$ all improve the H-score, with gains of +0.5, +1.3, +1.4, and +1.2, respectively. The best result is obtained at $\eta = 0.7$, which we use as the default. These results show that the improvement is not caused by a single isolated choice of $\eta$. We also observe that an overly large value, such as $\eta = 0.9$, weakens the effect and reduces the H-score, suggesting that the scaling should be moderate rather than close to 1.

Second, we evaluate different numbers of ID tasks $K$. As shown in Table 5, LwPTV consistently improves the H-score over LW AdaMerging for $K = 4, 5, 6, 7, 8$, with gains of +1.1, +1.2, +1.1, +1.4, and +1.5, respectively. The OOD performance is also consistently improved in all settings. These results suggest that the scaling strategy is stable across different numbers of tasks $K$.

---

**Algorithm 1** Overall algorithm when combining our LwPTV with existing model merging methods.

---

1: **Input:** pre-trained model parameters $\boldsymbol{\theta}_{pre}$, $K$ task-specific fine-tuned model parameters $\{\boldsymbol{\theta}_k\}_{k=1}^{K}$ and hyperparameter $\eta$;

2: **for** $k = 1$ **to** $K$ **do**

3: $\quad$ Compute the task vector $\boldsymbol{\tau}_k$ :

4: $\quad\quad$ $\boldsymbol{\tau}_k = \boldsymbol{\theta}_k - \boldsymbol{\theta}_{pre}$;

5: $\quad$ Compute the salience score $\mathbf{s}_k$ of $\boldsymbol{\tau}_k$ by equation 5;

6: $\quad$ Compute the mask $\mathbf{m}_k$ by equation 6;

7: **end for**

8: Calculate the final mask $\hat{\mathbf{m}}$ by equation 7.

9: # Model merging methods execute:

10: **If Task Arithmetic**: $\hat{\boldsymbol{\theta}}_m = \boldsymbol{\theta}_{pre} + \hat{\mathbf{m}} \odot \lambda \sum_{k=1}^{K} \boldsymbol{\tau}_k$.

11: **If Ties**: Perform trim and sign election operations on $\{\hat{\mathbf{m}} \odot \boldsymbol{\tau}_k\}_{k=1}^{K}$ to obtain $\{\hat{\boldsymbol{\tau}}_k\}_{k=1}^{K}$, then perform $\hat{\boldsymbol{\theta}}_m = \boldsymbol{\theta}_{pre} + \lambda \sum_{k=1}^{K} \hat{\boldsymbol{\tau}}_k$.

12: **If TW AdaMerging**: Get $\{\lambda_k\}_{k=1}^{K}$ by minimizing entropy based on equation 8 , then perform equation 8.

13: **If TW AdaMerging++**: Perform trim and sign election operations on $\{\hat{\mathbf{m}} \odot \boldsymbol{\tau}_k\}_{k=1}^{K}$ to obtain $\{\hat{\boldsymbol{\tau}}_k\}_{k=1}^{K}$, then get $\{\lambda_k\}_{k=1}^{K}$ by minimizing entropy based on $\boldsymbol{\theta}_{pre} + \sum_{k=1}^{K} \lambda_k \hat{\boldsymbol{\tau}}_k$ , then perform $\hat{\boldsymbol{\theta}}_m = \boldsymbol{\theta}_{pre} + \sum_{k=1}^{K} \lambda_k \hat{\boldsymbol{\tau}}_k$.

14: **If LW AdaMerging**: Get $\{\{\lambda_k^l\}_{l=1}^{L}\}_{k=1}^{K}$ by minimizing entropy based on equation 8, then perform $\hat{\boldsymbol{\theta}}_m = \boldsymbol{\theta}_{pre} + \hat{\mathbf{m}} \odot \{\sum_{k=1}^{K} \eta \lambda_k^l \boldsymbol{\tau}_k\}_{l=1}^{L}$.

15: **If LW AdaMerging++**: Perform trim and sign election operations on $\{\hat{\mathbf{m}} \odot \boldsymbol{\tau}_k\}_{k=1}^{K}$ to obtain $\{\hat{\boldsymbol{\tau}}_k\}_{k=1}^{K}$, and then get $\{\{\lambda_k^l\}_{l=1}^{L}\}_{k=1}^{K}$ by minimizing entropy based on $\boldsymbol{\theta}_{pre} + \{\sum_{k=1}^{K} \lambda_k^l \hat{\boldsymbol{\tau}}_k\}_{l=1}^{L}$, then perform $\hat{\boldsymbol{\theta}}_m = \boldsymbol{\theta}_{pre} + \{\sum_{k=1}^{K} \eta \lambda_k^l \hat{\boldsymbol{\tau}}_k\}_{l=1}^{L}$.

16: **If PCB**: Based on $\{\hat{\mathbf{m}} \odot \boldsymbol{\tau}_k\}_{k=1}^{K}$ to get importance score $\{\hat{\boldsymbol{\beta}}_k\}_{k=1}^{K}$, then perform $\hat{\boldsymbol{\theta}}_m = \boldsymbol{\theta}_{pre} + \hat{\mathbf{m}} \odot \sum_{k=1}^{K} (\hat{\boldsymbol{\beta}}_k \odot \lambda_k \boldsymbol{\tau}_k) / \sum_{k=1}^{K} \hat{\boldsymbol{\beta}}_i$.

17: **If FR-Merging**:Apply a Fourier filter to remove low-frequency components from $\{\hat{\mathbf{m}} \odot \boldsymbol{\tau}_k\}_{k=1}^{K}$, yielding $\{\tilde{\boldsymbol{\tau}}_k\}_{k=1}^{K}$. Then compute merging coefficients $\{\lambda_k \mathbb{E}(\tilde{\boldsymbol{\tau}}_k) \left(\sum_{k=1}^{K} \mathbb{E}(\tilde{\boldsymbol{\tau}}_k)\right)^{-1}\}_{k=1}^{K}$. Finally, perform $\hat{\boldsymbol{\theta}}_m = \boldsymbol{\theta}_{pre} + \sum_{k=1}^{K} \lambda_k \mathbb{E}(\tilde{\boldsymbol{\tau}}_k) \left(\sum_{k=1}^{K} \mathbb{E}(\tilde{\boldsymbol{\tau}}_k)\right)^{-1} \tilde{\boldsymbol{\tau}}_k$.

---

## B.2 Algorithm

For Task Arithmetic, Ties, TW AdaMerging, TW AdaMerging++, LW AdaMerging, LW AdaMerging++, PCB-MERGING and FR-Merging, we present the algorithmic workflows of their integration with LwPTV in Algorithm 1.

## B.3 Dataset details

Following Yadav et al. (Yadav et al., 2023; Zhou et al., 2022), we utilize eight datasets, including Stanford Cars (Krause et al., 2013), DTD (Cimpoi et al., 2014), EuroSAT (Helber et al., 2019), GTSRB (Stallkamp et al., 2011), MNIST (Deng, 2012), RESISC45 (Cheng et al., 2017), SUN397 (Xiao et al., 2016), and SVHN (Netzer et al., 2011), for merging vision models. These datasets are considered ID datasets. Following Wang et al. (Wang et al., 2024b), we select CIFAR100 (Krizhevsky et al., 2009), STL10 (Coates et al., 2011), Flowers102 (Nilsback & Zisserman, 2008), OxfordIIITPet (Parkhi et al., 2012), PCAM (S.Veeling et al., 2018), FER2013 (Goodfellow et al., 2013), EMNIST (Cohen et al., 2017), CIFAR10 (Krizhevsky et al., 2009), Food101 (Bossard et al., 2014), FashionMNIST (Xiao et al., 2017), RenderedSST2 (Socher et al., 2013; Radford et al., 2019), KMNIST (Clanuwat et al., 2018) and ImageNet-1k (Deng et al., 2009) as OOD datasets. The details of these datasets are shown in Tab.6

Table 6: In Domain and Out of Domain datasets details

| Feature(→) Dataset (↓) | Type | Sample Size | Test Set Sample Size | Number of Classes | Samples per Class | Class Balance |
|---|---|---|---|---|---|---|
| Stanford Cars | | 16185 | 8041 | 196 | 1:1 split | Yes |
| DTD | | 5640 | 1880 | 47 | 40 | Yes |
| EuroSAT | | 27000 | 2700 | 10 | 2000-3000 | Yes |
| GTSRB | ID | 51839 | 12630 | 43 | / | No |
| MNIST | | 70000 | 10000 | 10 | 1000 | Yes |
| RESISC45 | | 31500 | 6300 | 45 | approximately 700 | Yes |
| SUN397 | | 108754 | 7940 | 397 | at least 100 images per category | No |
| SVHN | | 60000 | 26032 | 10 | / | No |
| CIFAR100 | | 60000 | 10000 | 100 | 100 | Yes |
| STL10 | | 13000 | 8000 | 10 | 800 | Yes |
| Flowers102 | | 8189 | 6129 | 102 | per class 40-250 | No |
| OxfordIIITPet | | 7349 | 3669 | 37 | approximately 100 | Yes |
| PCAM | | 327680 | 40960 | 2 | / | / |
| FER2013 | OOD | 35887 | 7178 | 7 | / | No |
| EMNIST | | 280000 | 40000 | 10 | 4000 | Yes |
| CIFAR10 | | 60000 | 10000 | 10 | 1000 | Yes |
| Food101 | | 101000 | 25250 | 101 | 250 | Yes |
| FashionMNIST | | 70000 | 10000 | 10 | 1000 | Yes |
| RenderedSST2 | | 9613 | 1821 | 2 | / | / |
| KMNIST | | 70000 | 10000 | 10 | 1000 | Yes |
| ImageNet1000 | | 3200000 | 50000 | 1000 | / | No |

## B.4 Evaluation metrics

To rigorously assess the efficacy of our method in enhancing the generalization capability of model merging methods on OOD datasets, while concurrently preserving their efficacy on ID datasets, we utilize a key evaluation metrics: H-score. The H-score is defined by the harmonic mean of the mean performance for the ID datasets, denoted as $Avg(P_{ID})$, coupled with the mean performance for the OOD datasets, denoted as $Avg(P_{OOD})$. It can be formulated as follows:

$$P_H = \frac{2 \times Avg(P_{ID}) \times Avg(P_{OOD})}{Avg(P_{ID}) + Avg(P_{OOD})}. \tag{17}$$

## B.5 ID performance

We evaluate the performance of our proposed method alongside baseline approaches on ID datasets across three distinct architectures, namely ViT-B/32, ViT-L/14, and ViT-H/14. The corresponding results are systematically summarized in Tab.7 and Tab.8, respectively. The experimental results demonstrate that our method, when integrated with existing model merging methods, induces minimal performance degradation on ID datasets, and in some cases even enhances ID performance. This improvement can be attributed to the reduced parameter conflicts achieved through our task vector pruning strategy, which effectively mitigates interference between different task-specific parameters.

## B.6 Performance comparison on ViT-H/14

Tab.9 demonstrates the comparative evaluation of our method against multiple baselines across ID and OOD datasets, implemented on the ViT-H/14 architecture. It can be seen that although the ViT-H/14 model is already very large, there is still a gap in the generalization performance between the pretrained model and the individual fine-tuned model on the OOD datasets. Therefore, our method still works. Experimental results reveal that our method yields significant performance gains over existing techniques, with H-score improvements of 1.4%, 0.9%, 1.3%, and 0.5% over Task Arithmetic, Ties, PCB, and FR-Merging respectively, when combined with these approaches.

## B.7 More ablation study details

Tab.10 and Tab.11 provides a detailed summary of the experimental results from our comprehensive ablation studies performed on the ID dataset and OOD dataset.

Table 7: Performance comparison for ID datasets on ViT-B/32 and ViT-L/14.

| | Method (↓) | SUN397 | Cars | RESISC45 | EuroSAT | SVHN | GTSRB | MNIST | DTD | Avg. |
|---|---|---|---|---|---|---|---|---|---|---|
| | Pretrained | 62.3 | 59.7 | 60.7 | 45.5 | 31.4 | 32.6 | 48.5 | 43.8 | 48.0 |
| | Individual | 75.3 | 77.7 | 96.1 | 99.7 | 97.5 | 98.7 | 99.7 | 79.4 | 90.5 |
| | Weight Averaging (Wortsman et al., 2022a) | 65.3 | 63.4 | 71.4 | 71.7 | 64.2 | 52.8 | 87.5 | 50.1 | 65.8 |
| | Fisher Merging (Matena & Raffel, 2022) | 68.6 | 69.2 | 70.7 | 66.4 | 72.9 | 51.1 | 87.9 | 59.9 | 68.3 |
| | RegMean (Jin et al., 2023) | 65.3 | 63.5 | 75.6 | 78.6 | 78.1 | 67.4 | 93.7 | 52.0 | 71.8 |
| | AdaMerging w/ Surgery (Yang et al., 2024a) | 69.8 | 71.0 | 88.9 | 98.1 | 91.7 | 96.5 | 98.8 | 73.6 | 86.1 |
| ViT-B/32 | Task Arithmetic (Ilharco et al., 2023) | 55.2 | 54.9 | 66.7 | 78.9 | **80.2** | **69.7** | **97.3** | 50.4 | 69.1 |
| | **w/ LwPTV** (Ours) | **66.8** | **66.0** | **77.2** | **79.1** | 78.3 | 68.7 | 92.3 | **54.4** | **72.8**$^{(+3.7)}$ |
| | Ties (Yadav et al., 2023) | 65.0 | 64.4 | 74.8 | 77.4 | **81.2** | **69.3** | **96.5** | 54.5 | **72.9** |
| | **w/ LwPTV** (Ours) | **67.6** | **66.7** | **77.4** | 77.4 | 77.7 | 66.8 | 92.4 | **55.3** | 72.7$^{(-0.2)}$ |
| | TW AdaMerging (Yang et al., 2024b) | 58.0 | 53.2 | 68.8 | 85.7 | 81.1 | **84.4** | 92.4 | 44.8 | 71.1 |
| | **w/ LwPTV** (Ours) | **61.6** | **59.7** | **77.0** | **87.3** | **87.0** | 84.9 | **95.5** | **50.6** | **75.4**$^{(+4.3)}$ |
| | TW AdaMerging++ (Yang et al., 2024b) | 60.8 | 56.9 | 73.1 | 83.4 | 87.3 | **82.4** | 95.7 | 50.1 | 73.7 |
| | **w/ LwPTV** (Ours) | **63.0** | **63.2** | **78.5** | **84.9** | **88.4** | 79.1 | **96.7** | **55.1** | **76.1**$^{(+2.4)}$ |
| | LW AdaMerging (Yang et al., 2024b) | 64.5 | 68.1 | 79.2 | **93.8** | **87.0** | **91.9** | **97.5** | **59.1** | **80.1** |
| | **w/ LwPTV** (Ours) | **68.0** | **68.5** | **80.7** | 88.3 | 80.0 | 81.2 | 94.1 | 57.7 | 77.3$^{(-2.8)}$ |
| | LW AdaMerging++ (Yang et al., 2024b) | 66.6 | 68.3 | **82.2** | **94.2** | **89.6** | **89.0** | **98.3** | **60.6** | **81.1** |
| | **w/ LwPTV** (Ours) | **68.9** | **69.0** | 81.0 | 85.8 | 82.2 | 76.7 | 95.0 | 58.2 | 77.1$^{(-4.0)}$ |
| | PCB-MERGING (Du et al., 2024) | 63.8 | 62.0 | 77.1 | 80.6 | **87.5** | **78.5** | 98.7 | 58.4 | 75.8 |
| | **w/ LwPTV** (Ours) | **68.6** | **67.7** | **80.7** | **83.0** | 85.4 | 77.2 | 96.2 | **60.0** | **77.3**$^{(+1.5)}$ |
| | FR-Merging (Zheng & Wang, 2025) | 66.2 | 64.5 | 77.2 | **90.1** | **85.4** | **82.3** | **98.5** | **60.0** | **78.1** |
| | **w/ LwPTV** (Ours) | **67.2** | **67.8** | **77.9** | 81.9 | 75.5 | 74.9 | 93.8 | 58.4 | 74.7$^{(-3.4)}$ |
| | Pretrained | 66.8 | 77.7 | 71.0 | 59.9 | 58.4 | 50.5 | 76.3 | 55.3 | 64.5 |
| | Individual | 82.3 | 92.4 | 97.4 | 100 | 98.1 | 99.2 | 99.7 | 84.1 | 94.2 |
| | Weight Averaging (Wortsman et al., 2022a) | 72.1 | 81.6 | 82.6 | 91.9 | 78.2 | 70.7 | 97.1 | 62.8 | 79.6 |
| | Fisher Merging (Matena & Raffel, 2022) | 69.2 | 88.6 | 87.5 | 93.5 | 80.6 | 74.8 | 93.3 | 70.0 | 82.2 |
| | RegMean (Jin et al., 2023) | 73.3 | 81.8 | 86.1 | 97.0 | 88.0 | 84.2 | 98.5 | 60.8 | 83.7 |
| | AdaMerging w/ Surgery (Yang et al., 2024a) | 80.3 | 90.8 | 94.3 | 98.2 | 94.1 | 98.7 | 99.2 | 82.5 | 92.3 |
| ViT-L/14 | Task Arithmetic (Ilharco et al., 2023) | 73.9 | 82.1 | 86.6 | 94.1 | **87.9** | **86.7** | **98.9** | 65.6 | 84.5 |
| | **w/ LwPTV** (Ours) | **74.7** | **86.2** | **89.3** | **94.6** | 86.7 | 86.1 | 98.8 | **68.6** | **85.6**$^{(+1.1)}$ |
| | Ties (Yadav et al., 2023) | **76.5** | 85.0 | 89.3 | **95.7** | **90.3** | 83.3 | 99.0 | 68.8 | 86.0 |
| | **w/ LwPTV** (Ours) | 76.3 | **87.4** | **90.9** | 95.1 | 89.1 | **87.3** | 99.0 | **70.5** | **86.9**$^{(+0.9)}$ |
| | TW AdaMerging (Yang et al., 2024b) | 75.6 | 83.5 | 79.7 | 90.3 | 83.5 | 96.5 | 98.0 | 67.3 | 84.3 |
| | **w/ LwPTV** (Ours) | **76.7** | **88.5** | **87.7** | **96.3** | **90.7** | **97.6** | **98.7** | **75.6** | **89.0**$^{(+4.7)}$ |
| | TW AdaMerging++ (Yang et al., 2024b) | 76.6 | 86.2 | 85.6 | 94.6 | 89.7 | **96.8** | 98.2 | 72.1 | 87.5 |
| | **w/ LwPTV** (Ours) | **78.0** | **89.3** | **89.3** | **96.4** | **91.8** | 96.4 | **98.8** | **77.2** | **89.7**$^{(+2.2)}$ |
| | LW AdaMerging (Yang et al., 2024b) | 79.0 | 90.3 | 90.8 | **96.2** | **93.4** | **98.0** | **99.0** | **79.9** | **90.8** |
| | **w/ LwPTV** (Ours) | **79.0** | **90.3** | **91.6** | 95.7 | 89.3 | 96.4 | 98.8 | 77.6 | 89.8$^{(-1.0)}$ |
| | LW AdaMerging++ (Yang et al., 2024b) | **79.4** | **90.3** | 91.6 | **97.4** | **93.4** | **97.5** | **99.0** | **79.2** | **91.0** |
| | **w/ LwPTV** (Ours) | 79.2 | 90.1 | **92.4** | 96.4 | 89.1 | 94.9 | 98.8 | 76.6 | 89.7$^{(-1.3)}$ |
| | PCB-MERGING (Du et al., 2024) | 76.2 | 86.0 | 89.6 | **95.9** | **89.9** | 92.3 | **99.2** | 71.4 | 87.6 |
| | **w/ LwPTV** (Ours) | **76.8** | **88.2** | **91.1** | 95.8 | 89.2 | 92.1 | 99.1 | **72.7** | **88.1**$^{(+0.5)}$ |
| | FR-Merging (Zheng & Wang, 2025) | **76.4** | 87.0 | **90.2** | **96.8** | **92.0** | 92.8 | **99.3** | **71.5** | **88.3** |
| | **w/ LwPTV** (Ours) | 74.0 | **87.5** | 90.0 | 96.5 | 88.9 | **95.2** | 99.2 | 71.0 | 87.8$^{(-0.5)}$ |

Table 8: Performance comparison for ID datasets on ViT-H/14.

| Method (↓) | SUN397 | Cars | RESISC45 | EuroSAT | SVHN | GTSRB | MNIST | DTD | Avg. |
|---|---|---|---|---|---|---|---|---|---|
| Pretrained | 73.8 | 93.4 | 75.7 | 73.1 | 52.5 | 58.4 | 72.8 | 67.8 | 70.9 |
| Individual | 81.7 | 95.1 | 97.5 | 99.3 | 98.1 | 99.3 | 99.7 | 83.8 | 94.3 |
| Weight Averaging (Wortsman et al., 2022a) | 76.2 | 94.5 | 85.8 | 92.7 | 84.6 | 79.8 | 97.5 | 70.1 | 85.2 |
| Task Arithmetic (Ilharco et al., 2023) | 77.0 | 94.5 | 89.6 | 94.7 | 91.3 | **91.0** | **99.3** | 69.6 | 88.4 |
| **w/ LwPTV** (Ours) | **78.0** | **95.2** | **92.0** | **96.4** | **92.1** | 90.4 | 99.1 | **71.8** | **89.4**$^{(+1.0)}$ |
| Ties (Yadav et al., 2023) | 77.7 | 94.8 | 90.1 | 94.3 | 91.2 | 86.8 | 99.0 | 70.1 | 88.0 |
| **w/ LwPTV** (Ours) | **78.9** | **95.2** | **92.6** | **96.3** | **93.8** | **91.2** | **99.3** | **71.3** | **89.8**$^{(+1.8)}$ |
| PCB-MERGING (Du et al., 2024) | 77.8 | 94.6 | 90.6 | 96.2 | 93.5 | 95.1 | **99.4** | 76.4 | 90.5 |
| **w/ LwPTV** (Ours) | **78.9** | **95.3** | **92.8** | **96.7** | **94.0** | **95.2** | 99.3 | **78.7** | **91.4**$^{(+1.8)}$ |
| FR-Merging (Zheng & Wang, 2025) | 95.3 | 82.7 | **97.5** | **97.7** | **99.5** | 91.3 | **92.5** | **77.2** | **91.7** |
| **w/ LwPTV** (Ours) | **95.4** | **82.8** | 97.4 | 96.2 | 99.4 | **91.4** | 91.1 | 77.0 | 91.3$^{(-0.4)}$ |

Table 9: Performance comparison for ID and OOD datasets on ViT-H/14.

| Dataset (→)
Method (↓) | In Domain
Avg. | Out of Domain | | | | | | | | | | | | | | H-score |
|---|---|---|---|---|---|---|---|---|---|---|---|---|---|---|---|---|
| | | C10 | C100 | EMN | FMN | FER | F102 | F101 | KMN | OxP | PCA | RSS | STL | Ima | Avg. | |
| Pretrained | 70.9 | 97.4 | 84.7 | 14.1 | 79.0 | 36.2 | 80.1 | 92.2 | 11.7 | 94.5 | 51.0 | 64.0 | 98.5 | 77.9 | 67.8 | 69.3 |
| Individual | 94.3 | 91.7 | 72.6 | 18.7 | 74.2 | 37.4 | 77.7 | 88.8 | 10.1 | 93.4 | 52.1 | 63.9 | 97.4 | 75.0 | 65.6 | 77.4 |
| Weight Averaging (Wortsman et al., 2022a) | 85.2 | 97.1 | 83.0 | 21.5 | 79.7 | 38.2 | 79.7 | 91.5 | 9.9 | 94.3 | 51.3 | 63.9 | 98.6 | 77.7 | 68.2 | 75.8 |
| Task Arithmetic (Ilharco et al., 2023) | 88.4 | 94.4 | 74.8 | **28.9** | 78.7 | 38.1 | 77.9 | 88.0 | 9.4 | 93.3 | **52.3** | 62.9 | 97.6 | 75.0 | 67.0 | 76.2 |
| w/ **LwPTV** (Ours) | **89.4** | **96.4** | **81.4** | 28.6 | **79.9** | **39.0** | **79.3** | **91.6** | **9.7** | **94.3** | 51.7 | **64.1** | **98.5** | **77.7** | 68.6 $^{(+1.6)}$ | **77.6** $^{(+1.4)}$ |
| Ties (Yadav et al., 2023) | 88.9 | 95.0 | 76.4 | 27.0 | 78.8 | 38.4 | 78.6 | 89.2 | 9.2 | 93.8 | 51.9 | 64.1 | 98.0 | 76.0 | 67.4 | 76.7 |
| w/ **LwPTV** (Ours) | **89.8** | **96.0** | **79.9** | 27.7 | **79.5** | **38.8** | **79.1** | **91.4** | **9.5** | **94.3** | 51.7 | **64.1** | **98.5** | 77.5 | 68.3 $^{(+0.9)}$ | **77.6** $^{(+0.9)}$ |
| PCB-MERGING (Du et al., 2024) | 90.5 | 93.5 | 72.7 | 27.8 | 78.8 | 37.3 | 76.8 | 87.9 | 9.4 | 93.3 | **52.4** | 63.3 | 97.7 | 75.0 | 66.6 | 76.7 |
| w/ **LwPTV** (Ours) | **91.4** | **95.3** | **78.0** | **28.2** | **79.3** | **38.6** | **78.6** | **91.1** | 9.4 | **94.3** | 51.9 | **64.1** | **98.4** | **77.3** | 68.0 $^{(+1.4)}$ | **78.0** $^{(+1.3)}$ |
| FR-Merging (Zheng & Wang, 2025) | **91.7** | 94.5 | 75.7 | **29.9** | 78.7 | 38.3 | 77.4 | 90.4 | **9.7** | 93.7 | **52.8** | 64.1 | 98.2 | 76.5 | 67.7 | 77.9 |
| w/ **LwPTV** (Ours) | 91.3 | **96.4** | **81.0** | 29.8 | **80.0** | **38.8** | **78.8** | **91.6** | 9.6 | **94.4** | 52.7 | **64.3** | **98.6** | **77.7** | 68.7 $^{(+1.0)}$ | **78.4** $^{(+0.5)}$ |

Table 10: The ablation study of LwPTV on the ViT-B/32 regarding the ID dataset.

| Method | Components | | | Performance (%) | | | | | | | | |
|---|---|---|---|---|---|---|---|---|---|---|---|---|
| | Salience Score | ∨ | Scale | SUN397 | Cars | RESISC45 | EuroSAT | SVHN | GTSRB | MNIST | DTD | Avg. |
| Task Arithmetic (Ilharco et al., 2023) | ✗ | ✗ | - | 55.2 | 54.9 | 66.7 | 78.9 | 80.2 | 69.7 | 97.3 | 50.4 | 69.1 |
| | ✓ | ✗ | - | 68.1 | 65.7 | 76.0 | 77.4 | 69.2 | 61.2 | 87.1 | 54.4 | 69.9 |
| | ✗ | ✓ | - | 64.3 | 62.1 | 65.0 | 57.9 | 35.9 | 36.6 | 47.6 | 45.9 | 51.9 |
| | ✓ | ✓ | - | 66.8 | 66.0 | 77.2 | 79.1 | 78.3 | 68.6 | 92.3 | 54.4 | 72.8 |
| LW AdaMerging (Yang et al., 2024b) | ✗ | ✗ | ✗ | 64.5 | 68.1 | 79.2 | 93.8 | 87.0 | 91.9 | 97.5 | 59.1 | 80.1 |
| | ✓ | ✓ | ✗ | 66.0 | 67.7 | 82.0 | 91.4 | 87.5 | 87.7 | 97.3 | 59.0 | 79.8 |
| | ✓ | ✗ | ✓ | 68.6 | 66.6 | 79.8 | 83.7 | 79.1 | 75.4 | 91.4 | 57.3 | 75.2 |
| | ✗ | ✓ | ✓ | 64.5 | 62.5 | 67.2 | 62.4 | 38.3 | 39.7 | 47.8 | 46.7 | 53.6 |
| | ✓ | ✓ | ✓ | 68.0 | 68.5 | 80.7 | 88.3 | 80.0 | 81.2 | 94.1 | 57.7 | 77.3 |

Table 11: The ablation study of LwPTV on the ViT-B/32 regarding the OOD dataset.

| Method | Components | | | Performance (%) | | | | | | | | | | | | | | |
|---|---|---|---|---|---|---|---|---|---|---|---|---|---|---|---|---|---|---|
| | Salience Score | ∨ | Scale | C10 | C100 | EMN | FMN | FER | F102 | F101 | KMN | OxP | PCA | RSS | STL | Ima | Avg. |
| Task Arithmetic (Ilharco et al., 2023) | ✗ | ✗ | - | 76.3 | 41.9 | 28.5 | 63.9 | 26.3 | 49.4 | 55.8 | 9.0 | 75.4 | 54.0 | 53.4 | 87.8 | 45.0 | 51.3 |
| | ✓ | ✗ | - | 89.4 | 62.6 | 28.5 | 66.1 | 38.6 | 64.8 | 81.2 | 7.2 | 86.5 | 59.6 | 59.2 | 96.9 | 62.2 | 61.8 |
| | ✗ | ✓ | - | 90.3 | 64.4 | 17.9 | 63.8 | 39.4 | 66.5 | 82.7 | 9.4 | 86.9 | 59.0 | 57.0 | 97.1 | 62.7 | 61.3 |
| | ✓ | ✓ | - | 88.3 | 60.1 | 30.4 | 65.9 | 36.5 | 62.9 | 79.9 | 7.4 | 85.4 | 59.5 | 59.0 | 96.4 | 60.6 | 60.9 |
| LW AdaMerging (Yang et al., 2024b) | ✗ | ✗ | ✗ | 82.1 | 50.7 | 28.7 | 63.2 | 37.8 | 58.3 | 73.8 | 8.8 | 82.4 | 57.3 | 58.0 | 94.3 | 54.6 | 57.7 |
| | ✓ | ✓ | ✗ | 84.1 | 52.0 | 30.1 | 64.1 | 37.1 | 59.6 | 76.5 | 8.3 | 83.3 | 57.6 | 58.1 | 95.4 | 56.4 | 58.7 |
| | ✓ | ✗ | ✓ | 88.4 | 60.7 | 30.6 | 65.8 | 38.7 | 63.5 | 80.4 | 7.2 | 86.0 | 59.4 | 58.8 | 96.7 | 61.3 | 61.3 |
| | ✗ | ✓ | ✓ | 90.4 | 64.5 | 19.5 | 64.5 | 39.0 | 65.1 | 82.0 | 9.2 | 85.5 | 57.4 | 56.0 | 97.0 | 61.3 | 60.9 |
| | ✓ | ✓ | ✓ | 88.8 | 61.3 | 30.6 | 66.1 | 38.4 | 64.3 | 80.6 | 7.6 | 86.3 | 60.1 | 59.4 | 96.6 | 61.2 | 61.7 |

## B.8 Comparison with OOD methods for fine-tuned models

Tab.13 and Tab.14 provide a detailed summary of the experimental results comparing OOD methods for fine-tuned models on the ID dataset and OOD dataset. Following Wortsman et al. (Wortsman et al., 2022b), we set the mixing coefficient $\alpha = 0.5$ for WiSE-FT. For the Model Stock, we employed the fine-tuning code from Ilharco et al. (Ilharco et al., 2023) and set the random seed to 10 to obtain the second set of fine-tuned models. We set the search range for $\lambda$ in the Task Arithmetic to $(0, 1]$ with a step size of 0.1 based on Ilharco et al. (Ilharco et al., 2023; Wang et al., 2024a).

Table 12: The comparison of layer-wise pruning and parameter-wise pruning on ViT-B/32.

| Method (↓) | | H-score | Time consumption |
|---|---|---|---|
| Baseline | Task Arithmetic | 58.9 | 8.3 |
| Ours | w/ PwPTV | 64.2 | 135.4s |
| | w/ LwPTV | 66.3 | 9.0s |

## B.9 Comparison with other pruning methods

Tab.13 and Tab.14 provide a detailed summary of the experimental results comparing other pruning methods on the ID dataset and OOD dataset. In which, for the DARE and random mask methods, we select the best results obtained from three runs with different random seeds.

Table 13: Comparison with other OOD methods (above) and purning methods (below) on ViT-B/32 on the ID dataset.

| | Dataset ($\rightarrow$) Method ($\downarrow$) | SUN397 | Cars | RESISC4 | EuroSAT | SVHN | GTSRB | MNIST | DTD | Avg. |
|---|---|---|---|---|---|---|---|---|---|---|
| Baseline | Task Arithmetic (Ilharco et al., 2023) | 55.2 | 54.9 | 66.7 | 78.9 | 80.2 | 69.7 | 97.3 | 50.4 | 69.1 |
| OOD | w/ WiSE-FT (Wortsman et al., 2022b) | 60.6 | 59.0 | 70.2 | 79.4 | 78.2 | 68.6 | 96.1 | 51.5 | 70.5 |
| | Model Stock (Jang et al., 2024) | 60.7 | 65.0 | 75.0 | 81.7 | 80.0 | 68.5 | 95.2 | 49.6 | 72.0 |
| | w/ LiNeS (Wang et al., 2024a) | 63.9 | 63.9 | 75.1 | 85.6 | 79.4 | 72.2 | 96.2 | 56.5 | 74.1 |
| Pruning | w/ DARE (Yu et al., 2024) | 64.6 | 58.7 | 60.5 | 47.4 | 32.1 | 33.6 | 51.0 | 44.8 | 49.1 |
| | w/ MWP (Sanh et al., 2020) | 62.6 | 61.6 | 72.1 | 78.1 | 80.0 | 69.9 | 96.6 | 53.0 | 71.7 |
| | w/ random mask | 66.2 | 59.9 | 61.3 | 48.6 | 32.1 | 32.3 | 48.2 | 44.5 | 49.1 |
| | w/ absolute value | 64.6 | 61.0 | 60.7 | 47.1 | 32.0 | 32.8 | 48.4 | 44.4 | 48.9 |
| **Ours** | **w/ LwPT** | 66.8 | 66.0 | 77.2 | 79.1 | 78.3 | 68.6 | 92.3 | 54.4 | 72.8 |

Table 14: Comparison with other OOD methods (above) and purning methods (below) on ViT-B/32 on the OOD dataset.

| | Dataset ($\rightarrow$) Method ($\downarrow$) | C10 | C100 | EMN | FMN | FER | F102 | F101 | KMN | OxP | PCA | RSS | STL | Ima | Avg. |
|---|---|---|---|---|---|---|---|---|---|---|---|---|---|---|---|
| Baseline | Task Arithmetic (Ilharco et al., 2023) | 76.3 | 41.9 | 28.5 | 63.9 | 26.3 | 49.4 | 55.8 | 9.0 | 75.4 | 54.0 | 53.4 | 87.8 | 45.0 | 51.3 |
| OOD | w/ WiSE-FT (Wortsman et al., 2022b) | 81.6 | 49.9 | 29.6 | 64.5 | 30.3 | 55.2 | 65.9 | 8.4 | 81.1 | 56.7 | 54.2 | 91.4 | 52.0 | 55.4 |
| | Model Stock (Jang et al., 2024) | 86.6 | 57.7 | 30.9 | 65.1 | 32.1 | 60.8 | 74.9 | 8.7 | 84.2 | 57.0 | 54.4 | 94.9 | 56.5 | 58.7 |
| | w/ LiNeS (Wang et al., 2024a) | 84.4 | 54.9 | 30.3 | 65.4 | 33.6 | 60.4 | 73.7 | 8.4 | 83.6 | 58.8 | 51.4 | 93.9 | 55.6 | 58.0 |
| Pruning | w/ DARE (Yu et al., 2024) | 88.7 | 64.7 | 20.8 | 64.1 | 38.0 | 65.9 | 82.3 | 8.8 | 87.0 | 61.7 | 58.2 | 96.9 | 63.1 | 61.5 |
| | w/ MWP (Sanh et al., 2020) | 83.7 | 52.1 | 27.8 | 64.4 | 32.0 | 56.1 | 69.2 | 8.2 | 81.3 | 57.5 | 54.2 | 92.9 | 53.6 | 56.4 |
| | w/ random mask | 89.6 | 64.9 | 18.1 | 63.9 | 38.8 | 66.5 | 82.9 | 9.4 | 87.7 | 60.2 | 59.2 | 97.0 | 63.6 | 61.7 |
| | w/ absolute value | 89.9 | 64.6 | 17.5 | 63.7 | 38.8 | 66.6 | 82.9 | 9.4 | 87.4 | 60.2 | 57.9 | 97.2 | 63.3 | 61.5 |
| **Ours** | **w/ LwPT** | 88.3 | 60.1 | 30.4 | 65.9 | 36.5 | 62.9 | 79.9 | 7.4 | 85.4 | 59.5 | 59.0 | 96.4 | 60.6 | 60.9 |

### B.10 The analysis of layer-wise pruning and parameter-wise pruning

To investigate the distinctions between layer-wise pruning (LwPTV), which computes significance scores layer by layer, and parameter-wise pruning (PwPTV), which computes significance scores for individual parameters, under identical experimental setups, we conducted a comparative analysis, with a focus on performance and time consumption, based on the Task Arithmetic method on the ViT-B/32 architecture. The results are delineated in Tab.12. The detailed results of the two methods on the ID and OOD datasets are presented in Tab.15 and Tab.16, respectively. From this, it can be observed that LwPTV achieves better performance with less time consumption. This performance improvement can be attributed to LwPTV, which aligns the representations of OOD data generated by the merged model more closely with those produced by the pretrained model, thereby significantly enhancing OOD performance.

### B.11 Neural network component analysis

To investigate whether our method primarily prunes task vectors specific to certain structures, such as parameters in MLP, whose pre-trained parameters inherently have stronger generalization capabilities, leading to the effectiveness of LwPTV. We conducted a more in-depth analysis of ours by categorizing the layers of ViT into three types: **Attention Layer**, **MLP Layer**, and **Other Layer** (including position embedding, layer normalization, etc.). We then analyzed their salience scores of the task vectors for SUN397. As shown in Fig.11 the salience scores do not reflect this trend, suggesting that ours does not selectively filter out specific types of layers. Therefore, the effectiveness of ours lies in pruning the redundant parameters, instead of pruning certain layer types.

### B.12 LwPTV aligns the representations between the merged model and the pretrained model

To investigate how our method LwPTV influences the merged model in generating representations for OOD data, we compute two sets of $\ell_2$ distances: (1) Task Arithmetic *vs.* Pretrained model (2) Task Arithmetic w/ LwPTV *vs.* Pretrained model. The results are shown in Fig.10. It can be seen that LwPTV aligns the representations between the merged model and the pretrained model. This observation is consistent with our goal and further supports the notion that our method improves OOD performance.

Table 15: The comparison of layer-wise pruning and parameter-wise pruning on ID datasets.

| | Dataset (→)
Method (↓) | SUN397 | Cars | RESISC4 | EuroSAT | SVHN | GTSRB | MNIST | DTD | Avg. |
|---|---|---|---|---|---|---|---|---|---|---|
| Baseline | Task Arithmetic (Ilharco et al., 2023) | 55.2 | 54.9 | 66.7 | 78.9 | 80.2 | 69.7 | 97.3 | 50.4 | 69.1 |
| Ours | w/ PwPTV | 64.0 | 63.5 | 74.2 | 78.3 | 82.3 | 71.7 | 96.7 | 54.4 | 73.1 |
| | w/ LwPTV | 66.8 | 66.0 | 77.2 | 79.1 | 78.3 | 68.6 | 92.3 | 54.4 | 72.8 |

Table 16: The comparison of layer-wise pruning and parameter-wise pruning on OOD datasets.

| | Dataset (→)
Method (↓) | C10 | C100 | EMN | FMN | FER | F102 | F101 | KMN | OxP | PCA | RSS | STL | Ima | Avg. |
|---|---|---|---|---|---|---|---|---|---|---|---|---|---|---|---|
| Baseline | Task Arithmetic (Ilharco et al., 2023) | 76.3 | 41.9 | 28.5 | 63.9 | 26.3 | 49.4 | 55.8 | 9.0 | 75.4 | 54.0 | 53.4 | 87.8 | 45.0 | 51.3 |
| Ours | w/ PwPTV | 84.2 | 53.00 | 27.8 | 64.6 | 32.5 | 57.9 | 72.4 | 8.2 | 82.0 | 57.9 | 55.0 | 93.6 | 55.2 | 57.3 |
| | w/ LwPTV | 88.3 | 60.1 | 30.4 | 65.9 | 36.5 | 62.9 | 79.9 | 7.4 | 85.4 | 59.5 | 59.0 | 96.4 | 60.6 | 60.9 |

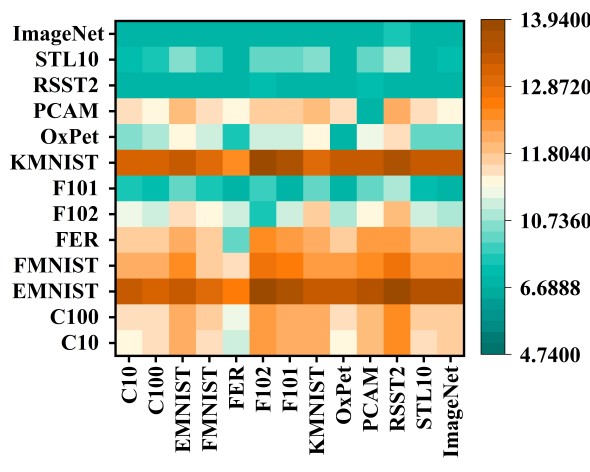

(a) Task Arithmetic *vs.* Pretrained model

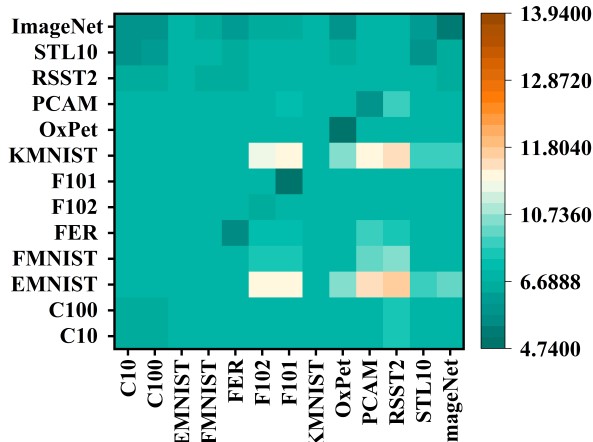

(b) Task Arithmetic w/ LwPTV *vs.* Pretrained model

Figure 10: Visualization of $\ell_2$ distance between merged model representations and pre-trained model representations.

## B.13 Additional analysis on out-of-domain (OOD) datasets

To further clarify the behavior of our method on out-of-domain (OOD) tasks, we include here a detailed supplementary analysis. Specifically, we investigate how the effectiveness of LwPTV varies with the degree of distributional shift between in-domain (ID) and OOD datasets. This experiment complements Sec.5.1 in the main paper and aims to demonstrate that our method primarily benefits truly challenging OOD scenarios rather than those that are superficially similar to the ID data.

Table 17: Performance and distance comparison on ViT-B/32.

| Groups (↓) | Method (↓) | OOD Acc. (%) | Distance |
|---|---|---|---|
| Demanding OOD | Task Arithmetic | 51.3 | 6.3 |
| | w / LwPTV (Ours) | 60.9 | |
| Easy OOD | Task Arithmetic | 61.7 | 3.5 |
| | w / LwPTV (Ours) | 60.9 | |

**Experiment setup:** We created two test groups. One is demanding OOD, i.e., our main benchmark of 13 OOD datasets and 8 ID tasks used in the paper. Another is easy OOD, a control group where we used two of the ID datasets (MNIST and EuroSAT) as the OOD tasks and set the remaining six as ID tasks. To prove the "easy" group was indeed more similar, we measured the Mahalanobis distance between the ID and OOD datasets, resulting in an 8×13 distance matrix for demanding OOD and a 6×2 matrix for easy OOD in the control group. As shown in Tab.17, the averaged distance for the "easy" group was much lower (3.5) than for our main demanding benchmark (6.3), confirming it was less of a challenge. **Results and insight:** We evaluate Task Arithmetic and Task Arithmetic + LwPTV under both settings. As shown in Tab. 17, LwPTV yields a large improvement under the demanding OOD setting but no gain under the Easy OOD setting. This outcome indicates that LwPTV is not a generic booster but a principled method that improves

performance on challenging, truly out-of-distribution scenarios by preserving the generalizable features of the pre-trained model.

## B.14 Analysis of layers with high mean magnitude but low variance

To further examine the validity of our salience score and explore the role of layers with high mean absolute values but low variance, we conducted an additional analysis on the ViT-B/32 model. First, in the ViT-B/32 model, we identified layers that exhibit high mean absolute values and low cross-task variance, and experimentally verified that removing these layers does not harm performance. Specifically, we found 17 such layers, all of which are pruned by our proposed method, LwPTV. Among them, 16 are bias layers and the remaining one is the final layer normalization weight (model.ln_final.weight). We then evaluated two variants of our pruning strategy: (V1) we keep all 17 of these layers unpruned, and (V2) we prune only the final layer normalization weight while keeping the other 16 bias layers. As shown in the Tab.18, neither variant improves performance over our method, suggest-

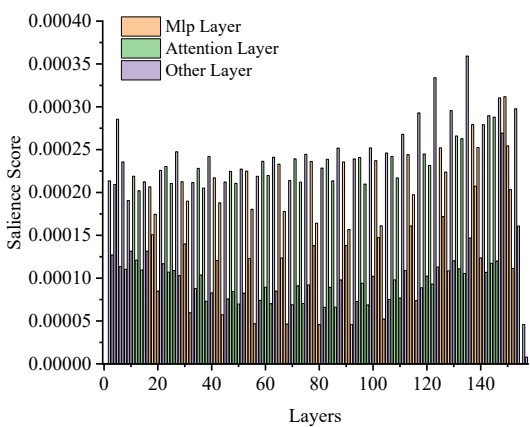

Figure 11: The salience score of the task vectors with different components on ViT-B/32.

ing that the salience score correctly identifies these layers as redundant and pruning them helps improve OOD generalization. Second, from the perspective of feature learning, we provide a simple theoretical justification for why pruning layers with high mean absolute values but low cross-task variance, which are mainly bias layers, is reasonable. Consider a linear layer $f(x) = Wx + b$. During gradient updates, the gradient with respect to $W$ depends on the input $x$, so the corresponding component of the task vector, i.e., the accumulated gradient updates for $W$, contains task-specific information related to $x$. In contrast, the gradient with respect to $b$ is independent of the input, and thus the corresponding component of the task vector lacks discriminative information across tasks. As a result, although the magnitude of the task vectors for the bias may be large, they lack diversity. According to Proposition 1, our diversity-based salience score is designed to identify and prune precisely such non-diverse, task-irrelevant layers, as its large magnitude is misleading in the context of finding task-specific information.

## B.15 Effect of the number of in-domain tasks on LwPTV

To further assess the stability of our salience score and examine whether LwPTV depends strongly on the number of in-domain tasks $K$, we conduct an additional study on ViT-B/32 by varying $K$ from 2 to 8. This analysis complements our main results (where $K = 8$) and evaluates whether the diversity-based salience estimation remains reliable in low-task settings. As shown in Tab.19, the relative H-score improvement when applying LwPTV to the Task Arithmetic baseline increases steadily as $K$

Table 18: Performance comparison on ViT-B/32.

| Method (↓) | ID | OOD | H-score |
|---|---|---|---|
| Task Arithmetic | 69.1 | 51.3 | 58.9 |
| V1 (keep 17 layers) | 72.9 | 61.0 | 66.4 |
| V2 (keep 16 layers) | 72.9 | 61.0 | 66.4 |
| Ours | 72.8 | 60.9 | 66.3 |

grows. This trend aligns with Proposition 1: our salience score identifies task-specific features (high diversity) by measuring deviations from the cross-task mean, and larger $K$ yields a more stable estimate of "common" adaptation. As a result, a greater number of tasks creates a clearer signal that leads to more precise pruning and larger OOD gains.

Importantly, LwPTV also remains effective under low-task scenarios. Even with only $K = 2$ tasks, we observe a substantial improvement of +2.0 in H-score over the baseline, indicating that the method is robust and beneficial even with a small number of tasks. Overall, these findings empirically support the theoretical analysis and confirm that LwPTV scales favorably with the number of tasks while retaining practical utility in small-$K$ scenarios.

## B.16 Effect of normalization on the salience score

To evaluate whether differences in parameter scale across layers may bias the salience score computation, we conduct a controlled experiment applying layer-wise normalization to task vectors before computing salience. Specifically, for each layer, we subtract the mean and divide by the standard deviation of its task-vector entries. This standardization enforces equal scale across layers and removes magnitude differences.

Table 19: Relative performance improvement as the number of ID tasks changes.

| Task Num | 2 | 3 | 4 | 5 | 6 | 7 | 8 |
|---|---|---|---|---|---|---|---|
| H-score | +2.0 | +3.3 | +3.5 | +3.8 | +4.5 | +6.0 | +7.4 |

As shown in Tab.20, this normalization substantially degrades performance: the H-score drops from 66.3 to 59.1 on ViT-B/32. The reason for this is that for task vectors, the scale of parameters across layers is a meaningful signal that reflects task-specific adaptation. The original magnitude and variance of the parameters at different layers naturally encode their relative importance for the downstream tasks. Normalization erases this vital information by putting all layers on an equal footing. This can amplify noise in layers with small-magnitude changes and mislead the pruning process. Therefore, we compute salience score in the original parameter space to preserve the informative structural differences across layers.

### B.17  Comparison of different mask aggregation n strategies

Table 21: Comparison of different mask aggregation strategies within LwPTV on ViT-B/32.

| Method | Mask Aggregation | ID Avg. | OOD Avg. | H-score |
|---|---|---|---|---|
| Task Arithmetic | w/ LwPTV (Majority) | 72.3 | 61.1 | 66.3 |
| | w/ LwPTV (Weighted) | 72.6 | **61.2** | **66.4** |
| | w/ LwPTV (Intersection) | 72.3 | 59.5 | 65.3 |
| | w/ LwPTV (OR) | **72.8** | 60.9 | 66.3 |
| Ties | w/ LwPTV (Majority) | 72.2 | 61.2 | 66.3 |
| | w/ LwPTV (Weighted) | 72.4 | 61.3 | 66.4 |
| | w/ LwPTV (Intersection) | 72.5 | 60.6 | 66.0 |
| | w/ LwPTV (OR) | **72.7** | **61.3** | **66.5** |
| PCB | w/ LwPTV (Majority) | 76.8 | **59.8** | 67.2 |
| | w/ LwPTV (Weighted) | 77.0 | 59.8 | 67.3 |
| | w/ LwPTV (Intersection) | 76.8 | 58.1 | 66.1 |
| | w/ LwPTV (OR) | **77.3** | 59.7 | **67.4** |

To further examine the choice of mask aggregation, we compare the OR rule with three alternative strategies in Tab.21: majority voting, weighted voting, and intersection masking. Majority voting retains a layer when it is selected by at least half of the tasks, weighted voting aggregates task votes using weights computed from the accuracy gain of each fine-tuned model over the pretrained model, and intersection masking retains only layers selected by all tasks. These strategies cover different levels of masking strictness, from the more inclusive OR rule to the more restrictive intersection rule.

Table 20: Effect of layer-wise normalization on H-score (ViT-B/32).

| Method | H-score |
|---|---|
| Task Arithmetic | 58.9 |
| w/ LwPTV (Ours) | 66.3 |
| w/ LwPTV + Normalization | 59.1 |

The results show that OR achieves the best overall performance among different aggregation strategies. It achieves the highest ID average for Task Arithmetic, TIES, and PCB, and obtains the best H-score for TIES and PCB while remaining close to the best result for Task Arithmetic. In contrast, intersection masking generally leads to lower OOD and H-score performance, indicating that overly strict aggregation may remove updates that are useful for certain tasks. Majority and weighted voting achieve comparable performance, but they introduce additional assumptions. Ma-

Table 22: Performance comparison on ViT-B/32.

| Method | ID | DermaMNIST | H-score |
|---|---|---|---|
| Pretrained | 48.0 | 13.7 | 21.3 |
| Task Arithmetic | 69.1 | 8.0 | 14.3 |
| w/ LwPTV | **72.8** | **11.6** | **20.0** |
| TIES | **72.9** | 10.0 | 17.6 |
| w/ LwPTV | 72.7 | **12.1** | **20.7** |
| PCB | 75.8 | 8.3 | 15.0 |
| w/ LwPTV | **77.3** | **12.2** | **21.1** |

jority voting treats all tasks equally and may discard task-specific updates that are important for individual tasks, while weighted voting requires estimating task weights. In contrast, the OR rule preserves updates selected by any task without introducing additional weighting schemes. Overall, the OR rule provides a reasonable trade-off between retaining task-relevant updates and removing redundant components.

Table 23: Comparison with the post-merge interpolation baseline on ViT-B/32.

| Merging Method | Variant | $\alpha$ | ID Avg | OOD Avg | H-score |
|---|---|---|---|---|---|
| Task Arithmetic | Original merge | 1.0 | 69.1 | 51.3 | 58.9 |
| | Post-interp. baseline | 0.9 | 70.1 | 54.0 | 61.0 |
| | w/ LwPTV | – | **72.8** | **60.9** | **66.3** |
| Ties | Original merge | 1.0 | **72.9** | 58.0 | 64.6 |
| | Post-interp. baseline | 0.9 | 72.0 | 59.0 | 64.8 |
| | w/ LwPTV | – | 72.7 | **61.3** | **66.5** |
| PCB | Original merge | 1.0 | 75.8 | 53.2 | 62.5 |
| | Post-interp. baseline | 0.9 | 68.5 | **60.7** | 64.4 |
| | w/ LwPTV | – | **77.3** | 59.7 | **67.4** |

*Note:* The post-interpolation baseline is applied after each merging method as $\theta_\alpha = \theta_{\mathrm{pre}} + \alpha(\theta_{\mathrm{merged}} - \theta_{\mathrm{pre}})$. The original merged model corresponds to $\alpha = 1.0$.

### B.18 Additional Evaluation on Medical-Domain OOD Dataset

To further evaluate the robustness of model merging across different visual domains, we conduct an additional evaluation on the medical imaging dataset DermaMNIST. The results are reported in Tab. 22. DermaMNIST is substantially different from both our eight fine-tuning datasets and the 13 OOD datasets, providing an additional evaluation under a specialized domain shift. On ViT-B/32, LwPTV consistently improves performance across the tested merging methods. DermaMNIST accuracy improves from 8.0 to 11.6 for Task Arithmetic, from 10.0 to 12.1 for TIES, and from 8.3 to 12.2 for PCB. The corresponding H-scores improve from 14.3 to 20.0, from 17.6 to 20.7, and from 15.0 to 21.1, respectively.

Although the absolute performance on DermaMNIST remains limited due to the large domain gap of medical images, this experiment is not designed to evaluate zero-shot medical recognition ability. Instead, it examines whether model merging introduces harmful interference under a domain shift. Task Arithmetic decreases the pretrained model performance on this domain (e.g., from 13.7 to 8.0), while LwPTV partially recovers this degradation by pruning low-salience task-vector layers and keeping the merged model closer to the pretrained state.

### B.19 Pruning Ratio Sensitivity on ViT-L/14

To investigate the sensitivity of the pruning ratio $\eta$ on larger architectures, we perform an additional sweep of $\eta$ on ViT-L/14. The results are shown in Fig.12. For Task Arithmetic, the best H-score is achieved around $\eta = 0.6$–$0.7$, with both settings reaching 75.7. For TIES, the best performance is obtained at $\eta = 0.7$, achieving an H-score of 76.1. For PCB-Merging, $\eta = 0.6$ and $\eta = 0.7$ achieve the same best H-score of 76.4. Overall, these results show that $\eta = 0.7$ provides a robust choice across different merging baselines on ViT-L/14, consistent with the observation on ViT-B/32.

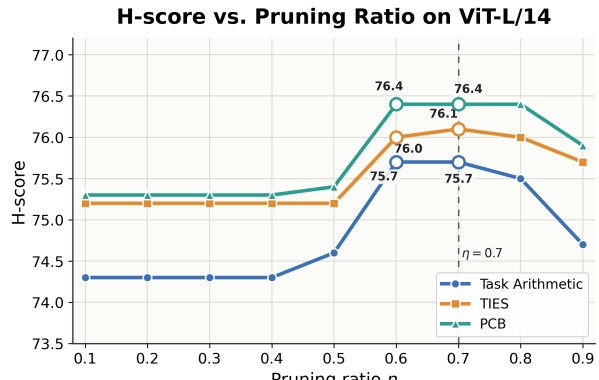

Figure 12: Effect of the pruning ratio $\eta$ on ViT-L/14.

### B.20 Distinguishing LwPTV from Generic Shrinkage and CART

Table 24: Comparison with CART under the same ID/OOD protocol on ViT-B/32.

| Method | ID Avg | OOD Avg | H-score |
|---|---|---|---|
| CART | 84.7 | 56.1 | 67.5 |
| Task Arithmetic w/ LwPTV | 72.8 | 60.9 | 66.3 |
| LW-AdaMerging w/ LwPTV | 77.3 | 61.7 | 68.6 |
| LW-AdaMerging++ w/ LwPTV | 77.1 | 61.4 | 68.4 |

Generic shrinkage, CART, and LwPTV all modify task-vector updates in ways that may reduce interference with pretrained representations. To disentangle these effects and clarify the specific contribution of LwPTV, we conduct two additional comparisons under the same ID/OOD evaluation protocol on ViT-B/32. First, generic shrinkage toward

the pretrained model reduces the overall magnitude of the merged task vector:

$$\theta_\alpha = \theta_{\text{pre}} + \alpha \left( \theta_{\text{merged}} - \theta_{\text{pre}} \right), \quad \alpha < 1. \tag{18}$$

This can improve OOD robustness because the merged model remains closer to the pretrained model. However, this operation is task-agnostic: it uniformly suppresses all task-vector updates, including both useful and harmful ones. As shown in Tab.23, such shrinkage can improve OOD accuracy, but it does not reliably preserve ID performance.

Second, CART uses centered task vectors and low-rank reconstruction. In the implementation we examined, CART first constructs a cross-task mean model, $\theta_{\text{avg}} = \theta_0 + \frac{1}{T} \sum_i \tau_i$, then centers each task-specific model around this mean and applies low-rank reconstruction to the centered residuals. The reconstructed centered components are finally added back to $\theta_{\text{avg}}$. Thus, CART mainly relies on cross-task centering and low-rank reconstruction to preserve shared in-domain components and reduce task interference. As shown in Tab.24, CART achieves the highest ID Avg of 84.7, which is consistent with its strength in preserving shared in-domain information. However, its OOD Avg is 56.1, lower than all LwPTV variants.

LwPTV follows a different mechanism. It neither uniformly shrinks the merged update nor reconstructs centered residuals. Instead, it estimates the salience of task-vector tensors and selectively prunes low-salience tensors before merging. As shown in Tab.24, LW-AdaMerging w/ LwPTV achieves an OOD Avg of 61.7 and the highest H-score of 68.6. Overall, CART better retains ID performance, while LwPTV achieves a stronger ID–OOD trade-off.

### B.21 Storage overhead

LwPTV can effectively diminish the storage requirements for task vectors for each merging method. As shown in Tab.25, we report the storage cost derived from various model merging techniques applied to the ViT-B/32 architecture. It can be seen that, LwPTV can significantly reduce the storage cost of the merged model in various methods. Since both Ties and PCB prune the parameters, applying ours to them further lead to smaller storage cost.

Table 25: The storage cost of various model merging methods on ViT-B/32 and ViT-L/14.

| Model Architecture | Method (→) w/LwPTV (↓) | Pretrained model | Task Arithmetic | Ties | TW AdaMerging | TW AdaMerging++ | LW AdaMerging | LW AdaMerging++ | PCB | FR-Merging |
|---|---|---|---|---|---|---|---|---|---|---|
| ViT-B/32 | ✗ | 432.77 MB | 432.77 MB | 285.25 MB | 432.77 MB | 285.25 MB | 432.77 MB | 285.25 MB | 123.39 MB | 432.77 MB |
| | ✓ | - | 226.75 MB | 220.03 MB | 226.75 MB | 220.03 MB | 226.75 MB | 220.03 MB | 115.23 MB | 226.75 MB |
| ViT-L/14 | ✗ | 1306.77 MB | 1306.77 MB | 965.18 MB | 1306.77 MB | 965.18 MB | 1306.77 MB | 965.18 MB | 397.26 MB | 1306.77 MB |
| | ✓ | - | 836.35 MB | 788.08 MB | 836.35 MB | 788.08 MB | 836.35 MB | 788.08 MB | 377.21 MB | 836.35 MB |

### B.22 Time complexity analysis

To investigate whether our method incurs significant computational overhead for layer-wise pruning on larger models, we conducted a time complexity analysis based on the ViT-H/14 architecture. Suppose the model has $L$ layers, the $l$-th layer has $d_l$ parameters, and $K$ models need to be merged: (1)

Table 26: Time complexity on ViT-H/14.

| Method (↓) | w/o LwPTV | w/ LwPTV |
|---|---|---|
| Task Arithmetic (Ilharco et al., 2023) | 57.5s | 74.9s |
| Ties (Yadav et al., 2023) | 147.2s | 214.8s |
| PCB (Du et al., 2024) | 1861.0s | 1928.6s |
| FR-Merging (Zheng & Wang, 2025) | 692.64s | 710.04s |

The time complexity of calculating $\mathbb{E}\left( \left| \boldsymbol{\tau}_k^l - \frac{1}{K} \sum_{k=1}^K \boldsymbol{\tau}_k^l \right| \right)$ is $\mathcal{O}(KLd_l)$. (2) When the $l$-th layer of the final mask $\hat{m}$ is 0, we directly set the $l$-th layer parameters of the merged task vector to 0. In the worst case, all parameters need to be set to 0. In this case, the time complexity is $\mathcal{O}(Ld_l)$. So the total time complexity is $\mathcal{O}(KLd_l)$. In addition, we list the time overhead of Task Arithmetic, Ties, PCB and FR-Merging with and without LwPTV on ViT-H/14 in Tab.26. The results show that whether we introduce ours into Task Arithmetic, Tie, PCB or FR-Merging, it takes no more than 67.6 seconds. The time introduced by ours is acceptable or can even be ignored, especially for the model merging algorithms, which are relatively time-consuming such as PCB.

Table 27: Performance comparison for ID and OOD datasets on T5-large.

| Dataset (→) | In Domain | Out of Domain | | | | | | | | | H-score |
| Method (↓) | Avg. | rte | cb | wic | copa | h-swag | anli-r1 | anli-r2 | anli-r3 | Avg. | |
|---|---|---|---|---|---|---|---|---|---|---|---|
| Pretrained | 44.9 | 53.1 | 33.9 | 51.4 | 55.0 | 30.5 | 32.3 | 35.4 | 32.8 | 40.6 | 42.7 |
| Individual | 85.6 | 56.0 | 39.8 | 51.1 | 58.3 | 29.6 | 32.5 | 34.1 | 33.2 | 41.8 | 56.2 |
| Weight Averaging (Wortsman et al., 2022a) | 60.5 | 47.7 | 48.2 | 49.5 | 55.0 | 30.1 | 33.7 | 34.9 | 33.3 | 41.5 | 49.2 |
| Task Arithmetic (Ilharco et al., 2023) | 73.3 | 48.0 | 50.0 | 49.7 | **73.0** | 28.6 | **34.1** | 34.9 | **33.1** | 43.9 | 54.9 |
| w/ Ours | **76.3** | **66.1** | **57.1** | **50.3** | 71.0 | **28.7** | 31.5 | **34.9** | 33.0 | 46.6 (+2.7) | 57.9 (+3.0) |
| Ties (Yadav et al., 2023) | 71.0 | 64.3 | 62.5 | **52.5** | 70.0 | 30.2 | **33.9** | 35.8 | **35.0** | 48.0 | 57.3 |
| w/ Ours | 71.9 | **67.9** | **67.9** | 52.4 | **74.0** | **31.2** | 33.0 | **37.2** | 34.8 | 49.8 (+1.8) | 58.8 (+1.5) |
| PCB-MERGING (Du et al., 2024) | **75.9** | 70.0 | **64.3** | 51.7 | 73.0 | 30.6 | **35.1** | 36.4 | 34.3 | 49.4 | 59.8 |
| w/ Ours | 75.8 | **70.0** | 62.5 | **52.2** | **78.0** | **31.2** | 35.0 | 36.3 | 33.7 | 49.9 (+0.5) | 60.2 (+0.4) |
| FR-Merging (Zheng & Wang, 2025) | 73.9 | 50.9 | 48.2 | 50.5 | 72.0 | 28.3 | 34.2 | 34.3 | 32.8 | 43.9 | 50.1 |
| w/ Ours | 75.4 | 52.3 | 48.2 | 50.3 | 74.0 | 28.9 | 34.5 | 34.6 | 32.5 | 44.4 (+0.5) | 55.9 (+0.8) |

Table 28: Performance comparison for ID datasets on T5-large.

| Method (↓) | PAWS | QASC | QuaRTz | StoryCloze | WikiQA | Winogrande | WSC | Avg. |
|---|---|---|---|---|---|---|---|---|
| Pretrained | 48.1 | 33.8 | 53.1 | 47.0 | 37.9 | 50.9 | 43.3 | 44.9 |
| Individual | 95.4 | 97.6 | 91.9 | 90.4 | 95.8 | 79.2 | 51.0 | 85.9 |
| Weight Averaging (Wortsman et al., 2022a) | 55.9 | 71.6 | 54.4 | 56.4 | 76.7 | 53.4 | 54.8 | 60.5 |
| Task Arithmetic (Ilharco et al., 2023) | 64.6 | 74.4 | 75.0 | 80.8 | **92.8** | 63.1 | **62.5** | 73.3 |
| w/ Ours | **85.5** | **77.4** | **77.9** | **81.0** | 85.7 | **67.2** | 59.6 | 76.3 (+3.0) |
| Ties (Yadav et al., 2023) | 92.6 | 62.6 | 75.0 | 78.5 | **58.7** | 75.7 | 53.8 | 71.0 |
| w/ Ours | **92.8** | **67.0** | **77.1** | **78.8** | 55.6 | **76.3** | **55.8** | 71.9 (+0.9) |
| PCB-MERGING (Du et al., 2024) | **92.3** | 78.4 | **77.3** | **76.8** | 70.8 | **74.4** | 61.5 | **75.9** |
| w/ Ours | 92.2 | **79.0** | 76.6 | 76.0 | 69.9 | 74.3 | **62.5** | 75.8 (-0.1) |
| FR-Merging (Zheng & Wang, 2025) | 84.7 | 66.0 | 70.8 | 78.4 | 92.9 | 64.9 | 59.6 | 73.9 |
| w/ Ours | 85.8 | 68.6 | 71.9 | 81.5 | 93.1 | 65.4 | 61.5 | 75.4 (+1.5) |

## B.23 Computational resources

Following Ilharco et al. (Ilharco et al., 2023), we fine-tune separate instances of the pretrained ViT-H/14 on eight distinct datasets: Cars, DTD, EuroSAT, GTSRB, MNIST, RESISC45, SUN397, and SVHN. Training employs the AdamW optimizer with a learning rate of 0.001 and a batch size of 128. To ensure convergence while mitigating overfitting, dataset-specific training epochs are assigned: 35 epochs for Cars, 76 for DTD, 12 for EuroSAT, 11 for GTSRB, 5 for MNIST, 15 for RESISC45, 14 for SUN397, and 4 for SVHN. All implementations use PyTorch on NVIDIA A800 GPUs.

## B.24 NLP task

To evaluate the performance of our method on NLP tasks, we conducted experiments using T5-Large-LM-Adapt as the base model. We selected PAWS, QASC, QuaRTz, StoryCloze, WikiQA, Winogrande, and WSC as the ID datasets (Derek et al., 2024), and rte, cb, wic, copa, h-swag, anli-r1, anli-r2, and anli-r3 as the OOD datasets (Sanh et al., 2022). As shown in Tab.27, in contrast to image classification tasks, the merged model demonstrates significantly better OOD performance than the pretrained model in the field of NLP. This superiority can be attributed to the high degree of semantic space overlap across different tasks, domains, and fields in NLP (Dosovitskiy et al., 2020; Hou & Kung, 2022). This overlap implies that the ID task vectors contain information that is beneficial for OOD tasks (Arora et al., 2021; Lewis et al., 2021; Elangovan et al., 2021). In other words, the discriminative patterns of OOD tasks have a non-zero mapping onto the discriminative patterns of ID tasks (Li et al., 2025). Under these circumstances, enhancing the OOD performance of the merged model requires amplifying the role of task-specific parameters within the ID task, which can be achieved by pruning redundant parameters. Unlike images, which are spatial and have higher redundancy with many local blocks containing overlapping information, allowing for robust compression by deleting entire Transformer layers while preserving important global information and reducing overfitting to high-frequency noise or specific dataset artifacts (Chen et al., 2022; Hou & Kung, 2022), language is sequential with meanings that are compositional and context-sensitive. Discarding entire layers might disrupt the sentence-level or token-level dependencies crucial for semantic understanding (Sajjad et al., 2023; Ding

et al., 2025; Kenton & Toutanova, 2019). Therefore, for NLP tasks, we perform parameter pruning at the parameter level. As shown in Tab.27, compared to the baseline methods Task Arithmetic, Ties, PCB, and FR-Merging, our method achieves relative improvements of 3%, 1.5%, 0.4% and 0.8% on the H-score, respectively. The detailed results on the ID dataset are presented in Tab.28.

### B.25  Parameter settings

Since the tasks and model architectures vary across our experiments, our hyperparameter selection strategy also differs accordingly.

#### B.25.1  Vision tasks

The vision experiments involve ViT-B/32, ViT-L/14, and ViT-H/14. To ensure fair comparison, we follow the tuning protocol below:

**Tuning protocol.**

- **ViT-B/32 / ViT-L/14:** We use the optimal hyperparameters reported in prior works and evaluate both baseline and baseline+Ours under these fixed settings.

- **ViT-H/14:** Since ViT-H/14 is fine-tuned by us, we first conduct a grid search within the hyperparameter ranges suggested by prior works to identify the optimal configuration for the baseline. We then apply the same configuration to the baseline+Ours setting for a fair comparison.

**Hyperparameters.** Following (Yadav et al., 2023; Yang et al., 2024b;a; Du et al., 2024; Zheng & Wang, 2025), we use:

- **Task Arithmetic:** $\lambda = 0.3$

- **Ties:** $\lambda = 0.3$ (ViT-B/32), $\lambda = 0.4$ (ViT-L/14, ViT-H/14); retain the top-20% parameters

- **PCB:** $\lambda = 1.2$, mask ratio $r = 0.05$

- **FR-Merging:** $\lambda = 0.7$(ViT-B/32, ViT-L/14), $\lambda = 0.6$(ViT-H/14); filter ratio $f = 0.7$;

- **Ours (LwPTV):** pruning ratio $\eta = 0.7$

In practice, when a small ID validation set is available, we recommend sweeping a narrow range around 0.6–0.8 and selecting $\eta$ according to ID performance or the desired ID-side constraint. OOD datasets should be reserved for final evaluation rather than used for selecting $\eta$. When no such validation set is available, $\eta = 0.7$ is a reasonable default based on our results on both ViT-B/32 and ViT-L/14.

#### B.25.2  NLP Tasks

For T5-Large-LM-Adapt, although baseline and baseline+Ours use separate hyperparameter selections, the search ranges are consistent across both settings. Specifically, for models based on T5-Large-LM-Adapt, the existing baseline results are incomplete and were obtained with relatively narrow hyperparameter ranges. To ensure fairness, we independently tune the hyperparameters for both the baseline and baseline+Ours within the broader ranges provided in the relevant literature.

**Search ranges.** Following (Ilharco et al., 2023; Yadav et al., 2023; Du et al., 2024; Derek et al., 2024; Zheng & Wang, 2025), we adopt:

- **Task Arithmetic:** $\lambda \in [0.1, 1.5]$ (step 0.1)

- **Ties / PCB:** mask ratio $r \in \{0.05, 0.1, 0.2\}$; $\lambda \in [0.1, 2.5]$ (step 0.1);

- **FR-Merging:** $\lambda \in [0.1, 1.0]$ (step 0.1); filter ratio $f \in [0.1, 1.0]$ (step 0.1)

- **Ours (LwPTV):** pruning ratio $\eta \in [0.1, 0.7]$ (step 0.1)

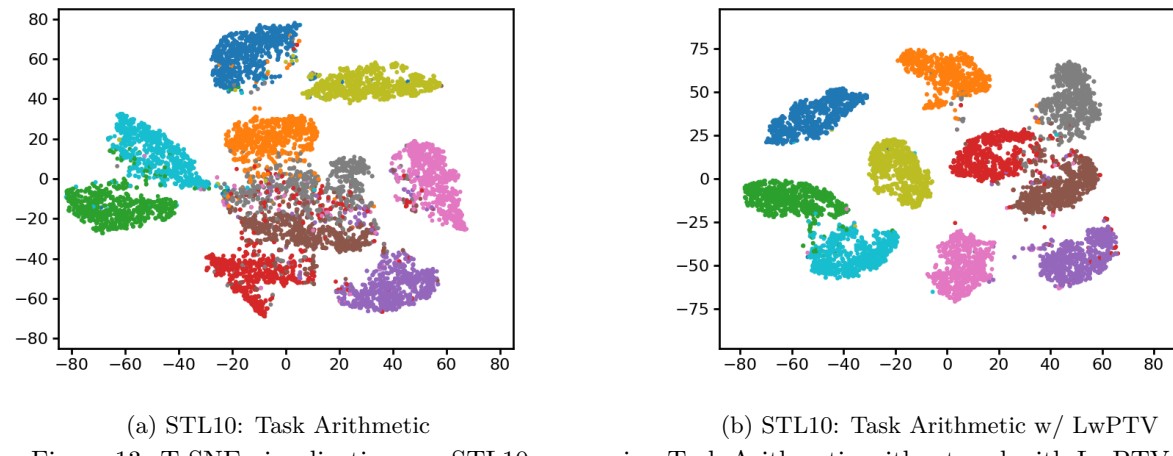

(a) STL10: Task Arithmetic            (b) STL10: Task Arithmetic w/ LwPTV

Figure 13: T-SNE visualizations on STL10 comparing Task Arithmetic without and with LwPTV .

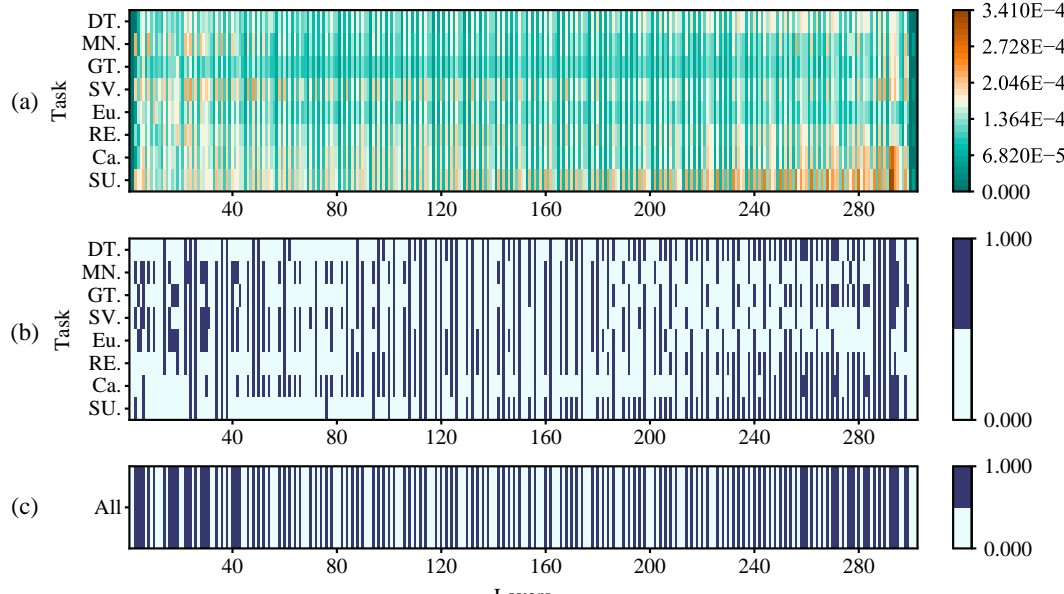

Figure 14: Visualization of (a) salience score matrix, (b) mask vector for each task, and (c) final mask vector, all on ViT-L/14, where x-axis denotes the layer index, y-axis in (a-b) denotes task name.

# C   Additional visualization results

## C.1   T-SNE visualization

To explore the effectiveness of ours intuitively, we also present a T-SNE visualization for Task Arithmetic and the Task Arithmetic+LwPTV on OOD tasks, where the STL10 dataset is shown in Fig.13. Compared to Task Arithmetic, our LwPTV demonstrates significantly enhanced the separability of representations across different categories. Furthermore, the intra-class clustering exhibits greater compactness when employing our approach. It indicates that introducing our proposed mask vector to prune the task vectors, LwPTV can extract discriminative features and thus enhances the generalization ability of the merged model.

## C.2   Salience score and mask vectors

To facilitate an intuitive analysis of salience score, masks associated with each task vector and their interrelationships, we also visualize them for ViT-L/14 in Fig.14 and ViT-H/14 in Fig.15.

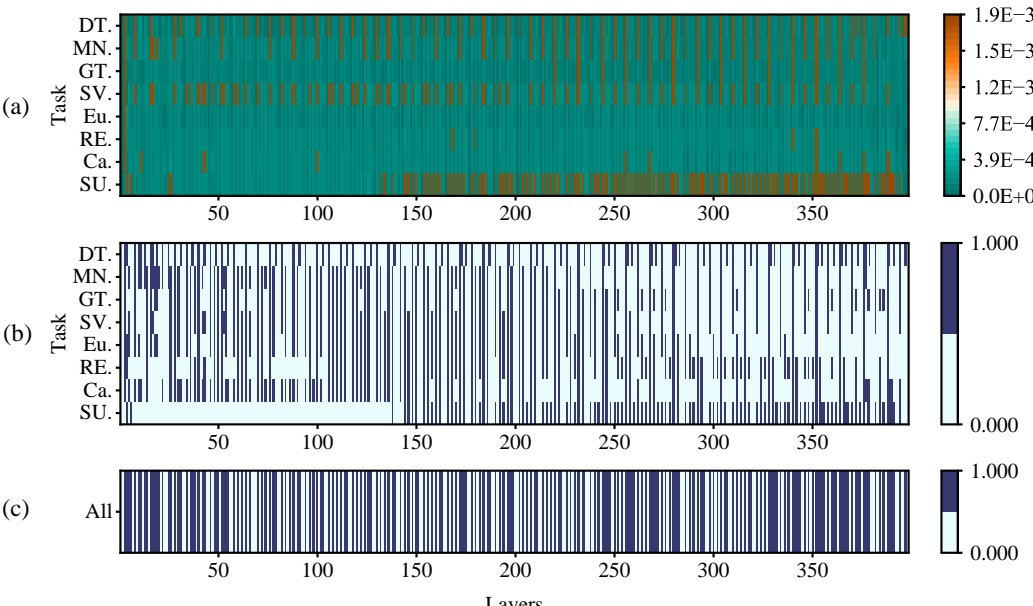

Figure 15: Visualization of (a) salience score matrix, (b) mask vector for each task, and (c) final mask vector, all on ViT-H/14, where x-axis denotes the layer index, y-axis in (a-b) denotes task name.

