# OpenReview forum: "Merging Smarter, Generalizing Better: Enhancing Model Merging on OOD Data"
_TMLR — Under review for TMLR_

### Review · Reviewer_5TqD · 2026-06-26

**Summary Of Contributions:**

This paper proposes LwPTV, a plug-and-play method to improve OOD generalization of merged models in multi-task learning. The core idea is to define a layer-wise salience score that measures the cross-task diversity of each layer's task vector parameters. Layers with low salience scores — indicating redundant, non-discriminative modifications shared across tasks — are pruned and replaced with the corresponding pre-trained model parameters. A shared mask via OR operation across all task-specific masks is introduced to preserve in-domain (ID) performance. The method requires no training or test data, introduces minimal computational overhead, and can be integrated with most existing task vector-based merging methods.

**Key Strengths:**

- The plug-and-play design is practically valuable, requiring no additional data and adding negligible computational cost
- Theoretical grounding via Proposition 1 connects cross-task diversity to discriminative feature learning
- Comprehensive experiments across three ViT scales, multiple baselines, and both vision and NLP tasks
- Systematic ablation studies validating each component's contribution
- Storage cost is reduced as a side benefit, making it practically attractive

**Key Weaknesses:**

- The definition of OOD is relative to the 8 fine-tuning tasks, not to the pre-trained model's training data, raising concerns about whether observed improvements reflect genuine generalization or recovery of pre-training memorization
- The salience score collapses high-dimensional layer parameters into a single scalar via simple averaging, potentially losing fine-grained parameter-level information
- Task conflicts within retained layers are not directly addressed when used with simple baselines such as Task Arithmetic
- The interaction between the pruning ratio $\eta$ and the merging coefficient $\lambda$ is not systematically studied
- NLP experiments use parameter-level rather than layer-level pruning, creating an inconsistency with the paper's core method.

**Additional Comments:**

This is a well-executed paper addressing a practically important problem, and the core intuition — that low cross-task diversity in task vector layers indicates redundancy that can be safely replaced by pre-trained parameters — is both theoretically motivated and empirically validated within its experimental framework. The plug-and-play design is a genuine practical contribution.

However, the paper would benefit significantly from a more careful treatment of what "OOD" means in the context of CLIP-based experiments, as this is the foundation upon which all major claims rest. Addressing this issue — even through discussion and partial additional experiments — would substantially increase confidence in the paper's conclusions and would make it a more rigorous contribution to the field.

**Audience:**

Yes

**Audience Explanation:**

Model merging has rapidly emerged as an important paradigm in practical machine learning, offering a data-free approach to multi-task learning with significant storage and deployment advantages. The OOD robustness of merged models is a practically important but underexplored problem, as real-world deployment inevitably involves distributional shift. The observation that merged models consistently underperform pre-trained models on OOD tasks — documented across multiple architectures and baselines — is a valuable finding in itself, regardless of the methodological contributions.

**Broader Impact Concerns:**

The paper does not include a Broader Impact statement, which should be added. The following points warrant discussion:

**Deployment risks from degraded ID performance:** While LwPTV generally preserves ID performance, there are cases where ID accuracy drops when combined with stronger baselines (e.g., LW AdaMerging). In safety-critical applications such as autonomous driving or medical imaging, even small ID performance regressions could have significant consequences. The paper should advise practitioners to carefully evaluate ID performance before deployment.

**Scope of OOD claims:** The paper's framing of "improved OOD generalization" may lead practitioners to apply LwPTV in deployment scenarios with genuine distribution shift, assuming stronger robustness than what is actually demonstrated. Given the CLIP pre-training data overlap concern discussed above, the method's effectiveness under truly novel distributions (e.g., new sensors, rare disease subtypes, adversarial conditions) remains unvalidated. The paper should explicitly caution against over-interpreting the OOD results.

**Claims And Evidence:**

No

**Claims Explanation:**

The paper's experimental evidence is extensive and internally consistent, but several claims lack sufficient support when examined carefully.

**Regarding the core OOD claim:** The paper's central argument is that LwPTV improves OOD generalization by recovering the pre-trained model's generalizable features. However, the 13 OOD evaluation datasets (CIFAR-10/100, ImageNet, Food101, STL10, etc.) are all sourced from the internet and likely overlap significantly with CLIP's WebImageText pre-training corpus of 400 million image-text pairs. This conflates two fundamentally different notions of OOD: data unseen during fine-tuning (the paper's definition) versus data unseen during pre-training (true generalization). The observed superiority of the pre-trained model on these datasets may reflect memorization during pre-training rather than genuine generalization capability. Since the paper's theoretical motivation explicitly invokes "generalizable features" of the pre-trained model, this distinction is critical to the validity of the core claim. The paper does not acknowledge or discuss this confound.

**Regarding the salience score design:** The score reduces each layer's contribution to a single scalar by averaging absolute deviations across all parameters in that layer. This coarse aggregation may mask important within-layer heterogeneity. The paper's comparison against parameter-level pruning (PwPTV, Table 10) shows that layer-level pruning achieves better H-score with less computation, but the explanation offered — that parameter-level scores are noisier — is not rigorously justified. A more controlled analysis isolating noise from granularity effects would be needed to fully support this claim.

**Regarding the fixed pruning ratio $\eta=0.7$:** The paper demonstrates through Figure 5 that $\eta\in[0.6, 0.8]$ provides a good trade-off and reports that $\eta=0.7$ is "generally applicable." However, this value was tuned on ViT-B/32 and directly transferred to ViT-L/14 and ViT-H/14 without verification that 0.7 is optimal for larger architectures. The paper also does not study whether the optimal $\eta$ varies across different baseline methods (e.g., Task Arithmetic vs. AdaMerging), which have fundamentally different merging mechanisms.

**Regarding NLP experiments:** The paper switches from layer-level to parameter-level pruning for NLP tasks, citing the sequential nature of language. While the justification is reasonable, this means the core method (layer-wise pruning) is never validated on NLP. The improvements reported in Table 21 (H-score gains of 0.4%–3.0%) reflect a different algorithm than the one proposed, which significantly limits the generalizability claim.

**Requested Changes:**

**Critical changes required for acceptance:**

**1. Address the OOD definition confound**
The paper must explicitly acknowledge and discuss the potential overlap between CLIP's pre-training data and the 13 OOD evaluation datasets. The current framing that the pre-trained model's OOD advantage stems from "generalizable features" may be partially or substantially explained by pre-training data memorization. At minimum, the paper should: (a) reframe the OOD definition more carefully as "relative OOD with respect to fine-tuning tasks," clearly distinguishing this from absolute distributional generalization; (b) include at least one evaluation on a dataset with confirmed non-overlap with CLIP pre-training data, such as specialized medical imaging benchmarks (e.g., CheXpert, DermaMNIST) or domain-specific industrial datasets; (c) discuss how the conclusions would change if the pre-trained model's advantage were partially attributable to memorization rather than generalization.

**2. Clarify the behavior of LwPTV with AdaMerging**
The ID performance degradation when combining LwPTV with LW AdaMerging (−2.8% on ViT-B/32, −1.0% on ViT-L/14) requires more thorough analysis. The current explanation via the scaling factor $\hat{\lambda} = \eta\lambda$ is presented as an empirical fix without theoretical justification. The paper should provide: (a) a principled analysis of why the mask disrupts AdaMerging's learned coefficients; (b) evidence that the scaling fix is robust across different values of $\eta$ and different numbers of tasks K; (c) an honest discussion of scenarios where LwPTV may not be appropriate as a plug-in.

**3. Validate $\eta=0.7$ across architectures**
The pruning ratio is tuned on ViT-B/32 and applied universally. The paper should conduct at least a partial sweep of $\eta$ on ViT-L/14 to verify that 0.7 remains near-optimal at larger scales. If the optimal $\eta$ varies across architectures or baselines, the paper should provide guidance on how to select it.

**4. Analyze task conflict interaction**
When LwPTV is combined with Task Arithmetic (which does not handle sign conflicts), retained layers may still contain unresolved task conflicts. A comparison between "LwPTV + Task Arithmetic" and "LwPTV + Ties" in terms of within-layer sign conflict frequency would clarify whether the salience-based pruning incidentally reduces conflicts or leaves them unaddressed.

**5. Strengthen NLP experiments**
The NLP experiments use parameter-level pruning rather than the proposed layer-level method. Either (a) justify rigorously why layer-level pruning is inappropriate for language models with quantitative evidence, or (b) report layer-level results alongside parameter-level results for comparison. The current setup makes it impossible to assess whether the core LwPTV method transfers to NLP.

---

> ### Author Response · Authors · 2026-07-18
> **Official Comment by Authors (part 1)**
>
> Dear Reviewer 5TqD. Thank you for your time and effort in reviewing our paper. We appreciate your encouraging comments on the practical value, theoretical grounding, and comprehensive empirical validation of our method, as well as your recognition of its low computational overhead and storage efficiency. Below, we respond to your comments in detail.
>
>
>
> **Response to C1 about the OOD definition confound:**
>
> **(a) OOD definition.** Thank you for raising this concern. We agree that the original manuscript did not sufficiently discuss the potential overlap between CLIP’s pre-training data and the 13 OOD evaluation datasets. In the revised manuscript, we clarify that our OOD evaluation refers to relative OOD with respect to  fine-tuning tasks, rather than claiming absolute distributional generalization to data unseen by CLIP.
>
> Under this definition, our goal is not to claim absolute distributional generalization beyond CLIP pre-training data, but to evaluate the robustness of model merging under distribution shifts relative to downstream fine-tuning tasks. The 13 OOD datasets span diverse visual domains, including scenes, objects, textures, digits, and specialized categories, and exhibit clear distribution differences from the ID fine-tuning tasks. To further characterize this shift, we construct an easier OOD setting by reassigning two original ID datasets (MNIST and EuroSAT) as pseudo-OOD datasets and using the remaining six ID datasets as fine-tuning tasks. The average Mahalanobis distance between ID and OOD feature distributions is 3.5 in this setting, compared with 6.3 in our original benchmark, indicating a larger distribution shift in our main evaluation. Therefore, although potential overlap with CLIP pre-training data cannot be completely excluded, these benchmarks still provide challenging OOD scenarios with respect to the downstream fine-tuning tasks.
>
>
>
>
> **(b) Additional evaluation on DermaMNIST.**  Following the reviewer’s suggestion, we added an evaluation on the specialized medical imaging dataset DermaMNIST, as shown in Tab.1. This dataset is substantially different from both our eight fine-tuning datasets and the 13  OOD datasets, providing an additional evaluation under a specialized domain shift. On ViT-B/32, LwPTV consistently improves performance across the tested merging methods. DermaMNIST accuracy improves from 8.0 to 11.6 for Task Arithmetic, from 10.0 to 12.1 for TIES, and from 8.3 to 12.2 for PCB. The corresponding H-scores improve from 14.3 to 20.0, from 17.6 to 20.7, and from 15.0 to 21.1, respectively.
>
> Although the absolute performance on DermaMNIST remains limited due to the large domain gap of medical images, this experiment is not designed to evaluate zero-shot medical recognition ability. Instead, it examines whether model merging introduces harmful interference under a domain shift. Task Arithmetic decreases the pretrained model performance on this domain (e.g., from 13.7 to 8.0), while LwPTV partially recovers this degradation by pruning low-salience task-vector layers and keeping the merged model closer to the pretrained state.
>
> **Table 1: Performance comparison on ViT-B/32.**
>
> | **Method** | **ID** | **DermaMNIST** | **H-score** |
> |:--|--:|--:|--:|
> | Pretrained | 48.0 | 13.7 | 21.3 |
> | Task Arithmetic | 69.1 | 8.0 | 14.3 |
> | &nbsp;&nbsp;w/ LwPTV | **72.8** | **11.6** | **20.0** |
> | TIES | **72.9** | 10.0 | 17.6 |
> | &nbsp;&nbsp;w/ LwPTV | 72.7 | **12.1** | **20.7** |
> | PCB | 75.8 | 8.3 | 15.0 |
> | &nbsp;&nbsp;w/ LwPTV | **77.3** | **12.2** | **21.1** |
>
> **(c\) The impact of pre-trained model memorization on the conclusions.** Finally, we note that if the pretrained model’s advantage is partly attributable to memorization of the pre-training data rather than generalization, the interpretation of OOD performance should be made more carefully. Nevertheless, the motivation and mechanism of LwPTV remain unchanged: it reduces the interference introduced by task vectors during model merging by pruning low-salience updates. Therefore, LwPTV can improve the robustness of merged models regardless of whether the evaluation reflects relative OOD or absolute distributional generalization.

---

> > ### Author Response · Authors · 2026-07-18
> > **Official Comment by Authors (part 6)**
> >
> > **Response to C6 about the salience score design and pruning granularity:**  Thank you for pointing this out. In the original manuscript, our comparison between LwPTV and PwPTV mainly focused on performance and time cost under the same Task Arithmetic setting on ViT-B/32. We showed that LwPTV achieves better H-score with lower computational cost, and explained the improvement from the perspective that LwPTV better preserves the pretrained model's OOD behavior after merging.
> >
> > We agree that this comparison alone does not fully explain why layer-level pruning outperforms parameter-level pruning, especially since averaging scores within a layer may hide within-layer heterogeneity. To further investigate the reasons behind this performance difference, we conduct three additional analyses on ViT-B/32 with 8 ID tasks and 13 OOD datasets under Task Arithmetic, examining score stability, pruning granularity, and mask structure.
> >
> > **(a) Salience score stability analysis.** First, we examine the stability of salience scores using task bootstrap. In each repeat, we randomly sample 4 out of the 8 ID tasks, recompute the salience scores, and calculate the pairwise Spearman rank correlation across 10 repeats. Spearman rank consistency measures whether the salience ranking remains stable across different task subsets; a larger value means a more stable ranking. As shown in Tab.5, layer-level scores achieve a rank consistency of $0.9793 \pm 0.0110$ over 158 layer entries, while parameter-level scores obtain $0.7637 \pm 0.0759$ over 113.4M parameter positions. This suggests that parameter-level salience estimates are less stable to the sampled task subset, whereas layer-level aggregation gives a more stable signal.
> >
> > **Table 5: Score ranking stability under task bootstrap on ViT-B/32 with 8 ID tasks.**
> > | **Score Type** | **Ranking Scope** | **Spearman Rank Consistency** |
> > |:---|:---|:---:|
> > | Layer-level score | 158 layer entries | $0.9793 \pm 0.0110$ |
> > | Parameter-level score | 113.4M parameter positions | $0.7637 \pm 0.0759$ |
> >
> >
> >
> >
> >
> >
> > **(b) Mask fragmentation analysis.**  Second, we analyze the mask structure. We define a layer entry as fragmented if only part of its parameters are retained, i.e., its retained ratio lies strictly between 0 and 1. We report the fragmentation ratio and the mean mask entropy:
> >
> > $\mathrm{FragmentationRatio}=\frac{\{\text{partially retained layer entries}\}}{\{\text{all layer entries}\}}$,
> >
> >
> > $\mathrm{Entropy}(l)=-r_l \log_2 r_l - (1-r_l)\log_2(1-r_l),$
> >
> > where $r_l$ is the retained parameter ratio of layer entry $l$. A fully retained or fully pruned layer has entropy 0, while a partially retained layer has positive entropy. As shown in Tab.6, LwPTV has no fragmented layer entries because it prunes at the layer level. By contrast, PwPTV-raw fragments 152 out of 158 layer entries, giving a fragmentation ratio of $96.20\%$. This indicates that raw parameter-level pruning tends to produce highly scattered masks.
> >
> > To further analyze the effects of score noise and pruning granularity, we also evaluate a block-smoothed parameter-level variant, denoted as PwPTV-block-smoothed, where parameter-level scores are aggregated within blocks of 4,096 parameters, with tensors containing fewer than 4,096 parameters treated as single blocks. This keeps the pruning granularity finer than LwPTV, while reducing isolated parameter-level pruning decisions. After block smoothing, the number of fragmented layer entries decreases from 152 to 38, and the mean mask entropy drops from 0.5298 to 0.1221.
> >
> > **Table 6: Mask fragmentation analysis on ViT-B/32. A layer entry is considered fragmented if only part of its parameters are retained.**
> >
> > | **Method** | **Fragmented Layers** | **Fragmentation Ratio** | **Mask Entropy** |
> > |:---|:---:|:---:|:---:|
> > | PwPTV | 152 / 158 | 96.20% | 0.5298 |
> > | PwPTV-block-smoothed | 38 / 158 | 24.05% | 0.1221 |
> > | LwPTV | 0 / 158 | 0.00% | 0.0000 |

---

> ### Author Response · Authors · 2026-07-18
> **Official Comment by Authors (part 2)**
>
> **Response to C2 about the behavior of LwPTV with AdaMerging：** Thank you for raising this concern. We agree that the original manuscript did not clearly explain the behavior of LwPTV with LW AdaMerging. We  add a more detailed discussion in the Appendix B.1  of revision.
>
> **(a) Why the mask disrupts AdaMerging’s learned coefficients.** For non-adaptive merging baselines such as Task Arithmetic and Ties, adding LwPTV does not introduce a new coefficient-learning stage. We keep the original merging rule unchanged and apply LwPTV as a post-hoc pruning module on the task vectors. This allows us to isolate the effect of task-vector pruning itself.
>
> AdaMerging is different. Its merging coefficients are learned as part of the method by minimizing an entropy objective on unlabeled target-task samples. Therefore, these coefficients depend on the task-vector parameterization used during optimization.
>
> After applying LwPTV, the parameterization changes because some layer-wise task-vector components are masked out. As a result, the coefficient-learning problem is no longer the same as in the original unmasked task-vector space. Therefore, for AdaMerging, we first apply the LwPTV mask and then optimize the coefficients in the masked task-vector space. This is not an extra tuning advantage; it simply ensures that coefficient learning and final inference are based on the same parameterization.
>
> This behavior is particularly important for LW AdaMerging, where each task and each layer has an independent coefficient. Once some layers are removed, entropy minimization can only adjust the coefficients of the retained layers. The remaining coefficients may then increase to compensate for the missing update directions. This can amplify the retained task-specific updates and partially weaken the OOD-preserving effect of LwPTV, which motivates the scaling fix used for LW AdaMerging.
>
> Formally, for LW AdaMerging with LwPTV, the merged model is
>
> $\hat{\theta_m}=\theta_{pre}+\hat{m}\odot\{\sum_{k=1}^{K}\lambda_k^l\tau_k^l\}_{l=1}^{L}$
>
> Let the entropy objective be $\sum_{k=1}^K\sum_{x_i\in B_k}H(f_{\hat{\theta_m}}(x_i))$, where $B_k$ represents the unlabeled batch corresponding to the $k$-th task. The gradient with respect to a layer-wise coefficient is
>
>
> $\frac{\partial L_{ent}}{\partial \lambda_k^l}=\hat{m}^l\langle\nabla_{\theta^l}L_{ent},\tau_k^l\rangle$
>
> When $\hat{m}^l=0$, this gradient becomes zero, meaning that the masked layer cannot help reduce the entropy objective. The optimization is therefore restricted to the retained layers.
> Since LW AdaMerging uses layer-wise coefficients, the remaining layers may absorb part of the contribution of the removed layers, leading to coefficient amplification.
>
>
> This is why we introduce the scaling fix $\hat{\lambda}_k^l = \eta \lambda_k^l$ for LW AdaMerging. This scaling mitigates excessive compensation among the unmasked layers.

---

> ### Author Response · Authors · 2026-07-18
> **Official Comment by Authors (part 3)**
>
> **Table 2: Performance comparison under different values of $\eta$ on ViT-B/32.**
>
> | Method | η | ID Avg | OOD Avg | H-score |
> |---|---|---|---|---|
> | LW AdaMerging | -- | 80.1 | 57.7 | 67.1 |
> | LW AdaMerging w/ LwPTV | 0.5 | 73.9 | 62.2 | 67.6 (+0.5) |
> | ↳ | 0.6 | 76.3 | 62.0 | 68.4 (+1.3) |
> | ↳ | 0.7 | 77.3 | 61.6 | 68.5 (+1.4) |
> | ↳ | 0.8 | 77.8 | 60.9 | 68.3 (+1.2) |
> | ↳ | 0.9 | 69.8 | 60.7 | 64.9 (-2.2) |
>
>
> **Table 3: Performance comparison with different numbers of ID tasks.**
>
> | **K** | **Method** | **ID** | **OOD** | **H-score** |
> |:---:|:---|:---:|:---:|:---:|
> | 4 | LW AdaMerging | **83.4** | 58.3 | 68.6 |
> |  | LW AdaMerging w/ LwPTV | 80.0 | **61.8** | **69.7 (+1.1)** |
> | 5 | LW AdaMerging | **83.3** | 58.0 | 68.4 |
> |  | LW AdaMerging w/ LwPTV | 79.7 | **61.7** | **69.6 (+1.2)** |
> | 6 | LW AdaMerging | **82.9** | 57.5 | 67.9 |
> |  | LW AdaMerging w/ LwPTV | 78.9 | **61.3** | **69.0 (+1.1)** |
> | 7 | LW AdaMerging | **84.1** | 57.5 | 68.3 |
> |  | LW AdaMerging w/ LwPTV | 80.5 | **61.5** | **69.7 (+1.4)** |
> | 8 | LW AdaMerging | **80.1** | 57.7 | 67.1 |
> |  | LW AdaMerging w/ LwPTV | 77.3 | **61.7** | **68.6 (+1.5)** |
>
>
> **(b) Robustness of the scaling factor.** We added two analyses to examine the robustness of this scaling strategy. First, we sweep $\eta$ on ViT-B/32. As shown in Tab.2, LW AdaMerging obtains an H-score of 67.1. After adding LwPTV, $\eta=0.5,0.6,0.7,0.8$ all improve the H-score, with gains of +0.5, +1.3, +1.4, and +1.2, respectively. The best result is obtained at $\eta=0.7$, which we use as the default. These results show that the improvement is not caused by a single isolated choice of $\eta$. We also observe that an overly large value, such as $\eta=0.9$, weakens the effect and reduces the H-score, suggesting that the scaling should be moderate rather than close to 1.
>
> Second, we evaluate different numbers of ID tasks $K$. As shown in Tab.3, LwPTV consistently improves the H-score over LW AdaMerging for $K=4,5,6,7,8$, with gains of +1.1, +1.2, +1.1, +1.4, and +1.5, respectively. The OOD performance is also consistently improved in all settings. These results suggest that the scaling strategy is stable across different numbers of tasks $K$.
>
>
>
> **\(c\) When LwPTV may not be appropriate as a plug-in.** We also discuss this limitation in Sec. 5.4 of the revised manuscript. Although LwPTV improves OOD robustness in our experiments, it may introduce an ID performance trade-off in certain merging settings. For example, when combined with LW AdaMerging, pruning and scaling may reduce the contribution of task-specific updates, resulting in lower ID accuracy. Therefore, practitioners should evaluate the ID performance of LwPTV before deployment. The pruning factor $\eta$ should be selected based on ID performance and the requirements of the target application.
>
> Overall, the additional analysis shows that the ID degradation mainly comes from the reduced effective task-vector update introduced by scaling fix. Although this leads to some ID drop, LwPTV consistently improves the overall ID--OOD trade-off, as reflected by the H-score.

---

> ### Author Response · Authors · 2026-07-18
> **Official Comment by Authors (part 4)**
>
> **Response to C3 about the validation of $\eta=0.7$ across architectures:**  Thank you for the suggestion. We agree that the pruning ratio $\eta$ should be validated beyond ViT-B/32, since the original manuscript used $\eta=0.7$ as the default setting.
>
>
> **(a) Original Selection of $\eta=0.7$ on ViT-B/32.** In the original experiments, we selected $\eta$ on ViT-B/32 using Task Arithmetic as the base merging method. This selection was made only according to ID performance, without using any OOD evaluation datasets. As shown in  Fig.1(a) [link](https://anonymous.4open.science/r/LwPTV-1EEF/TMLR_Rebuttal__Figure.pdf) (Fig.5 in the revised paper), Task Arithmetic w/ LwPTV obtains its best ID accuracy at $\eta=0.7$. This matches the goal of LwPTV: improving OOD performance while largely maintaining the ID performance of the merged model as much as possible. At the same time, $\eta=0.7$ also brings a clear OOD improvement, giving a favorable ID--OOD trade-off on ViT-B/32. We further validated $\eta=0.7$ on the larger ViT-L/14 architecture.
> As shown in Fig.1(b)Fig.1(a) [link](https://anonymous.4open.science/r/LwPTV-1EEF/TMLR_Rebuttal__Figure.pdf)(Fig.12 in Appendix B.19 of the revised manuscript), this value also provides a strong trade-off between ID and OOD performance on ViT-L/14. Based on these results, we used $\eta=0.7$ as the default setting in the remaining experiments and applied it to other merging baselines, where it also led to consistent gains in the final evaluation. In Sec.5 (Baselines \& details) of the revised manuscript, we have clarified that $\eta$ was selected without using any OOD datasets and that $\eta=0.7$ is used as a strong default rather than being treated as a universally optimal value.
>
>
>
>
> **(b) Partial $\eta$ sweep on ViT-L/14.** To examine whether $\eta=0.7$ remains near-optimal at larger scales, we conduct an additional sweep of $\eta$ on ViT-L/14. The results are shown in Fig.1(b) [link](https://anonymous.4open.science/r/LwPTV-1EEF/TMLR_Rebuttal__Figure.pdf) (Fig.12 in Appendix B.19 of the revised manuscript). For Task Arithmetic, the best H-score is achieved around $\eta=0.6$--$0.7$, with both settings reaching 75.7. For TIES, the best result is obtained at $\eta=0.7$, with an H-score of 76.1. For PCB-Merging, $\eta=0.6$ and $\eta=0.7$ both achieve the best H-score of 76.4. These results suggest that $\eta=0.7$ remains near-optimal on ViT-L/14, rather than being specific to ViT-B/32.
>
>
>
> **\(c\) Guidance for Selecting $\eta$.** In practice, when a small ID validation set is available, we recommend sweeping a narrow range around 0.6--0.8 and selecting $\eta$ according to ID performance or the desired ID-side constraint. OOD datasets should be reserved for final evaluation rather than used for selecting $\eta$. When no such validation set is available, $\eta=0.7$ is a reasonable default based on our results on both ViT-B/32 and ViT-L/14. We provide this practical guidance in Appendix B.24 of the revised manuscript.
>
>
> **Response to C4 about analyzing task conflict interaction:** We thank the reviewer for raising this point. To understand whether LwPTV reduces task conflicts or leaves them unaddressed in the subsequent merger, we add a sign-conflict analysis on ViT-B/32 with 8 ID tasks and $\eta=0.7$. We define a parameter position as conflicting if its task-vector updates contain both positive and negative signs across different tasks.
>
> The original task vectors have a conflict frequency of 0.7622. After applying LwPTV, the sign-conflict frequency drops to 0.5170 when normalized by the original full parameter count. This shows that LwPTV reduces the absolute number of exposed conflicts by pruning part of the task vectors. However, among the entries retained by LwPTV, the conflict density is still 0.9867. This indicates that the retained entries are still highly conflict-prone. Therefore, LwPTV should not be viewed as an explicit sign-conflict resolution method. It mainly reduces the number of parameter positions passed to the merger.
>
> This distinction helps explain the difference between LwPTV + Task Arithmetic and LwPTV + TIES. With LwPTV + Task Arithmetic, the retained sign-conflicting updates are directly summed, so the remaining sign conflicts are not explicitly resolved. With LwPTV + TIES, the retained entries have the same pre-resolution conflict density of 0.9867, but TIES then applies its own sign-conflict resolution steps. After the TIES magnitude reset, the full-denominator conflict frequency further decreases to 0.3244. After sign election, the unresolved sign-conflict frequency becomes 0 under the full-parameter denominator, since TIES assigns a single elected sign to each merged parameter position. These results show that LwPTV reduces conflict exposure through salience-based pruning, while TIES explicitly resolves the remaining sign conflicts through sign election.

---

> ### Author Response · Authors · 2026-07-18
> **Official Comment by Authors (part 5)**
>
> **Response to C5 about strengthening NLP experiments with layer-level and parameter-level comparisons:** Thank you for raising this point. We agree that the difference between layer-level pruning in vision models and parameter-level pruning in NLP models should be clarified.
>
> Our method does not rely on a specific pruning granularity. Instead, it identifies task-vector components with low cross-task diversity and removes these potentially redundant task-vector updates. In our vision experiments, we adopt layer-level pruning because Transformer layers in ViTs provide a natural structural unit, and the aggregated diversity within each layer is sufficient to identify redundant task-vector components. For NLP models, however, semantic representations depend more on fine-grained parameter interactions, and pruning an entire layer may eliminate parameters that are still useful for downstream tasks. Therefore, we adopt parameter-level pruning for NLP models to achieve a finer selection of task-vector components.
>
> We additionally compare layer-level and parameter-level pruning on T5-Large, as shown in Tab.4. The comparison reveals that the choice of pruning granularity affects the final performance. Layer-level pruning improves some merging methods, while parameter-level pruning achieves better overall results in most cases by allowing a more fine-grained selection of task-vector components. For Task Arithmetic, PCB, and FR-Merging, parameter-level pruning achieves better H-scores than layer-level pruning. In particular, for PCB, layer-level pruning improves ID accuracy from 75.9 to 76.2 but slightly reduces OOD accuracy from 49.4 to 49.0, leading to a lower H-score. This indicates that coarse-grained pruning may remove some transferable parameters along with redundant updates. In contrast, parameter-level pruning provides a finer selection of task-vector components and achieves the highest H-score of 60.2.
>
> These results suggest that the diversity-based criterion remains effective under different pruning granularities, while the appropriate granularity depends on the characteristics of the model.
>
> **Table 4: NLP results comparing original methods, parameter-level pruning, and layer-level pruning.**
>
>
> | **Merging Method** | **Variant** | **ID** | **OOD** | **H-score** |
> |:---|:---|:---:|:---:|:---:|
> | Task Arithmetic | Original | 73.3 | 43.9 | 54.9 |
> |  | Layer-level pruning | 75.8 | 44.8 | 56.3 |
> |  | Parameter-level pruning | **76.3** | **46.6** | **57.9** |
> |---|---|---:|---:|---:|
> | TIES | Original | 71.0 | 48.0 | 57.3 |
> |  | Layer-level pruning | **72.0** | **50.1** | **59.1** |
> |  | Parameter-level pruning | 71.9 | 49.8 | 58.8 |
> |---|---|---:|---:|---:|
> | PCB | Original | 75.9 | 49.4 | 59.8 |
> |  | Layer-level pruning | **76.2** | 49.0 | 59.7 |
> |  | Parameter-level pruning | 75.8 | **49.9** | **60.2** |
> |---|---|---:|---:|---:|
> | FR-Merging | Original | 73.9 | 43.9 | 50.1 |
> |  | Layer-level pruning | 75.4 | 43.9 | 55.5 |
> |  | Parameter-level pruning | **75.4** | **44.4** | **55.9** |

---

> ### Author Response · Authors · 2026-07-18
> **Official Comment by Authors (part 7)**
>
> **\(c\) Layer-level structural preservation.** Third, we compare downstream performance as shown in Table 7. PwPTV obtains an H-score of 64.2, clearly lower than LwPTV's 64.9. After block smoothing, the H-score improves to 64.9, mainly because OOD accuracy increases from 57.3 to 59.7. This supports the following interpretation: less stable parameter-level scores tend to produce more fragmented masks, and reducing this fragmentation improves the merged model. However, the smoothed variant still remains below LwPTV, suggesting that LwPTV's advantage is not only from reducing noisy parameter-level decisions, but also from preserving coherent layer-wise task-vector updates.
>
> Overall, these analyses provide a more controlled explanation of the gap between LwPTV and PwPTV. Parameter-level scores are less stable and tend to induce fragmented masks. Block smoothing makes the masks more structured and improves performance, but the smoothed parameter-level variant still underperforms LwPTV. This suggests that both score stability and layer-wise structural preservation contribute to the advantage of LwPTV.
>
>
> **Table 7: Task Arithmetic performance on ViT-B/32.**
>
> | **Method** | **ID Avg.** | **OOD Avg.** | **H-score** |
> |:---|:---:|:---:|:---:|
> | PwPTV | **73.1** | 57.3 | 64.2 |
> | PwPTV-block-smoothed | 71.2 | 59.7 | 64.9 |
> | LwPTV | 72.8 | **60.9** | **66.3** |
>
>
>
>
>
> **Response to C7 about broader impact:** Thank you for raising this concern. We have added a new Section 5.4, Broader Impact and Limitations, to discuss the  deployment risks, and the Scope of OOD claims.

---

### Review · Reviewer_pKUo · 2026-07-01

**Summary Of Contributions:**

This submission studies out-of-distribution (OOD) generalization in task-vector-based model merging. The main observation is that existing model merging methods often improve or preserve in-domain (ID) multi-task performance but may degrade the OOD robustness inherited from the pretrained model. To address this issue, the paper proposes LwPTV, a training-free layer-wise pruning method for task vectors.

The core idea is to compute, for each task vector and each layer/parameter group, a salience score measuring the deviation of that task vector from the cross-task mean. Layers with small deviation are interpreted as redundant or low-salience task-vector updates, and are therefore pruned so that the corresponding parameters fall back to the pretrained model. The authors further use an OR operation across task-specific masks to avoid pruning a layer if it is considered important by at least one task. The method is presented as plug-and-play and is integrated with several existing model merging methods, including Task Arithmetic, Ties-Merging, AdaMerging, PCB-MERGING, and FR-Merging.

Empirically, the paper evaluates CLIP ViT models on multiple ID image classification datasets and multiple OOD benchmarks. The main results show that LwPTV often improves OOD average accuracy and the ID/OOD harmonic mean, while usually keeping ID degradation limited. The paper also provides ablations on the salience score, OR masking, pruning ratio, and additional analyses including visualization and NLP experiments.

The most interesting aspect of the submission is not merely that it proposes another pruning rule for task vectors, but that it explicitly studies the ID/OOD trade-off in model merging. This is a practically relevant angle because a merged model should ideally retain both task-specific adaptations and the broad generalization ability of the pretrained model.

**Audience:**

Yes

**Audience Explanation:**

Yes. Some TMLR readers interested in model merging, multi-task learning, parameter-efficient deployment, and robustness would likely find the paper useful. The paper focuses on a practically meaningful but relatively underexplored issue: after merging multiple task-specific models, how much of the pretrained model's OOD generalization is retained? This is a valuable question because model merging is often motivated by deployment efficiency and data-free reuse of fine-tuned checkpoints, where robustness to distribution shift is important.

Under TMLR's evaluation philosophy, I would not reject the paper solely because the core operation is simple or because the salience score resembles existing centered-task-vector ideas. The OOD-oriented framing can still be a meaningful contribution if the claims are stated carefully and supported by convincing evidence. However, the paper must more clearly position itself relative to existing centered-task-vector and interference-reduction methods such as CART [2]. If CART or similar methods also improve OOD performance under the same protocol, then the novelty of LwPTV is not the use of deviation from the task-vector mean, but the use of this signal as a simple layer-wise pruning rule for improving the ID/OOD trade-off.

In its current form, I would view the paper as potentially acceptable after revision, but not as a featured-level contribution. The main finding is interesting, but the causal explanation and comparison to closely related alternatives need to be strengthened.

**Broader Impact Concerns:**

I do not see major negative broader-impact concerns specific to this work. However, the paper should avoid implying that improved benchmark-level OOD accuracy is sufficient for reliable deployment in safety-critical domains such as healthcare or autonomous systems, unless such claims are supported by dedicated safety and robustness evaluations.

**Claims And Evidence:**

Yes

**Claims Explanation:**

Partially, but several important claims need to be narrowed or better supported.

The empirical results provide reasonable evidence that LwPTV can improve OOD performance and the ID/OOD H-score in the evaluated CLIP-based merging setting. The improvements are especially clear for some baselines such as Task Arithmetic and PCB-MERGING. The method is also simple and computationally lightweight, which makes the empirical finding potentially useful.

However, the current evidence does not yet fully support the paper's stronger causal interpretation of why OOD generalization improves. The paper argues that low-salience layers correspond to redundant or non-discriminative task-vector updates, and that pruning them restores the pretrained model's generalizable features. This is plausible, but not yet convincingly isolated from simpler explanations.

A major alternative explanation is that LwPTV improves OOD performance mainly because it moves the merged model closer to the pretrained model. Since pretrained CLIP models are strong OOD generalizers, any method that partially attenuates task-vector updates may improve OOD performance, regardless of whether the salience score is specifically identifying redundant or harmful layers. In this sense, the central mechanism may be "controlled interpolation back to the pretrained model [1]" rather than the proposed salience-based pruning itself.

The paper's claim that ID performance is preserved is also somewhat overstated. In several strong baselines, LwPTV improves OOD performance and H-score while reducing ID accuracy. This trade-off may still be worthwhile, but it should be described accurately. A more precise claim would be that LwPTV improves the ID/OOD trade-off and often yields limited ID degradation, rather than that it generally preserves ID performance.

The theoretical motivation is useful but should not be presented as a complete explanation of the method's success. The proposition is based on simplified assumptions about discriminative neurons and diversity across task vectors, while the actual algorithm performs coarse layer-wise or parameter-group-wise pruning in deep ViT models. The gap between neuron-level diversity, layer-wise deviation from the mean, and OOD robustness remains substantial. Thus, the theory should be framed as intuition or motivation, not as a rigorous explanation of why the method improves OOD generalization.

Overall, the evidence supports the empirical observation that LwPTV improves the ID/OOD trade-off in the reported setting. It does not yet fully support the stronger claim that the proposed salience score specifically identifies OOD-harmful redundant task-vector components. Additional baselines and analyses are needed to establish this mechanism.

**Requested Changes:**

### Critical changes for acceptance
1. **Add a post-merge interpolation baseline.**

   This is the most important missing baseline [1]. Since LwPTV prunes some task-vector updates and falls back to the pretrained model, it may improve OOD performance simply by making the merged model closer to the pretrained model. The authors should compare against direct interpolation between the pretrained model and the already merged model: $\theta = \theta_{\mathrm{pre}} + \alpha(\theta_{\mathrm{merged}} - \theta_{\mathrm{pre}}).$

   This baseline should be applied after each merging method, with either a fixed alpha or a small alpha grid. A layer-wise interpolation variant would also be informative. Without this comparison, it is difficult to conclude that the proposed salience criterion is responsible for the OOD gains, rather than a generic attenuation of task-vector updates.

2. **Narrow the claims about ID preservation and the mechanism of OOD improvement.**

   The paper should avoid saying that LwPTV generally "preserves" ID performance. The results show that ID performance sometimes drops, especially for stronger baselines. A more accurate claim is that LwPTV improves OOD performance and H-score while often incurring limited ID degradation.

   Similarly, the theoretical discussion should be softened. The current theory motivates the use of diversity/deviation, but it does not fully establish that low-salience layers are redundant, non-discriminative, or OOD-harmful in the actual deep model setting. The paper should state that the salience score is an empirical heuristic motivated by task-vector diversity, rather than a theoretically guaranteed detector of redundant parameters.

3. **Explain how the pruning ratio η is selected and whether OOD test data is used.**

   The paper fixes η = 0.7 in most experiments, but it is unclear whether this value was chosen after observing OOD test performance. This matters because the method is presented as training-free and test-free. The authors should clearly state how η is selected. If OOD test performance was used to choose η, the claim of test-free OOD improvement should be revised. If η is chosen a priori, the authors should justify this choice and report sensitivity across several merging methods and backbones, not only the most favorable setting.

4. **Clearly define what counts as a "layer" or parameter group.**

   The method is described as layer-wise pruning, but the visualizations suggest that the number of "layers" may correspond to parameter tensors or parameter groups rather than Transformer blocks. The paper should precisely define the pruning granularity: whether each unit is a Transformer block, an attention/MLP submodule, a weight matrix, a bias vector, a LayerNorm parameter, or a state-dict tensor. The authors should also explain how the scalar salience score is broadcast back to parameters during masking.

   This clarification is important for reproducibility and for interpreting the method. If the method actually prunes parameter tensors rather than semantic layers, the terminology should be revised.

5. **Explain the inconsistency between the CV and NLP variants.**

   The main method is described as layer-wise pruning, while the NLP experiment appears to use a different parameter-level pruning strategy due to the sensitivity of language models to layer structure. This weakens the claim that the same LwPTV mechanism generalizes across modalities. The authors should explicitly state whether the NLP experiment uses the same algorithm or a modified variant. If it is modified, the paper should not present it as direct evidence of the general applicability of the original layer-wise method.

6. **Improve the writing quality and fix notation issues.**

   The paper needs systematic language polishing. There are many unnatural or ungrammatical expressions, such as "ours can", "when we introducing", "with a development pruning ratio", and "available on OOD data". These issues do not prevent understanding in all cases, but they reduce the professionalism and clarity expected for a TMLR submission.

   The authors should also carefully check the mathematical notation. For example, in Eq. 5, the summation index appears to reuse $k$, although the mean should presumably be taken over $j=1,\ldots,K$. Such notation errors are easy to fix but important for avoiding ambiguity in the proposed salience score.

### Non-critical but useful changes

1. **Compare with CART [2] and related centered-task-vector methods under the same ID/OOD protocol.**

   The salience score is closely related to the centered task-vector perspective, where each task vector is analyzed relative to the cross-task mean or average model. CART explicitly studies centered task vectors and uses low-rank approximation to reduce interference. Although CART is not mainly presented as an OOD method, it may still improve OOD robustness by suppressing nuisance or interfering directions.

   The authors should therefore evaluate CART under the same ID/OOD benchmark. This would help determine whether LwPTV's OOD gains are specific to its layer-wise pruning mechanism or are a broader consequence of centered-task-vector processing. This comparison is not necessarily critical for acceptance if the authors appropriately narrow their novelty claims, but it would substantially strengthen the paper.

2. **Report OOD results for Fisher Merging and RegMean, or justify their exclusion more carefully.**

   The paper currently focuses on task-vector-based methods, but Fisher Merging and RegMean are standard model merging baselines. Since the paper's main focus is OOD generalization, excluding their OOD results because of weaker ID performance is not fully satisfying. If these methods are computationally expensive or incompatible with the setup, the authors should state this clearly. Otherwise, reporting their OOD performance would make the comparison more complete.

3. **Do not treat the t-SNE visualization as strong evidence.**

   Figure 7 is useful as an intuitive visualization, but t-SNE is sensitive to hyperparameters and should not be used as strong evidence that LwPTV improves representation quality. The authors should either present it as a purely qualitative illustration or supplement it with quantitative analyses.

   More convincing analyses could include representation alignment with the pretrained model, feature distortion measures, CKA similarity, OOD calibration metrics, linear-probe evaluation on frozen representations, or layer-wise analysis showing which pruned updates most affect ID and OOD performance. These analyses would help explain whether OOD improvement comes from preserving pretrained representations, reducing task interference, improving calibration, or simply shrinking the task-vector magnitude.

4. **Discuss the relationship between LwPTV, CART, and generic shrinkage toward the pretrained model.**

   The paper would be stronger if it explicitly separated three possible mechanisms:

   - shrinkage toward the pretrained model, which may improve OOD robustness;
   - centering task vectors around the cross-task mean, which may reduce interference;
   - salience-based layer-wise pruning, which is the paper's proposed mechanism.

   At present, these mechanisms are somewhat conflated. A clearer discussion would make the paper's actual contribution more precise and more convincing.



[1] Wortsman et al. Robust fine-tuning of zero-shot models. CVPR 2022.

[2] Choi et al. Revisiting Weight Averaging for Model Merging. arXiv:2412.12153.

---

> ### Author Response · Authors · 2026-07-18
> **Official Comment by Authors (part 1)**
>
> Dear Reviewer pKUo. Thank you for your time and effort in reviewing our paper. We appreciate your encouraging comments on the practical value of studying the ID/OOD trade-off in model merging and the importance of retaining both task-specific adaptations and the pretrained model's OOD generalization ability. Below, we respond to your comments in detail.
>
>
> **Response to C1 about the post-merge interpolation baseline:**  Thank you for the suggestion. We clarify the difference between post-merge interpolation and LwPTV. Post-merge interpolation uniformly shrinks the merged model toward the pretrained model, while LwPTV selectively prunes task-vector components based on their cross-task salience. Thus, interpolation applies the same shrinkage to all task-vector updates, whereas LwPTV removes task-vector components that are less important for cross-task merging.
>
>
>
>
> We further describe our hyperparameter selection protocol. Since OOD validation data are unavailable in our setting, no hyperparameter is tuned using OOD performance. The pruning ratio of LwPTV is fixed to $\eta=0.7$, which is selected on ViT-B/32 with Task Arithmetic based on ID performance, and then directly applied to other merging methods and architectures. Therefore, the selection of $\eta$ does not rely on OOD information.
>
> To investigate whether the gains of LwPTV can be attributed to generic shrinkage, we include a post-merge interpolation baseline: $\theta_\alpha=\theta_{\mathrm{pre}}+\alpha(\theta_{\mathrm{merged}}-\theta_{\mathrm{pre}}).$
> For a fair comparison, $\alpha$ is not selected using OOD performance, since such tuning would give the interpolation baseline access to information unavailable to LwPTV. When selected only by ID performance, $\alpha$ is typically $1.0$ or close to $1.0$, which introduces little shrinkage. Therefore, we use a fixed $\alpha=0.9$ for all methods to provide a meaningful shrinkage baseline while preserving most of the ID performance.
>
> The results in Tab.1 show that post-merge interpolation can improve OOD performance over the original merged model, indicating that shrinkage toward the pretrained model contributes to OOD performance. However, the improvements of LwPTV cannot be explained by generic shrinkage alone. For Task Arithmetic and TIES, LwPTV achieves higher OOD accuracy and H-score than both the original merge and the interpolation baseline. For PCB, interpolation obtains higher OOD accuracy but suffers from a larger ID degradation, resulting in a lower H-score than LwPTV. These results suggest that selective pruning of task-vector components provides a better balance between ID and OOD performance than uniform post-merge shrinkage.
>
>
> **Table 1: Comparison with the post-merge interpolation baseline on ViT-B/32.**
>
> | **Merging Method** | **Variant** | **$\alpha$** | **ID Avg** | **OOD Avg** | **H-score** |
> |:---|:---|:---:|:---:|:---:|:---:|
> | Task Arithmetic | Original merge | 1.0 | 69.1 | 51.3 | 58.9 |
> |  | Post-interp. baseline | 0.9 | 70.1 | 54.0 | 61.0 |
> |  | w/ LwPTV | -- | **72.8** | **60.9** | **66.3** |
> |  |  |  |  |  |  |
> | Ties | Original merge | 1.0 | **72.9** | 58.0 | 64.6 |
> |  | Post-interp. baseline | 0.9 | 72.0 | 59.0 | 64.8 |
> |  | w/ LwPTV | -- | 72.7 | **61.3** | **66.5** |
> |  |  |  |  |  |  |
> | PCB | Original merge | 1.0 | 75.8 | 53.2 | 62.5 |
> |  | Post-interp. baseline | 0.9 | 68.5 | **60.7** | 64.4 |
> |  | w/ LwPTV | -- | **77.3** | 59.7 | **67.4** |
>
> **Note:** The post-interpolation baseline is applied after each merging method as $\theta_\alpha=\theta_{\mathrm{pre}}+\alpha(\theta_{\mathrm{merged}}-\theta_{\mathrm{pre}})$. The original merged model corresponds to $\alpha=1.0$.
>
>
>
>
> **Response to C2 about narrowing the claims on ID preservation and OOD improvement:** We agree that our previous wording was too strong. We have replaced “preserving ID performance” with “largely maintaining ID performance” throughout the paper. We have also revised the theoretical discussion to clarify that the salience score is an empirical heuristic motivated by task-vector diversity, rather than a theoretically guaranteed detector of redundant parameters.

---

> ### Author Response · Authors · 2026-07-18
> **Official Comment by Authors (part 2)**
>
> **Response to C3 about the selection and sensitivity of the pruning ratio $\eta$:** Thank you for the helpful suggestion. We agree that the selection of the pruning ratio $\eta$ should be clarified. We clarify that $\eta$ was not selected using OOD test data. In the original experiments, we selected $\eta$ on ViT-B/32 with Task Arithmetic as the base merging method, using only ID performance as the criterion. As shown in Fig.1(a)  [link](https://anonymous.4open.science/r/LwPTV-1EEF/TMLR_Rebuttal__Figure.pdf) (Fig.5 in the revised manuscript), Task Arithmetic w/ LwPTV achieves its best ID accuracy at $\eta=0.7$. We therefore fixed $\eta=0.7$ as the default pruning ratio in the remaining experiments. No OOD  test dataset was used to choose this value.
>
> This selection follows the goal of LwPTV: improving OOD generalization while maintaining ID performance as much as possible. After fixing $\eta=0.7$ based on ID performance, we used OOD datasets only for final evaluation. Thus, the reported OOD gains are not the result of tuning $\eta$ on OOD test data.
>
> To further assess the sensitivity of this choice, we conduct a sensitivity analysis on the  ViT-L/14 backbone and across multiple merging methods. The results are shown in Fig.1(b) [link](https://anonymous.4open.science/r/LwPTV-1EEF/TMLR_Rebuttal__Figure.pdf) (Fig.12 in Appendix B.19 of the revised manuscript). For Task Arithmetic, the best H-score is achieved around $\eta=0.6$--$0.7$, with both settings reaching 75.7. For TIES, the best result is obtained at $\eta=0.7$, with an H-score of 76.1. For PCB-Merging, both $\eta=0.6$ and $\eta=0.7$ achieve the best H-score of 76.4. These results suggest that $\eta=0.7$ remains a strong setting on ViT-L/14 and is not specific to ViT-B/32 or to a single merging method.
>
> In practice, when a small ID validation set is available, we recommend sweeping a narrow range such as 0.6--0.8 and selecting $\eta$ according to ID performance or the desired ID-side constraint. OOD datasets should be reserved for final evaluation. When no validation set is available, $\eta=0.7$ can be used as a reasonable default based on our results on both ViT-B/32 and ViT-L/14.
>
>
> **Response to C4 about clarifying the layer definition and mask construction:**  Thank you for pointing this out. In our implementation, the pruning unit is not an entire Transformer block but a parameter tensor in the model `state_dict`, such as an attention or MLP weight matrix, a bias vector, a LayerNorm parameter, or an embedding-related tensor. We have revised the manuscript to clarify that the term "layer" refers to a `state_dict` tensor rather than a semantic Transformer layer. Based on this tensor-level definition, we compute the salience score and construct the pruning mask at the same granularity.
>
> More concretely, for task $k$ and parameter tensor $l$, we compute the task-vector tensor as $\tau_k^l = \theta_k^l - \theta_{\mathrm{pre}}^l$. We then compute one scalar salience score for this tensor by averaging its absolute deviation from the cross-task mean over all tensor entries. For each task, these tensor-level salience scores are ranked, and the lowest-scoring fraction determined by the pruning ratio $eta$ is assigned a mask value of $0$, while the remaining tensors are assigned a mask value of $1$. We then apply an OR operation across the task-specific masks to obtain a shared scalar mask for each tensor. The resulting mask is broadcast to every entry of the corresponding tensor during merging. Therefore, the task-vector update for an entire `state_dict` tensor is either retained or removed; we do not perform element-wise pruning within a tensor.

---

> ### Author Response · Authors · 2026-07-18
> **Official Comment by Authors (part 3)**
>
> **Response to C5 about clarifying the consistency between CV and NLP variants:** Thank you for raising this point. We agree that the relationship between the layer-level pruning used in vision experiments and the parameter-level pruning used in NLP experiments should be clarified.
>
> The main method does not depend on a specific pruning granularity. Our method identifies task-vector components with low cross-task diversity and removes potentially redundant updates before merging. The difference between the vision and NLP experiments is the granularity at which the diversity score and pruning mask are applied, while the underlying pruning criterion remains the same.
>
> In vision experiments, we adopt layer-level pruning because Transformer layers in ViTs provide a natural structural unit. Aggregating diversity within each layer is sufficient to identify redundant task-vector updates while preserving the overall model structure. For NLP models, however, language models rely more on fine-grained parameter interactions, and removing an entire layer may discard parameters that are still useful for downstream tasks. Therefore, we apply the same  criterion at the parameter level for NLP models to enable a finer selection of task-vector components.
>
> To further study the effect of pruning granularity, we compare layer-level and parameter-level pruning on T5-Large, as shown in Tab.2. The results show that parameter-level pruning generally provides better performance on NLP models, although the improvement varies across merging methods. Parameter-level pruning achieves higher H-scores for Task Arithmetic, PCB, and FR-Merging, while layer-level pruning performs slightly better for TIES. This indicates that the improvement does not come from a different pruning mechanism, but from applying the same diversity-based criterion with a granularity that better matches the model characteristics.
>
> For example, in PCB, layer-level pruning improves ID accuracy from 75.9 to 76.2 but slightly decreases OOD accuracy from 49.4 to 49.0, resulting in a lower H-score. This suggests that coarse-grained pruning may remove some potentially transferable parameters together with redundant updates. In contrast, parameter-level pruning provides a finer selection of task-vector components and achieves the highest H-score of 60.2.
>
> These results show that the diversity-based criterion of LwPTV can be applied across different modalities, while the pruning granularity should be selected according to the structural characteristics of the model.
>
>
>
> **Table 2: NLP results comparing original methods, parameter-level pruning, and layer-level pruning.**
>
>
> | **Merging Method** | **Variant** | **ID** | **OOD** | **H-score** |
> |:---|:---|:---:|:---:|:---:|
> | Task Arithmetic | Original | 73.3 | 43.9 | 54.9 |
> |  | Layer-level pruning | 75.8 | 44.8 | 56.3 |
> |  | Parameter-level pruning | **76.3** | **46.6** | **57.9** |
> |---|---|---:|---:|---:|
> | TIES | Original | 71.0 | 48.0 | 57.3 |
> |  | Layer-level pruning | **72.0** | **50.1** | **59.1** |
> |  | Parameter-level pruning | 71.9 | 49.8 | 58.8 |
> |---|---|---:|---:|---:|
> | PCB | Original | 75.9 | 49.4 | 59.8 |
> |  | Layer-level pruning | **76.2** | 49.0 | 59.7 |
> |  | Parameter-level pruning | 75.8 | **49.9** | **60.2** |
> |---|---|---:|---:|---:|
> | FR-Merging | Original | 73.9 | 43.9 | 50.1 |
> |  | Layer-level pruning | 75.4 | 43.9 | 55.5 |
> |  | Parameter-level pruning | **75.4** | **44.4** | **55.9** |
>
>
>
>
>
> **Response to C6 about improving writing quality and notation clarity:** Thank you for pointing this out. We have polished the manuscript throughout and corrected the expressions mentioned above to improve clarity and professionalism. We also checked the mathematical notation and corrected Eq.5 by replacing the reused summation index $k$ with $j$ in the  cross-task mean calculation.

---

> ### Author Response · Authors · 2026-07-18
> **Official Comment by Authors (part 4)**
>
> **Response to C7 about comparing LwPTV with CART:**  Thank you for the suggestion. We evaluated CART under the same ID/OOD protocol on ViT-B/32, as shown in Tab.3. CART gives the best ID Avg among the compared methods 84.7, which is consistent with its goal of preserving the shared component through centered low-rank reconstruction. Its OOD Avg, however, is 56.1, below the LwPTV variants. For instance, LW-AdaMerging w/LwPTV reaches an OOD Avg of 61.7 and an H-score of 68.6. These results show the different trade-offs. CART preserves ID accuracy better, while LwPTV achieves stronger OOD performance under the same benchmark. Since CART already uses centered task vectors, the gap in OOD accuracy suggests that LwPTV's gains are not solely a consequence of centering. The layer-wise pruning step appears to add a distinct effect by removing low-salience task-vector components instead of centered-task-vector processing.
>
>
> **Table 3: Comparison with CART under the same ID/OOD protocol on ViT-B/32.**
>
> | **Method** | **ID Avg** | **OOD Avg** | **H-score** |
> |:---|:---:|:---:|:---:|
> | CART | 84.7 | 56.1 | 67.5 |
> | Task Arithmetic w/ LwPTV | 72.8 | 60.9 | 66.3 |
> | LW-AdaMerging w/ LwPTV | 77.3 | 61.7 | 68.6 |
> | LW-AdaMerging++ w/ LwPTV | 77.1 | 61.4 | 68.4 |
>
>
>
>
>
>
>
>
>
> **Response to C8 about reporting OOD results for Fisher Merging and RegMean:**  Thank you for the suggestion. We have added the ID, OOD, and H-score results for Fisher Merging and RegMean in Tab. 2 of the revised paper, following the same evaluation protocol used for the other merging methods.
>
> LwPTV is designed for task-vector-based merging. Specifically, it prunes task-vector tensor updates according to their cross-task deviations before merging. Fisher Merging and RegMean are not task-vector-based methods: Fisher Merging aggregates parameters based on Fisher information, while RegMean estimates merged parameters using feature statistics. Since they do not explicitly construct task vectors, LwPTV is not directly applicable to these methods in its current formulation. We therefore include them as additional baselines and compare them with task-vector-based methods using LwPTV.
>
>
>
>
> On ViT-B/32, Fisher Merging achieves an ID accuracy of 68.3, an OOD accuracy of 60.8, and an H-score of 64.3, while RegMean achieves 71.8, 59.7, and 65.2, respectively. In comparison, Task Arithmetic with LwPTV achieves 72.8, 60.9, and 66.3, outperforming both Fisher Merging and RegMean in ID accuracy, OOD accuracy, and H-score.
>
> The results show the same trend on ViT-L/14, where Task Arithmetic with LwPTV achieves higher OOD accuracy and H-score than both Fisher Merging and RegMean. These results indicate that LwPTV achieves a better ID--OOD trade-off, as reflected by the improved H-score.
>
>
>
>
>
>
>
> **Response to C9 about strengthening the representation analysis beyond the t-SNE visualization:** Thank you for the helpful suggestion. We have revised the discussion of Fig.7 and now present the t-SNE results only as a qualitative illustration, rather than as strong evidence that LwPTV improves representation quality.
>
> We further add a quantitative representation analysis on ViT-B/32 to quantitatively examine how different merging methods affect the pretrained representations. Specifically, we extract the final-layer image embeddings before the classification head on CIFAR-10 and STL-10, and measure their distances to the embeddings of the pretrained model. We evaluate several metrics, including $\ell_2$ distance, cosine distance, centered L2 distance, mean shift, variance shift, and CKA distance. As shown in the newly added Fig.2 [link](https://anonymous.4open.science/r/LwPTV-1EEF/TMLR_Rebuttal__Figure.pdf)  (Fig.8 in the revised manuscript), Task Arithmetic+LwPTV consistently produces smaller representation distances than Task Arithmetic across both CIFAR-10 and STL-10.
>
> These results indicate that the OOD gains of LwPTV are related to better preservation of pretrained representations. Compared with Task Arithmetic, LwPTV keeps the merged model embeddings closer to those of the pretrained model and results in lower feature distortion introduced by task-vector updates.

---

> ### Author Response · Authors · 2026-07-18
> **Official Comment by Authors (part 5)**
>
> **Response to C10 about clarifying the relationship between LwPTV, CART, and generic shrinkage:** Thank you for the insightful suggestion. We agree that the relationship between generic shrinkage, CART, and LwPTV should be clarified. We have added a clearer discussion in Appendix B.20, to separate generic shrinkage toward the pretrained model, centered-task-vector processing in CART, and salience-based pruning in LwPTV.
> First, generic shrinkage toward the pretrained model reduces the overall magnitude of the merged task vector: $\theta_\alpha=\theta_{\mathrm{pre}}+\alpha(\theta_{\mathrm{merged}}-\theta_{\mathrm{pre}}), \quad \alpha < 1$. This can improve OOD robustness because the merged model remains closer to the pretrained model. However, this operation is task-agnostic: it uniformly suppresses all task-vector updates, including both useful and harmful ones. As shown in Tab.4, such shrinkage can improve OOD accuracy, but it does not reliably preserve ID performance.
>
> Second, CART uses centered task vectors and low-rank reconstruction. In the implementation we examined, CART first constructs a cross-task mean model, $\theta_{\mathrm{avg}} = \theta_0 + \frac{1}{T}\sum_i \tau_i$, then centers each task-specific model around this mean and applies low-rank reconstruction to the centered residuals. The reconstructed centered components are finally added back to $\theta_{\mathrm{avg}}$. Thus, CART mainly relies on cross-task centering and low-rank reconstruction to preserve shared in-domain components and reduce task interference. As shown in Tab.5, CART achieves the highest ID Avg of 84.7, which is consistent with its strength in preserving shared in-domain information. However, its OOD Avg is 56.1, lower than all LwPTV variants.
>
>
>
>
>
> LwPTV follows a different mechanism. It neither uniformly shrinks the merged update nor reconstructs centered residuals. Instead, it estimates the salience of task-vector tensors and selectively prunes low-salience tensors before merging. As shown in Tab.5, LW-AdaMerging w/ LwPTV achieves an OOD Avg of 61.7 and the highest H-score of 68.6. Overall, CART better retains ID performance, while LwPTV achieves a stronger ID--OOD trade-off.
>
>
>
>
> **Table 4: Comparison with the post-merge interpolation baseline on ViT-B/32.**
>
> | **Merging Method** | **Variant** | **$\alpha$** | **ID Avg** | **OOD Avg** | **H-score** |
> |:---|:---|:---:|:---:|:---:|:---:|
> | Task Arithmetic | Original merge | 1.0 | 69.1 | 51.3 | 58.9 |
> |  | Post-interp. baseline | 0.9 | 70.1 | 54.0 | 61.0 |
> |  | w/ LwPTV | -- | **72.8** | **60.9** | **66.3** |
> |  |  |  |  |  |  |
> | Ties | Original merge | 1.0 | **72.9** | 58.0 | 64.6 |
> |  | Post-interp. baseline | 0.9 | 72.0 | 59.0 | 64.8 |
> |  | w/ LwPTV | -- | 72.7 | **61.3** | **66.5** |
> |  |  |  |  |  |  |
> | PCB | Original merge | 1.0 | 75.8 | 53.2 | 62.5 |
> |  | Post-interp. baseline | 0.9 | 68.5 | **60.7** | 64.4 |
> |  | w/ LwPTV | -- | **77.3** | 59.7 | **67.4** |
>
>
>
>
> **Table 5: Comparison with CART under the same ID/OOD protocol on ViT-B/32.**
>
> | **Method** | **ID Avg** | **OOD Avg** | **H-score** |
> |:---|:---:|:---:|:---:|
> | CART | 84.7 | 56.1 | 67.5 |
> | Task Arithmetic w/ LwPTV | 72.8 | 60.9 | 66.3 |
> | LW-AdaMerging w/ LwPTV | 77.3 | 61.7 | 68.6 |
> | LW-AdaMerging++ w/ LwPTV | 77.1 | 61.4 | 68.4 |

---

> > ### Comment · Reviewer_pKUo · 2026-07-21
> >
> > Thank you for the detailed response and the additional experiments. I appreciate that the authors have addressed my other concerns, including the clarification of the method, the additional baselines, and the revised positioning of the claims. I now have one remaining question about the boundary of the proposed mechanism.
> >
> > In Table 5, CART achieves a much higher ID Avg than the LwPTV variants, although its OOD Avg is lower. In Table 4, post-merge interpolation toward the pretrained model improves the ID/OOD trade-off for several merged models. This suggests a natural and potentially strong baseline: using CART as the merged model and then applying post-merge interpolation,
> >
> > $$
> > \theta_{\mathrm{CART\text{-}interp}} =
> > \theta_{\mathrm{pre}} + \alpha(\theta_{\mathrm{CART}}-\theta_{\mathrm{pre}}).
> > $$
> >
> > This comparison is important because CART starts from a very strong ID point. If a moderate interpolation coefficient can substantially improve CART's OOD Avg while retaining much of its ID advantage, CART + interpolation may achieve a trade-off close to, or possibly better than, LW-AdaMerging w/ LwPTV. Conversely, if LwPTV still dominates this baseline, it would provide much stronger evidence that salience-based selective pruning offers benefits beyond generic shrinkage toward the pretrained model and beyond centered-task-vector processing.
> >
> > My intention is not to require the authors to show that LwPTV is always superior to every possible variant. Rather, I would like to understand the success boundary of this family of methods. The current results suggest three related mechanisms: (i) generic shrinkage toward the pretrained model, (ii) centered-task-vector processing as in CART, and (iii) salience-based selective pruning as in LwPTV. A CART + post-interpolation baseline would help disentangle these mechanisms and clarify when LwPTV is genuinely needed, rather than when a strong ID-oriented merge followed by simple interpolation is sufficient.
> >
> > I therefore encourage the authors to add a small alpha sweep for CART + post-merge interpolation under the same ID/OOD protocol, ideally reporting the full ID/OOD/H-score trade-off curve rather than only the best selected point. This would substantially strengthen the paper's mechanistic explanation and make the empirical findings more useful to the TMLR audience.

---

> > > ### Author Response · Authors · 2026-07-22
> > >
> > > **Table 6: CART with post-merge interpolation on ViT-B/32.**
> > >
> > > | $\alpha$ | ID Avg. | OOD Avg. | H-score |
> > > |:--------:|:-------:|:--------:|:-------:|
> > > | 0.1 | 55.6 | 61.9 | 58.6 |
> > > | 0.2 | 62.4 | 62.1 | 62.3 |
> > > | 0.3 | 68.3 | 62.0 | 65.0 |
> > > | 0.4 | 73.3 | 61.7 | 67.0 |
> > > | 0.5 | 77.3 | 61.3 | 68.3 |
> > > | 0.6 | 80.2 | 60.6 | 69.1 |
> > > | 0.7 | 82.5 | 59.8 | 69.3 |
> > > | 0.8 | 83.8 | 58.8 | 69.1 |
> > > | 0.9 | 84.5 | 57.5 | 68.5 |
> > > | 1.0 | 84.7 | 56.1 | 67.5 |
> > >
> > >
> > >
> > >
> > >
> > >
> > >
> > >
> > > **Table 7: Comparison of baseline methods and their variants with LwPTV on ViT-B/32.**
> > >
> > > | Method | ID Avg | OOD Avg | H-score |
> > > |:---|:---:|:---:|:---:|
> > > | Task Arithmetic | 69.1 | 51.3 | 58.9 |
> > > | Task Arithmetic w/ LwPTV | 72.8 (+3.7) | 60.9 (+9.6) | 66.3 (+7.4) |
> > > | --- | --- | --- | --- |
> > > | TIES | 72.9 | 58.0 | 64.6 |
> > > | TIES w/ LwPTV | 72.7 (-0.2) | 61.3 (+3.3) | 66.5 (+1.9) |
> > > | --- | --- | --- | --- |
> > > | TW AdaMerging | 71.1 | 53.2 | 60.9 |
> > > | TW AdaMerging w/ LwPTV | 75.4 (+4.3) | 56.5 (+3.3) | 64.6 (+3.7) |
> > > | --- | --- | --- | --- |
> > > | TW AdaMerging++ | 73.7 | 53.7 | 62.1 |
> > > | TW AdaMerging++ w/ LwPTV | 76.1 (+2.4) | 56.6 (+2.9) | 64.9 (+2.8) |
> > > | --- | --- | --- | --- |
> > > | LW AdaMerging | 80.1 | 57.7 | 67.1 |
> > > | LW AdaMerging w/ LwPTV | 77.3 (-2.8) | 61.7 (+4.0) | 68.6 (+1.5) |
> > > | --- | --- | --- | --- |
> > > | LW AdaMerging++ | 81.1 | 57.6 | 67.4 |
> > > | LW AdaMerging++ w/ LwPTV | 77.1 (-4.0) | 61.4 (+3.8) | 68.4 (+1.0) |
> > > | --- | --- | --- | --- |
> > > | PCB-Merging | 75.8 | 53.2 | 62.5 |
> > > | PCB-Merging w/ LwPTV | 77.3 (+1.5) | 59.7 (+6.5) | 67.4 (+4.9) |
> > > | --- | --- | --- | --- |
> > > | FR-Merging | 78.1 | 55.4 | 64.8 |
> > > | FR-Merging w/ LwPTV | 74.7 (-3.4) | 60.6 (+5.2) | 66.9 (+2.1) |
> > >
> > >
> > > **Response to the comment on CART with post-merge interpolation:**
> > >
> > > **(a) CART interpolation provides a strong baseline but requires OOD-aware coefficient selection.** We thank the reviewer for suggesting the CART + post-merge interpolation baseline. We agree that this comparison helps disentangle generic shrinkage from selective pruning. Following the suggestion, we perform an $\alpha$ sweep for CART interpolation. As shown in Tab.6, CART interpolation indeed improves the ID/OOD trade-off. For example, $\alpha=0.7$ achieves 82.5 ID Avg, 59.8 OOD Avg, and 69.3 H-score. However, the best $\alpha$ is selected based on the ID/OOD/H-score sweep, which assumes access to OOD evaluation information. In contrast, LwPTV selects $\eta$ using only ID  performance, while OOD datasets remain unseen. Under an ID-only setting, a conservative choice such as $\alpha=0.9$ achieves 84.5 ID Avg, 57.5 OOD Avg, and 68.5 H-score, which is comparable to LW-AdaMerging w/ LwPTV (68.6 H-score).
> > >
> > > **(b) LwPTV provides selective pruning beyond uniform shrinkage.** CART interpolation applies uniform scaling to the entire task vector, which reduces all task-specific updates equally and mainly adjusts the ID/OOD trade-off through task-vector magnitude shrinkage. In contrast, LwPTV selectively removes low-salience task-vector components. By suppressing potentially conflicting or less useful updates while retaining important task-specific components, LwPTV can improve OOD performance while largely maintaining ID performance. As shown in Tab.7, LwPTV consistently improves OOD performance and largely maintain or improves ID performance for several baselines.For example, Task Arithmetic with LwPTV improves ID Avg from 69.1 to 72.8 while improving OOD Avg from 51.3 to 60.9; PCB-Merging with LwPTV improves ID Avg from 75.8 to 77.3 while improving OOD Avg from 53.2 to 59.7. These results suggest that LwPTV provides benefits beyond uniform shrinkage by selectively modifying task-vector components rather than uniformly scaling all updates.
> > >
> > >
> > > **\(c\) Comparison under similar ID degradation.** To further compare the OOD improvement efficiency, we compare LwPTV with FR-Merging under a similar setting. We choose FR-Merging as the reference because, similar to CART interpolation, it does not require additional coefficient optimization and improves OOD performance at the cost of ID degradation. Therefore, this comparison focuses on how effectively different post-hoc strategies improve OOD robustness while maintaining ID performance.
> > >
> > > For FR-Merging, applying LwPTV improves OOD Avg from 55.4 to 60.6 (+5.2) with an ID degradation of 3.4 points. In comparison, CART interpolation with $\alpha=0.6$ improves OOD Avg from 56.1 to 60.6 (+4.5), but introduces a larger ID degradation of 4.5 points (from 84.7 to 80.2). This suggests that LwPTV achieves a more favorable ID/OOD trade-off by selectively pruning task-vector components rather than uniformly scaling all task-specific updates.

---

### Review · Reviewer_V3Rv · 2026-07-06

**Summary Of Contributions:**

This paper studies improving out-of-domain generalization in multi-task model merging and proposes LwPTV, a layer-wise pruning method for task vectors. LwPTV computes a layer-wise salience score by measuring how much each task vector deviates from the cross-task mean, so that low-salience layers are treated as less task-specific. It then constructs task-specific masks and combines them with an OR operation, aiming to remove only layers that appear redundant across all tasks while retaining layers needed by at least one task. The resulting mask can be plugged into several task-vector-based merging methods, with an additional scaling strategy for layer-wise AdaMerging variants. Experiments on vision and NLP model-merging benchmarks show that LwPTV generally improves OOD performance and H-score while largely preserving ID performance.

---

**Strengths:**
1. The paper studies an important but relatively underexplored issue in model merging.
2. The proposed salience score is simple, training-free, and easy to combine with existing task-vector-based merging methods.
3. The experiments cover several representative merging baselines and show consistent gains in H-score.
4. The paper is well-written and easy to follow.

---

**Weaknesses:**
1. The central assumption that low cross-task diversity corresponds to redundancy is plausible, but it is not fully established. Shared directions across task vectors may also encode genuinely useful transferable features, especially when tasks are semantically related. The authors may want to more carefully distinguish "redundant shared shifts" from "shared useful adaptations", and discuss when pruning common task-vector components could hurt generalization.
2. The theoretical analysis is only loosely connected to the actual algorithm. Proposition 1 is based on a simplified one-layer Transformer setting and neuron-level discriminative features, while the method performs layer-wise pruning in large pretrained ViTs and parameter-wise pruning for NLP. The authors may want to clarify which parts of the theory directly justify the practical salience score and which parts should be viewed only as motivation.
3. The paper does not sufficiently connect LwPTV to recent studies on multi-domain adaptation and optimization conflicts. Recent works such as [1] and [2] study related issues from complementary angles: [1] analyzes evolving interactions between samples during multi-domain fine-tuning, while [2] addresses cross-domain conflicts in multi-domain RL through gradient and curvature-guided interactions. Although these works focus on training-time adaptation rather than post-hoc model merging, they are relevant because the current paper also aims to manage interactions among task-specific adaptations and improve multi-domain/OOD behavior. Please cite and discuss [1] and [2] to better position LwPTV within this broader line of work. If adding experimental comparisons is outside the scope, at least a detailed method-level discussion would be useful, especially on how task-vector diversity, sample-level interactions, and gradient-level cross-domain conflicts are related or different.
4. The OR-based shared mask is designed to preserve ID performance, but it may also make the pruning rule conservative and task-count dependent. As the number of tasks grows, a layer only needs to be salient for one task to be retained, so the final mask may preserve many layers even if most tasks regard them as redundant. The authors may want to analyze this behavior more explicitly and discuss whether alternative aggregation rules, such as majority voting or weighted voting, would provide a better ID/OOD trade-off.
5. The method uses a fixed pruning ratio for the main vision experiments, but the best trade-off appears sensitive to the relation between ID and OOD distributions. Please provide a clearer guideline for choosing the pruning ratio in practice, especially when OOD validation data are unavailable.

---

**References:**

[1] Boosting Multi-Domain Fine-Tuning of Large Language Models through Evolving Interactions between Samples. *ICML 2025*.

[2] Boosting Multi-Domain Reasoning of LLMs via Curvature-Guided Policy Optimization. *ICLR 2026*.

**Audience:**

Yes

**Audience Explanation:**

Researchers working on model merging, multi-task learning, parameter-efficient reuse of fine-tuned models, and OOD robustness would likely find the paper relevant. The method is simple and practical, and the paper focuses on an important limitation of current model-merging methods.

**Broader Impact Concerns:**

I do not see serious broader impact concerns. The work is mainly methodological and aims to improve model merging and OOD robustness. A possible positive impact is reducing the need to store or deploy many task-specific models. A possible concern is that more efficient model merging could make it easier to combine models whose training data or behaviors are not fully understood, so careful evaluation remains important before deployment.

**Claims And Evidence:**

Yes

**Claims Explanation:**

The main empirical claim, that LwPTV can improve OOD performance and H-score for several task-vector-based merging methods, is reasonably supported by the reported experiments. The paper includes main results, ablations, pruning baselines, OOD-oriented comparisons, larger ViT experiments, and an NLP extension. However, the mechanistic claim that low-salience layers are truly redundant and that pruning them recovers pretrained generalizable features is less fully established. The evidence supports this interpretation, but does not completely prove it.

**Requested Changes:**

1. Please provide a more careful discussion of the salience-score assumption. In particular, the authors should clarify why low cross-task diversity should be interpreted as redundancy rather than shared transferable adaptation. It would also be helpful to discuss cases where pruning low-diversity task-vector components may hurt performance, especially when the merged tasks are semantically related.
2. Please clarify the relationship between the theoretical analysis and the practical algorithm. The paper should explicitly state the gap between the simplified one-layer, neuron-level setting in Proposition 1 and the layer-wise pruning strategy used for large ViTs. The authors should also explain why the same intuition remains applicable when the method is changed to parameter-wise pruning for NLP tasks.
3. Please cite and discuss [1] and [2] in the related work or discussion section. The authors should clarify that LwPTV is a post-hoc, training-free model merging method that prunes task-vector layers, whereas [1] manages multi-domain data during fine-tuning through evolving sample interactions, and [2] modifies the optimization process to induce beneficial cross-domain gradient/curvature interactions. The discussion could compare these works along several dimensions, including static post-hoc pruning versus dynamic training-time adaptation, task-vector conflicts versus sample- or gradient-level conflicts, low-diversity directions as redundancy versus shared transfer, and whether these approaches could be complementary by first producing less conflicting task-specific checkpoints and then merging them more robustly.
4. Please analyze the OR-based shared mask more explicitly. Since the final mask retains a layer as long as it is selected by at least one task, the method may become increasingly conservative as the number or diversity of tasks grows. The authors may want to discuss this task-count dependence and, if possible, compare the OR rule with alternative aggregation strategies such as majority voting, weighted voting, or intersection-based masking.
5. Please provide practical guidance for selecting the pruning ratio $\eta$. Since the method is intended to be training-free and plug-and-play, readers need to know how to choose this hyperparameter when OOD validation data are unavailable. The authors may want to discuss whether $\eta$ can be selected based on ID validation performance, salience-score statistics, the number of tasks, or other observable quantities.

---

**References:**

[1] Boosting Multi-Domain Fine-Tuning of Large Language Models through Evolving Interactions between Samples. *ICML 2025*.

[2] Boosting Multi-Domain Reasoning of LLMs via Curvature-Guided Policy Optimization. *ICLR 2026*.

---

> ### Author Response · Authors · 2026-07-18
> **Official Comment by Authors (part 1)**
>
> Dear Reviewer V3Rv. Thank you for your time and effort in reviewing our paper. We appreciate your positive comments on the importance of this relatively underexplored issue, the simplicity and training-free nature of the proposed salience score, its compatibility with existing task-vector-based merging methods, and the consistent H-score gains across representative baselines. We also appreciate your comments on the clarity of the paper. Below, we respond to your comments in detail.
>
>
> **Response to C1 about clarifying the salience-score assumption and redundancy interpretation:**  Thank you for this helpful suggestion. We provide a more careful explanation of why low cross-task diversity be interpreted as redundancy in our setting. A low-diversity component means that different task vectors update this component in similar directions. We distinguish two cases. If such a component also has low mean magnitude, it is more likely to contain limited task-specific information, and pruning it is relatively safe.
>
> The more subtle case is high mean magnitude but low cross-task diversity. These components may look important from magnitude alone, but their low diversity suggests that they do not distinguish different task adaptations. We examined this case in Appendix B.15. On ViT-B/32, we identified 17 layers with high mean absolute values and low cross-task variance, all of which are pruned by LwPTV. Among them, 16 are bias layers and the remaining one is `model.ln_final.weight`. We evaluated two variants: V1 keeps all 17 layers, and V2 keeps the 16 bias layers while pruning only the final layer normalization weight. As shown in Tab.1, both variants perform very similarly to LwPTV and do not bring a meaningful improvement. This suggests that these high-magnitude but low-diversity layers are not critical to the final ID--OOD trade-off. To understand why these high-magnitude but low-diversity components can be safely pruned, we further analyze their parameter types. For a linear layer $f(x)=Wx+b$, the weight matrix $W$  determines input-dependent transformations, whereas the bias term $b$ mainly introduces an input-independent shift. Therefore, large but similar bias updates across tasks may behave more like common calibration shifts rather than task-discriminative adaptations.
>
>
> **Table 1: Performance comparison on ViT-B/32.**
>
> | **Method (↓)** | **ID** | **OOD** | **H-score** |
> |:---|:---:|:---:|:---:|
> | Task Arithmetic | 69.1 | 51.3 | 58.9 |
> | V1 (keep 17 layers) | 72.9 | 61.0 | 66.4 |
> | V2 (keep 16 layers) | 72.9 | 61.0 | 66.4 |
> | Ours | 72.8 | 60.9 | 66.3 |
>
>
>
>
>
> **Response to C2 about clarifying the relationship between theoretical analysis and practical algorithm:**  Thank you for pointing out this gap. We agree that Proposition 1 is derived under a simplified one-layer, neuron-level setting, whereas the practical algorithm is applied to large ViTs with tensor-level pruning. We clarify that the purpose of Proposition 1 is to motivate the pruning criterion by revealing the relationship between task-vector diversity and task-specific information, rather than to provide a direct theoretical guarantee for a specific pruning granularity.
>
> The main implication of Proposition 1 is that task-vector components containing task-specific information may exhibit larger cross-task diversity across tasks, while less task-specific components may show lower diversity. Based on this observation, LwPTV extends the diversity measurement from individual neurons to parameter tensors in deep models. Specifically, we aggregate the deviations within each tensor to obtain the salience score and use it to identify task-vector components with lower cross-task diversity for pruning. Therefore, although the practical pruning unit is different from the theoretical analysis, the underlying principle remains the same.
>
> For NLP experiments, we further apply this principle with a finer pruning granularity. While the pruning unit differs from the ViT setting, the criterion is unchanged: task-vector components with low cross-task deviation are considered less task-specific and are replaced with the corresponding pretrained parameters.

---

> ### Author Response · Authors · 2026-07-18
> **Official Comment by Authors (part 2)**
>
> **Response to C3 about clarifying the relationship between LwPTV and multi-domain adaptation methods:** Thank you for the suggestion. We agree that the relationship between LwPTV and recent multi-domain adaptation methods should be clarified. We have added a new discussion section (Sec.5.3, Discussion with Multi-domain Adaptation Methods) to compare LwPTV with [1] and [2].
>
> In this discussion, we clarify that LwPTV is a post-hoc, training-free model merging method that prunes task-vector components after task-specific checkpoints are obtained, whereas [1] and [2] improve multi-domain models during training through sample-level interactions and gradient/curvature-aware optimization, respectively. We also discuss how these methods differ in their optimization stages and objectives, and how they can be complementary: training-time adaptation methods can  provide a better starting point for model merging, while LwPTV can further improve the merged model by pruning potentially redundant task-vector components.

---

> ### Author Response · Authors · 2026-07-18
> **Official Comment by Authors (part 3)**
>
> **Response to C4 about the task-count dependence, and aggregation strategies:** Thank you for the insightful suggestion. We agree that the effect of the OR-based shared mask with different numbers of tasks should be clarified. Although the OR rule is conservative because a layer is retained if it is selected by at least one task, our experiments show that increasing the number of ID tasks does not reduce the effectiveness of LwPTV. Instead, the relative H-score improvement over Task Arithmetic becomes larger as the number of ID tasks increases.
>
> Specifically, we analyze the effect of the number of ID tasks in Appendix B.15 by varying the number of ID tasks $K$ from 2 to 8 on ViT-B/32. As shown in Tab.2, LwPTV consistently improves the H-score over Task Arithmetic for all values of $K$. The improvement increases from $+2.0$ at $K=2$ to $+7.4$ at $K=8$. This trend is consistent with Proposition 1: the salience score measures the deviation from the cross-task mean to identify task-specific components with high diversity, and a larger number of tasks provides a more reliable estimate of the shared adaptation pattern. Therefore, more tasks lead to a clearer distinction between redundant and task-specific updates, resulting in more effective pruning and larger OOD gains.
>
> To further examine the choice of mask aggregation, we compare the OR rule with three alternative strategies in Tab.3: majority voting, weighted voting, and intersection masking. Majority voting retains a layer when it is selected by at least half of the tasks, weighted voting aggregates task votes using weights computed from the accuracy gain of each fine-tuned model over the pretrained model, and intersection masking retains only layers selected by all tasks. These strategies cover different levels of masking strictness, from the more inclusive OR rule to the more restrictive intersection rule.
>
> The results show that OR achieves the best overall performance among different aggregation strategies. It achieves the highest ID average for Task Arithmetic, TIES, and PCB, and obtains the best H-score for TIES and PCB while remaining close to the best result for Task Arithmetic. In contrast, intersection masking generally leads to lower OOD and H-score performance, indicating that overly strict aggregation may remove updates that are useful for certain tasks. Majority and weighted voting achieve comparable performance, but they introduce additional assumptions. Majority voting treats all tasks equally and may discard task-specific updates that are important for individual tasks, while weighted voting requires estimating task weights. In contrast, the OR rule preserves updates selected by any task without introducing additional weighting schemes. Overall, the OR rule provides a reasonable trade-off between retaining task-relevant updates and removing redundant components.
>
>
>
> **Table 2: Relative H-score improvement as the number of ID tasks changes on ViT-B/32.**
>
> | **Task Num.** | **2** | **3** | **4** | **5** | **6** | **7** | **8** |
> |:---:|:---:|:---:|:---:|:---:|:---:|:---:|:---:|
> | **H-score** | +2.0 | +3.3 | +3.5 | +3.8 | +4.5 | +6.0 | +7.4 |
>
>
> **Table 3: Comparison of different mask aggregation strategies within LwPTV on ViT-B/32.**
>
> | **Method** | **Mask Aggregation** | **ID Avg.** | **OOD Avg.** | **H-score** |
> |:---|:---|:---:|:---:|:---:|
> | Task Arithmetic | w/ LwPTV (Majority) | 72.3 | 61.1 | 66.3 |
> |  | w/ LwPTV (Weighted) | 72.6 | **61.2** | **66.4** |
> |  | w/ LwPTV (Intersection) | 72.3 | 59.5 | 65.3 |
> |  | w/ LwPTV (OR) | **72.8** | 60.9 | 66.3 |
> |---|---|---:|---:|---:|
> | Ties | w/ LwPTV (Majority) | 72.2 | 61.2 | 66.3 |
> |  | w/ LwPTV (Weighted) | 72.4 | 61.3 | 66.4 |
> |  | w/ LwPTV (Intersection) | 72.5 | 60.6 | 66.0 |
> |  | w/ LwPTV (OR) | **72.7** | **61.3** | **66.5** |
> |---|---|---:|---:|:---:|
> | PCB | w/ LwPTV (Majority) | 76.8 | **59.8** | 67.2 |
> |  | w/ LwPTV (Weighted) | 77.0 | 59.8 | 67.3 |
> |  | w/ LwPTV (Intersection) | 76.8 | 58.1 | 66.1 |
> |  | w/ LwPTV (OR) | **77.3** | 59.7 | **67.4** |

---

> ### Author Response · Authors · 2026-07-18
> **Official Comment by Authors (part 4)**
>
> **Response to C5 about the selection and practical guidance of the pruning ratio $\eta$:** Thank you for the suggestion. We agree that the paper should provide clearer guidance on how to choose the pruning ratio $\eta$. We also clarify that $\eta$ was not selected using OOD validation data.
>
> Our goal is to improve the merged model while preserving its ID performance as much as possible. Therefore, we select $\eta$ using only ID-side information.  In the original experiments, we selected $\eta$ on ViT-B/32 using Task Arithmetic as the base merging method. This selection was based only on ID performance and did not use any OOD evaluation datasets. As shown in Fig.1(a)  [link](https://anonymous.4open.science/r/LwPTV-1EEF/TMLR_Rebuttal__Figure.pdf) (Fig.5 in the revised manuscript), Task Arithmetic w/ LwPTV achieves its best ID accuracy at $\eta=0.7$. This value also brings a clear OOD improvement, giving a favorable ID--OOD trade-off. We then used $\eta=0.7$ as the default value for the remaining experiments and applied it to other merging baselines, where it also led to consistent gains in the final evaluation.
>
> To further check this default choice, we conduct an additional sweep of $\eta$ on ViT-L/14. As shown in Fig.1(b) [link](https://anonymous.4open.science/r/LwPTV-1EEF/TMLR_Rebuttal__Figure.pdf) (Fig.12 in Appendix B.19 of the revised manuscript), $\eta=0.7$ remains near-optimal across different merging baselines. For Task Arithmetic, the best H-score is achieved around $\eta=0.6$--$0.7$, with both settings reaching 75.7. For TIES, the best result is obtained at $\eta=0.7$, with an H-score of 76.1. For PCB-Merging, $\eta=0.6$ and $\eta=0.7$ both achieve the best H-score of 76.4. These results suggest that $\eta=0.7$ is not specific to ViT-B/32 and also works as a strong default on ViT-L/14.
>
> In practice, if a small ID validation set is available, we recommend sweeping a narrow range around 0.6--0.8 and selecting $\eta$ based on ID performance or the desired ID-side constraint. OOD datasets should be reserved for final evaluation and should not be used to select $\eta$. If no validation data are available, $\eta=0.7$ is a reasonable default based on our results on both ViT-B/32 and ViT-L/14.

---

> > ### Comment · Reviewer_V3Rv · 2026-07-21
> > **Thank you for your detailed responses.**
> >
> > Thank you for your detailed responses, which have addressed most of my concerns.
> >
> > Best regards,
> >
> > Reviewer V3Rv